# Pharmacological Gq inhibition induces strong pulmonary vasorelaxation and reverses pulmonary hypertension

Alexander Seidinger [1], Richard Roberts[2], Yan Bai[3], Marion Müller [4,5], Eva Pfeil[6], Michaela Matthey[1], Sarah Rieck [7], Judith Alenfelder [6], Gabriele M König[8], Alexander Pfeifer [9], Evi Kostenis[6], Anna Klinke[4,5], Bernd K Fleischmann [7] & Daniela Wenzel [1,7]✉

## Abstract

**Pulmonary arterial hypertension (PAH) is a life-threatening disease with limited survival. Herein, we propose the pharmacological inhibition of Gq proteins as a novel concept to counteract pulmonary vasoconstriction and proliferation/migration of pulmonary artery smooth muscle cells (PASMCs) in PAH. We demonstrate that the specific pan-Gq inhibitor FR900359 (FR) induced a strong vasorelaxation in large and small pulmonary arteries in mouse, pig, and human subjects ex vivo. Vasorelaxation by FR proved at least as potent as the currently used triple therapy. We also provide in vivo evidence that local pulmonary application of FR prevented right ventricular systolic pressure increase in healthy mice as well as in mice suffering from hypoxia (Hx)-induced pulmonary hypertension (PH). In addition, we demonstrate that chronic application of FR prevented and also reversed Sugen (Su)Hx-induced PH in mice. We also demonstrate that Gq inhibition reduces proliferation and migration of PASMCs in vitro. Thus, our work illustrates a dominant role of Gq proteins for pulmonary vasoconstriction as well as remodeling and proposes direct Gq inhibition as a powerful pharmacological strategy in PH.**

**Keywords** Vasoconstriction; Pulmonary Arterial Hypertension; Gq Inhibition
**Subject Category** Vascular Biology & Angiogenesis

## Introduction

G-Protein coupled receptors (GPCRs) are the largest family of drug targets with approximately one third of all approved drugs targeting these receptors (Santos et al, 2017). In particular, GPCRs activating Gq proteins are key modulators of pulmonary arterial tone (Strassheim et al, 2018). Sustained vasoconstriction and vascular remodeling can result in the persistent elevation of pulmonary blood pressure, which is a typical feature of PAH. This disease is characterized by a mean pulmonary blood pressure above 20 mmHg, a pulmonary vascular resistance >2 Wood units and a pulmonary arterial wedge pressure of ≤15 mmHg (Yaghi et al, 2020; Humbert et al, 2022) and has a poor prognosis (Hendriks et al, 2022; Chang et al, 2022). Biochemical evidence reveals that Gq signaling is central in the pathophysiology of PAH as the endothelin 1 (ET-1) system is upregulated (Stewart et al, 1991), serotonin (5-HT) activity enhanced (Hervé et al, 1995) and thromboxane metabolite levels are increased in PAH patients (Christman et al, 1992), likewise, antagonists of GPCRs activating Gq are also of therapeutic relevance (Rubin et al, 2002). Even though improved treatment strategies have resulted in better clinical outcomes (Vachiéry et al, 2019; Kramer et al, 2024), still a substantial number of PAH patients does not respond adequately to existing therapies (Hendriks et al, 2022; Chang et al, 2022). Current developments in PAH research focus on reverse-remodeling strategies in the pulmonary vasculature (Berghausen et al, 2021; Savai et al, 2014; Yung et al, 2020), but also optimized strategies to promote vasodilation are very important, as recent evidence demonstrates that reduction of pulmonary arterial pressure and right ventricular afterload are associated with better long-term survival (Vizza et al, 2022). Thus, novel efficient molecules target both, vasoconstriction and remodeling, are urgently required. The macrocyclic depsipeptide FR is a highly specific inhibitor of $G\alpha_{q/11/14}$ signaling (Schrage et al, 2015; Pfeil et al, 2020). It acts via prevention of GDP/GTP exchange that is crucial for the activation of G proteins. Given the potency and specificity of FR,

[1]Institute of Physiology, Department of Systems Physiology, Medical Faculty, Ruhr University of Bochum, Bochum, Germany. [2]Pharmacology Research Group, University Hospital of Nottingham, Nottingham, UK. [3]Division of Neonatology and Newborn Medicine, Department of Pediatrics, Massachusetts General Hospital and Harvard Medical School, Boston, USA. [4]Clinic for General and Interventional Cardiology/Angiology, Herz- und Diabeteszentrum NRW, University Hospital of the Ruhr University of Bochum, Bad Oeynhausen, Germany. [5]Agnes Wittenborg Institute for Translational Cardiovascular Research, Herz- und Diabeteszentrum NRW, University Hospital of the Ruhr University of Bochum, Bad Oeynhausen, Germany. [6]Molecular-, Cellular-, and Pharmacobiology Section, Institute of Pharmaceutical Biology, University of Bonn, Bonn, Germany. [7]Institute of Physiology I, Life&Brain Center, Medical Faculty, University of Bonn, Bonn, Germany. [8]Institute of Pharmaceutical Biology, University of Bonn, Bonn, Germany. [9]Institute of Pharmacology and Toxicology, University Hospital Bonn, University of Bonn, Bonn, Germany. ✉E-mail: daniela.wenzel@rub.de

we have investigated the impact of Gq inhibition by FR onto pulmonary vasoconstriction, proliferation, and migration as well as PH.

Herein, we demonstrate that FR acts as a powerful vasorelaxant of pulmonary arteries (PAs) in mice, humans, and pigs in various ex vivo models. In addition, the pulmonary application of FR in vivo diminished acute right ventricular systolic pressure (RVSP) increase induced by 5-HT in healthy mice and in animals suffering from Hx-induced PH. Importantly, FR was also able to prevent proliferation and migration of murine (m)PASMCs in vitro and it prevented and even reversed chronic SuHx-induced PH in mice.

# Results

## Single dose FR induces strong pulmonary vasorelaxation and enables to identify G protein subtypes involved in vasoconstriction

First, we assessed the effect of a single dose of the pan-Gq-inhibitor FR ($10^{-6}$ M) on vascular tone of mouse PAs in isometric force measurements in a wire-myograph. PAs were either constricted by Gq-dependent agonists, such as 5-HT, the thromboxane analog U-46619 or ET-1, or by Gq-independent depolarization via KCl. Upon pre-constriction with the Gq-dependent agonists, FR induced a pronounced vasorelaxation (Fig. 1A–C,F): After 5-HT-induced ($5 \times 10^{-7}$ M) pulmonary vasoconstriction FR evoked a strong vasorelaxation of $79.2 \pm 2.5\%$ ($n = 12$); the relaxing effect of FR after constriction with U-46619 ($10^{-7}$ M, $52.2 \pm 3.0\%$, $n = 15$) or ET-1 ($3 \times 10^{-9}$ M, $71.8 \pm 1.2\%$, $n = 13$) was still substantial, but lower than after 5-HT. As expected, neither the solvent DMSO nor FR affected KCl-induced ($4 \times 10^{-2}$ M) vasoconstriction (Fig. 1D–F). These results demonstrate that FR is a strong vasorelaxant in PAs after pre-constriction with pathophysiologically relevant pulmonary vasoconstrictors that signal via Gq. Because the effect of FR was less pronounced after U-44619- and ET-1-dependent constriction, we hypothesized that these agonists also signal via G proteins other than Gq. Because of earlier reports suggesting U-46619-mediated activation of $G\alpha_{12/13}$ in PAs (Alapati et al, 2007; McKenzie et al, 2009), we pre-incubated PAs with Y-27632 ($10^{-5}$ M), an inhibitor of Rho kinase (ROCK), which is a downstream mediator of $G\alpha_{12/13}$. This treatment reduced U-46619-dependent pre-constriction in PAs (w/o Y-27632: $3.9 \pm 0.2$ mN, $n = 15$, vs. with Y-27632: $2.6 \pm 0.3$ mN, $n = 6$, $p = 0.0185$) (Fig. 1B,G) and the subsequent FR application induced an almost complete relaxation ($89.1 \pm 1.3\%$, $n = 6$, $p < 0.0001$) of PAs (Fig. 1F,G). For ET-1-constriction, we first pre-incubated PAs with pertussis toxin (PTX, 1 µg/ml, 12 h) to inhibit potentially involved Gi/o proteins. This treatment did neither affect the extent of ET-1-induced constriction nor that of subsequent FR-induced vasorelaxation (Fig. 1C,F,H). This result demonstrates that ET-1-induced pulmonary vasoconstriction is independent of Gi/o. Pre-incubation with Y-27632 ($10^{-5}$ M), however, resulted in an almost complete vasorelaxation by FR ($95.7 \pm 1.3\%$, $n = 6$, $p < 0.0001$ vs ET-1 alone) (Fig. 1F,I). Thus, FR is a Gq-specific pharmacological inhibitor in PAs ex vivo and our results underscore the contribution of different G proteins in response to pulmonary vasoconstrictors. To further prove that FR abrogates Gq signaling completely and to compare the effect of FR with known receptor antagonists, we then performed experiments

in which PAs were pre-incubated with the vasorelaxant followed by dose–response curves of the vasoconstrictors.

## FR prevents Gq-dependent pulmonary vasoconstriction and modulates tone of murine and porcine PAs

We pre-treated PAs in the wire-myograph with FR ($10^{-6}$ M), the solvent DMSO or equal concentrations of the 5-HT$_{2A/2C}$ receptor antagonist Ketanserin (Ket, $10^{-6}$ M) (Frenken and Kaumann, 1984). Then, 5-HT dose–response curves ($10^{-9}$ M–$10^{-5}$ M) were generated. Ket as well as FR completely abolished 5-HT-induced constriction (Figs. EV1A and 2A,B) demonstrating that in PAs Ket and FR are similarly effective in blocking 5-HT-induced pulmonary vasoconstriction, a response that is entirely Gq-dependent.

The pan-Gq protein inhibitor FR inhibits $G\alpha_q$, $G\alpha_{11}$, and $G\alpha_{14}$. To identify the contribution of single Gq protein family members, we used genetic tools to ablate individual G proteins. First, expression of $G\alpha_q$, $G\alpha_{11}$, and $G\alpha_{14}$ was assessed by PCR in mPASMCs, as this cell type determines vascular tone in PAs. The analysis revealed expression of $G\alpha_q$ and $G\alpha_{11}$, but not $G\alpha_{14}$ (Fig. EV1B). Then, we used lentivirus containing small-hairpin (sh)- RNA-constructs (sh-ctrl, sh-Gq, sh-G11) to knock down either $G\alpha_{11}$, $G\alpha_q$ or both. QRT-PCR demonstrated specific downregulation of the target subunit (Fig. EV1C,D), no effect on the expression of other G protein subtypes such as $G\alpha_i$ (Fig. EV1E) or $G\alpha_s$ (Fig. EV1F) was observed. Analysis of protein expression shows the downregulation of $G\alpha_{11}$ but not $G\alpha_q$ as there are neither $G\alpha_{11}$ nor $G\alpha_q$-specific antibodies commercially available and an antibody against both $G\alpha_q$ and $G\alpha_{11}$ still detects the more prevalent $G\alpha_{11}$ isoform after $G\alpha_q$ downregulation in PASMCs (Fig. EV1G,H). After transduction, the cells were used for label-free dynamic mass redistribution (DMR) experiments. The specific 5-HT$_{2A/2C}$ agonist α-methyl-5-HT ($10^{-14}$ M–$10^{-6}$ M) induced strong dose-dependent responses in native and in sh-ctrl cells. The response to the highest α-methyl-5-HT concentration was similarly reduced by about 60% in sh-G11 and sh-Gq cells compared to sh-ctrl. MRNA and protein analysis revealed that sh-RNA treatment strongly reduced $G\alpha_q$ and $G\alpha_{11}$ expression, but can neither abrogate $G\alpha_q$ nor $G\alpha_{11}$ completely. Accordingly, the combination of sh-Gq and sh-G11 could not diminish the DMR signal to a similar extent as the positive control FR (Fig. 2C). This can be explained by residual $G\alpha_q$ and $G\alpha_{11}$ expression and further highlights the inhibitory power of the pharmacological pan-Gq inhibitor FR. These experiments reveal that $G\alpha_q$ and $G\alpha_{11}$ are expressed in mPASMCs and suggest that both contribute to acute 5-HT-dependent constriction to a similar extent.

After the analysis of 5-HT-induced constriction we generated dose–response curves of U-46619 ($10^{-10}$ M–$10^{-5}$ M) after pre-incubation with FR ($10^{-6}$ M), the solvent DMSO or an equal concentration of the competitive thromboxane-receptor antagonist SQ 29,584 (SQ, $10^{-6}$ M) in isometric force measurements. SQ as well as FR shifted U-46619 dose–response curves to the right (Figs. EV2A and 3A). In accordance with our experiments above (see Fig. 1) pre-incubation of PAs with a combination of Y-27632 and FR completely abolished U-46619-dependent vasoconstriction (Figs. EV2A and 3A,B). Interestingly, FR alone reduced the maximum constriction by U46619 to ~22%. Thus, FR is more efficient than SQ at higher U-46619 concentrations. Finally, we also recorded ET-1 dose–response curves ($10^{-12}$ M–$10^{-7}$ M) after pre-incubation of PAs with FR ($10^{-6}$ M), the solvent DMSO or an equal concentration of the competitive dual ET$_{A/B}$ receptor antagonist Bosentan (Bos). Bos and

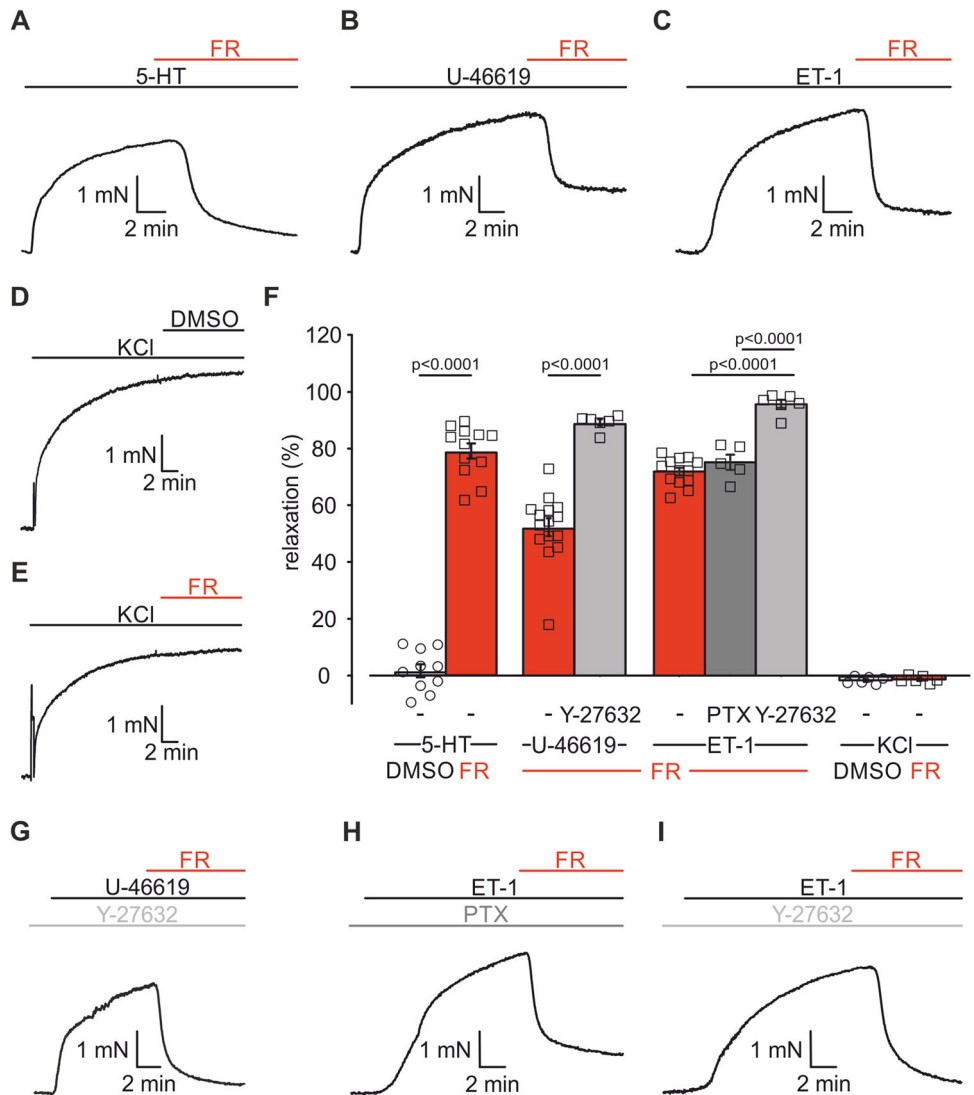

**Figure 1. FR induces pulmonary vasorelaxation after pre-constriction with several Gq-dependent agonists.**

(A–E) Original traces of isometric force measurements in mouse PAs displaying pulmonary vasorelaxation by single dose FR ($10^{-6}$ M) or the solvent DMSO after pre-constriction with 5-HT ($5 \times 10^{-7}$ M, A), U-46619 ($10^{-7}$ M, B) and ET-1 ($3 \times 10^{-9}$ M, C) or KCl ($4 \times 10^{-2}$ M, D, E). (F) Statistical analysis of DMSO or FR-dependent pulmonary vasorelaxation after constriction with Gq-dependent agonists (5-HT + DMSO ($n = 10$ PA rings), 5-HT + FR ($n = 12$ PA rings), U-46619 + FR ($n = 15$ PA rings), Y27632 + U-46619 + FR ($n = 6$ PA rings), ET-1 + FR ($n = 13$ PA rings), PTX + ET-1 + FR ($n = 5$ PA rings), Y-27632 + ET-1 + FR ($n = 6$ PA rings)) or KCl (each $n = 6$ PA rings). (G–I) Original traces of isometric force measurements displaying FR-induced relaxation after pre-constriction with Gq-dependent agonists and pre-incubation with Y-27632 ($10^{-5}$ M, G, I) or PTX (1 µg/ml, H). Data information: Values are expressed as mean ± SEM. (F) Unpaired student's t-test, one-way ANOVA, Tukey's post hoc test. Source data are available online for this figure.

FR caused a pronounced right shift of the ET-1 dose–response curve (Figs. EV2B and 3C,D). Similar to our experiments with FR application after ET-1 pre-constriction (see Fig. 1), PTX could not augment the effect of FR whereas the combination of Y-27632 and FR abolished ET-1-induced constriction completely (Figs. EV2B and 3C,D). Again, FR alone reduced the maximum constriction by ET-1 to ~12% indicating that FR is more efficient than Bos at higher ET-1 concentrations. Thus, the pharmacological pan-Gq inhibitor FR can completely abrogate Gq signaling in PAs and it is more efficient than the specific receptor antagonists SQ and Bos. The higher efficiency of FR compared with competitive receptor antagonists can be explained by pseudo-irreversible inhibition with very slow

dissociation kinetics at the Gq protein level downstream of the receptor by FR (Schrage et al, 2015).

As we could show that FR strongly reduces vascular tone increase by various pathophysiolgcally relevant Gq-dependent agonists in mouse PAs we next examined the effect of FR also in pig tissue, because the physiological and pharmacological profile of porcine vessels is more similar to those of humans. Also in this species pre-treatment with FR ($10^{-7}$ M) diminished vasoconstriction evoked by ET-1 dose–response experiments ($10^{-10}$ M–$5 \times 10^{-8}$ M) (Fig. 3E,F) to a similar extent as in mouse (Fig. 3C,D). In addition, we also changed the order of drug application in mouse and pig to assess the dose-dependency of pulmonary vasorelaxation by FR. The data revealed

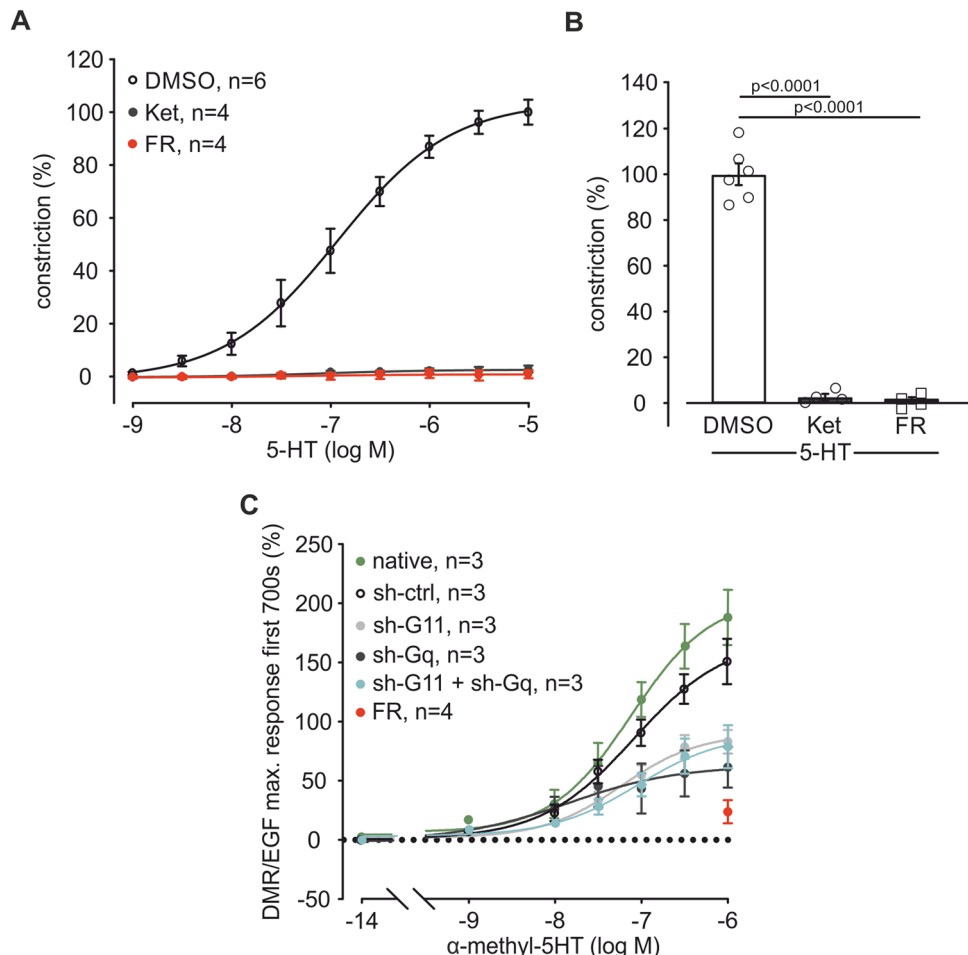

**Figure 2. FR abolishes 5-HT-induced pulmonary vasoconstriction, which is mediated via $G\alpha_{11}$ and $G\alpha_q$.**

(A) Dose–response curve of 5-HT ($10^{-9}$ M–$10^{-5}$ M) after pre-incubation with the solvent DMSO ($n = 6$), the 5-HT$_{2A/2C}$ receptor antagonist Ket ($10^{-6}$ M, $n = 4$) or FR ($10^{-6}$ M, $n = 4$) in isometric force measurements of mouse PAs. (B) Statistical analysis of vasoconstriction at the highest 5-HT concentration applied ($10^{-5}$ M) in the dose–response curve in (A). (C) DMR dose–response curve of α-methyl-5-HT ($10^{-14}$ M–$10^{-6}$ M) in native mPASMC ($n = 3$) and mPASMC transduced with control virus (sh-ctrl, $n = 3$) or virus containing a sh-G11 ($n = 3$), sh-Gq ($n = 3$) or sh-G11 + sh-Gq ($n = 3$) RNA plasmid. As positive control FR ($10^{-6}$ M, $n = 4$) was applied to native cells before the highest α-methyl-5-HT ($10^{-6}$ M) concentration, DMR responses were analyzed in three independent experiments and normalized to Gq-independent signals induced by EGF. Data information: Values are expressed as mean ± SEM. (B) One-way ANOVA, Tukey's post hoc test. Source data are available online for this figure.

that FR induced dose-dependent vasorelaxation after pre-constriction of PAs by the Gq-dependent vasoconstrictors 5-HT ($5 \times 10^{-7}$ M, Fig. EV2C) and U-46619 ($10^{-7}$ M, Fig. EV2D) in mouse and U-46619 ($3 \times 10^{-7}$ M, Fig. EV2E) and phenylephrine (Phe, $3 \times 10^{-5}$ M, Fig. EV2F) in pig. As expected, FR showed no effect on vascular tone after pre-constriction with KCl ($3 \times 10^{-2}$ M, Fig. EV2G) in porcine PAs. Thus, FR prevents and reverses Gq-dependent vasoconstriction by different agonists in PAs of mouse and pig.

## FR has a stronger relaxing effect on PAs than clinically used drugs for PH

Pulmonary vasoconstriction is a key feature of PH pathophysiology and therefore a major pharmacological target for the treatment of PAH patients. We, therefore, compared the effect of FR with that of the clinically applied ET$_{A/B}$-receptor antagonist Bos, the prostacyclin analog Iloprost (Ilo) or the phosphodiesterase type

5 (PDE5) inhibitor Sildenafil (Sil) in isometric force measurements. Because ET-1 is a very powerful vasoconstrictor and has been shown to be of special importance in the pathophysiology of PAH (Chester and Yacoub, 2014) we pre-constricted mouse pulmonary vessels with ET-1 ($3 \times 10^{-9}$ M) and then increasing concentrations of FR or the drugs mentioned above were applied at similar concentrations ($10^{-9}$ M–$10^{-5}$ M). FR (Fig. 4A,C) showed a much stronger relaxing effect compared to Bos (Figs. EV3A and 4C), Ilo (Figs. EV3B and 4C) or Sil (Fig. 4B,C) alone. Interestingly, when an additional dose of FR ($10^{-6}$ M) was applied on top of the highest dose of each PH drug (Figs. 4B,D and EV3A,B) a prominent additional relaxation was achieved indicating a much stronger vasorelaxing effect of FR.

As the current most aggressive vasorelaxing therapy for patients with severe PAH is a triple therapy comprising ET-1 antagonist, PDE5 inhibitor and prostacyclin (Humbert et al, 2022) we also compared equal concentrations of FR ($10^{-6}$ M) with triple therapy

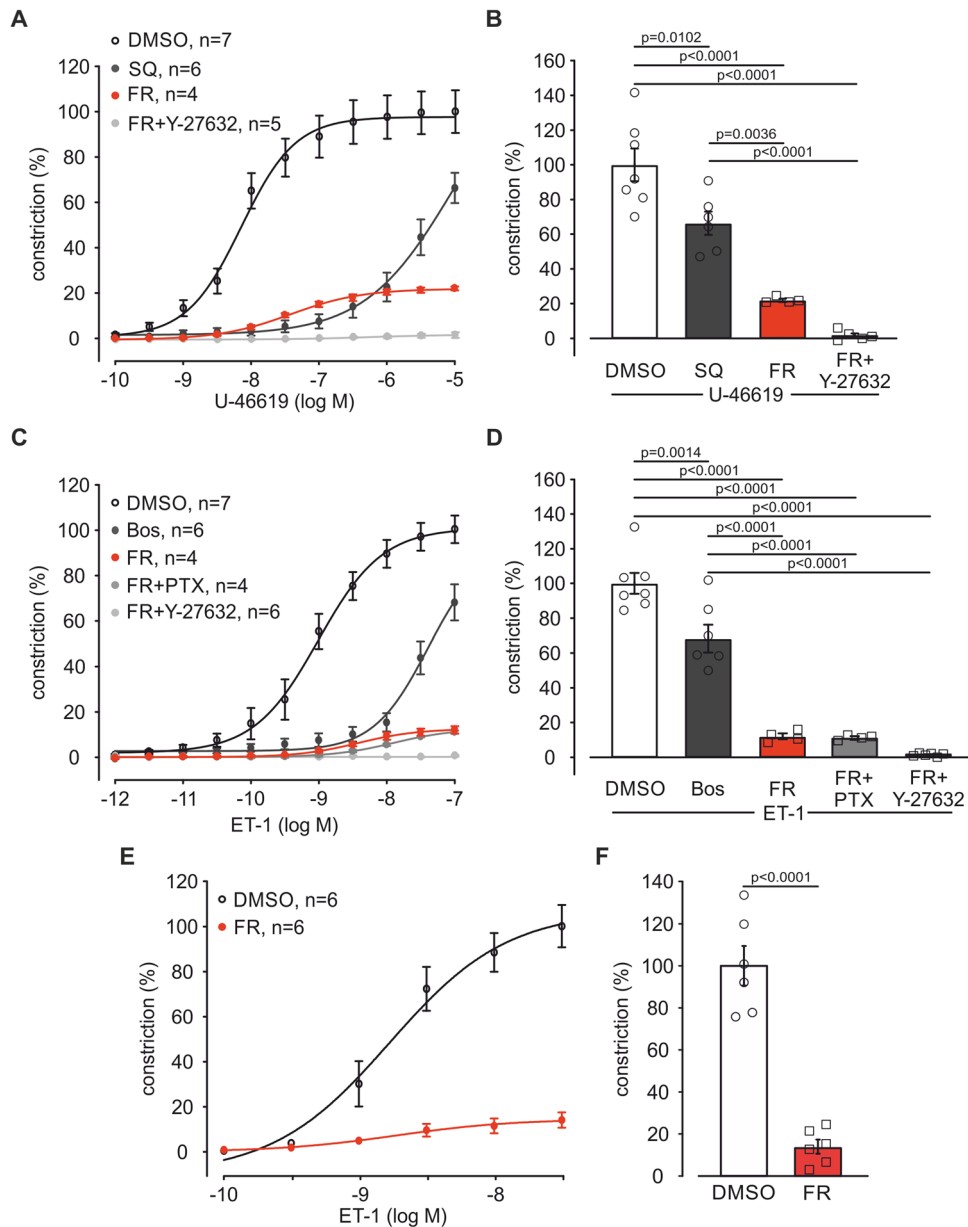

**Figure 3. FR strongly diminishes U-46619- or ET-1-evoked vasoconstriction in PAs of mouse and pig.**

(A) Dose–response curve of U-46619 ($10^{-10}$ M–$10^{-5}$ M) after pre-incubation with the solvent DMSO ($n = 7$), SQ ($10^{-6}$ M, $n = 6$), FR ($10^{-6}$ M, $n = 4$) or FR + Y-27632 ($10^{-5}$ M, $n = 5$) in isometric force measurements of mouse PAs. (B) Statistical analysis of vasoconstriction at the highest U-46619 concentration applied ($10^{-5}$ M) in the dose–response curve in (A). (C) Dose–response curve of ET-1 ($10^{-12}$ M–$10^{-7}$ M) after pre-incubation with the solvent DMSO ($n = 7$), Bos ($10^{-6}$ M, $n = 6$), FR ($10^{-6}$ M, $n = 4$), FR + PTX ($1\,\mu$g/ml, $n = 4$) or FR + Y-27632 ($10^{-5}$ M, $n = 6$) in isometric force measurements of mouse PAs. (D) Statistical analysis of vasoconstriction at the highest ET-1 concentration applied ($10^{-7}$ M) in the dose–response curve in (C). (E) Dose–response curve of ET-1 ($10^{-10}$ M–$5 \times 10^{-7}$ M) after pre-incubation with the solvent DMSO ($n = 6$) or FR ($10^{-7}$ M, $n = 6$) in porcine PAs. (F) Statistical analysis of vasoconstriction at the highest ET-1 concentration applied ($5 \times 10^{-7}$ M) in the dose–response curve in (E). Data information: Values are expressed as mean ± SEM. (B, D) One-way ANOVA, Tukey's post hoc test, (F) Unpaired student's t-test. Source data are available online for this figure.

(Bos + Sil + Ilo in equal parts, $10^{-6}$ M in total, Fig. 4E). We found that a single dose FR induced a similar vasorelaxation (71.8 ± 1.2%, $n = 13$) in murine PAs as the combination of Bos, Ilo, and Sil (Triple, 63.8 ± 4.8%, $n = 5$, $10^{-6}$ M in total, Fig. 4F). Importantly, after application of the triple therapy FR induced an additional relaxation on top ($10^{-6}$ M, 93.4 ± 2.3%, $n = 5$) indicating that direct Gq inhibition can even enhance the effect of triple therapy.

**FR induces strong pulmonary vasorelaxation in small PAs of mice and human subjects ex vivo**

Because of the decisive impact of small intrapulmonary arteries on the total vascular resistance in the lung, we also tested FR in precision-cut lung slices. Intrapulmonary vessels were pre-constricted by 5-HT ($10^{-7}$ M) and then the solvent DMSO or a

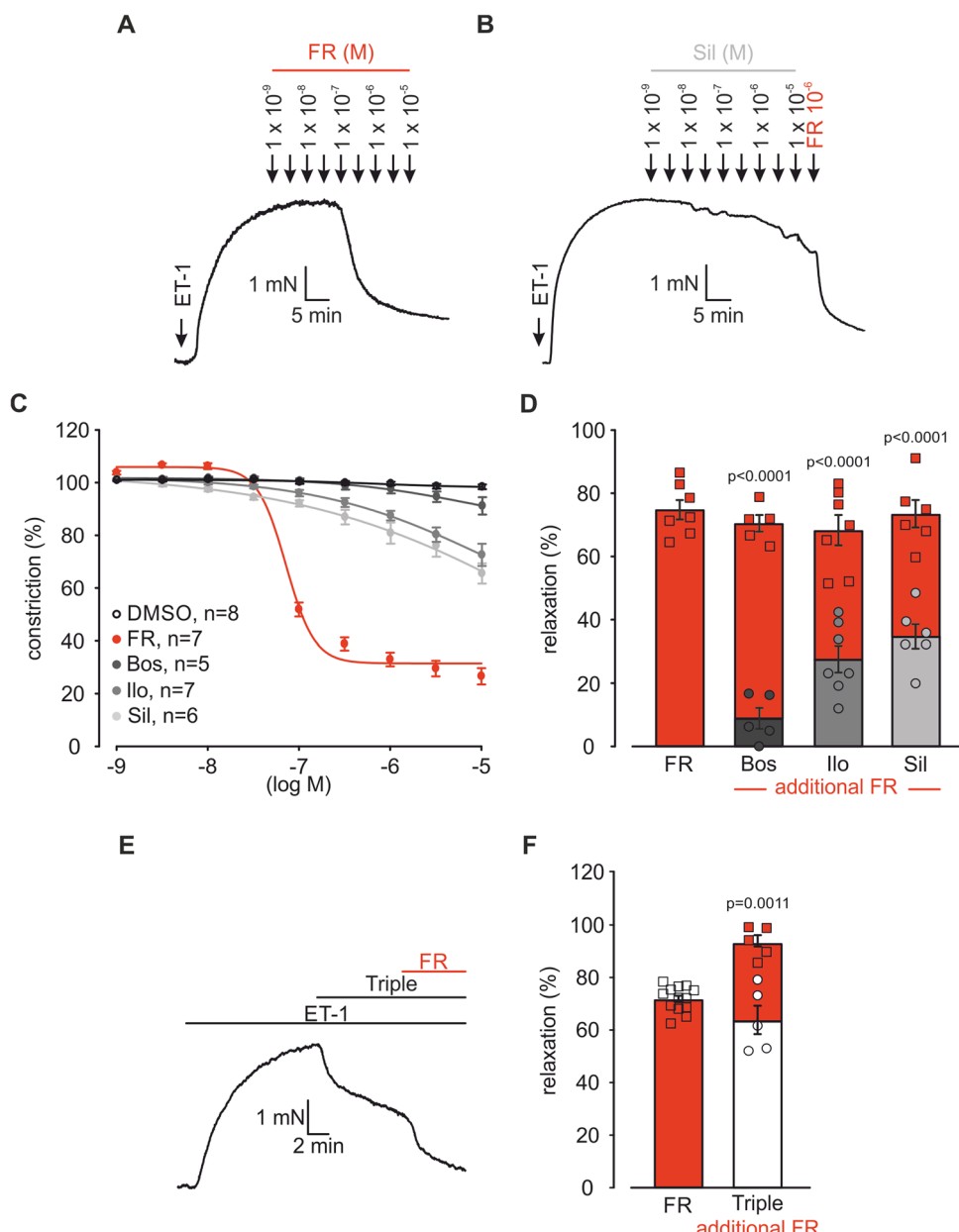

**Figure 4. FR is a stronger pulmonary vasorelaxant than clinically used drugs for PAH.**

(A, B) Original traces of FR (A) or Sil (B) ($10^{-9}$ M–$10^{-5}$ M) dose–response curves after pre-constriction with ET-1 ($3 \times 10^{-9}$ M) in mouse PAs. (C) Dose–response curve of DMSO, FR, Bos, Ilo, or Sil ($10^{-9}$ M–$10^{-5}$ M) after ET-1 ($3 \times 10^{-9}$ M) pre-constriction in mouse PAs. (D) Statistical analysis of the maximal relaxation upon application of the highest dose of FR ($n = 7$ PA rings), Bos ($10^{-5}$ M, $n = 5$ PA rings), Ilo ($10^{-5}$ M, $n = 7$ PA rings) or Sil ($10^{-5}$ M, $n = 6$ PA rings) followed by an additional dose of FR ($10^{-6}$ M). (E) Original trace of Bos, Ilo, and Sil combined application (Triple, $3.3 \times 10^{-7}$ M each, $10^{-6}$ M in total) after pre-constriction with ET-1 ($3 \times 10^{-9}$ M) in mouse PAs. (F) Statistical analysis of the relaxation of FR (same bar as in Fig. 1F) or combined application of Bos, Ilo, and Sil (Triple) followed by an additional dose of FR ($10^{-6}$ M, $n = 5$ PA rings). Data information: Values are expressed as mean ± SEM. (D, F) Paired student's t-test. Source data are available online for this figure.

single dose of FR ($10^{-6}$ M) was applied. FR was found to induce an almost complete vasorelaxation (Fig. 5A–C) while the solvent had no effect. To highlight the translational potential of FR we applied FR in precision-cut lung slices of human subjects after ET-1 ($10^{-7}$ M) pre-constriction (Fig. 5D–F). Also in human tissue FR strongly relaxed PAs (78.3 ± 5.4%, $n = 8$), while the solvent DMSO had no effect (0.0 ± 2.1%, $n = 5$).

Next, we examined FR in the isolated perfused lung (IPL) model of mouse in which, similar to the in vivo situation, PAs of all calibers contribute to pulmonary arterial pressure (PAP). Also in this setting, FR ($10^{-6}$ M) abrogated PAP increase after 5-HT pre-treatment (Fig. 5G,H). We also assessed the effect of FR on hypoxic pulmonary vasoconstriction (HPV) representing a physiological response to optimize ventilation/perfusion matching in

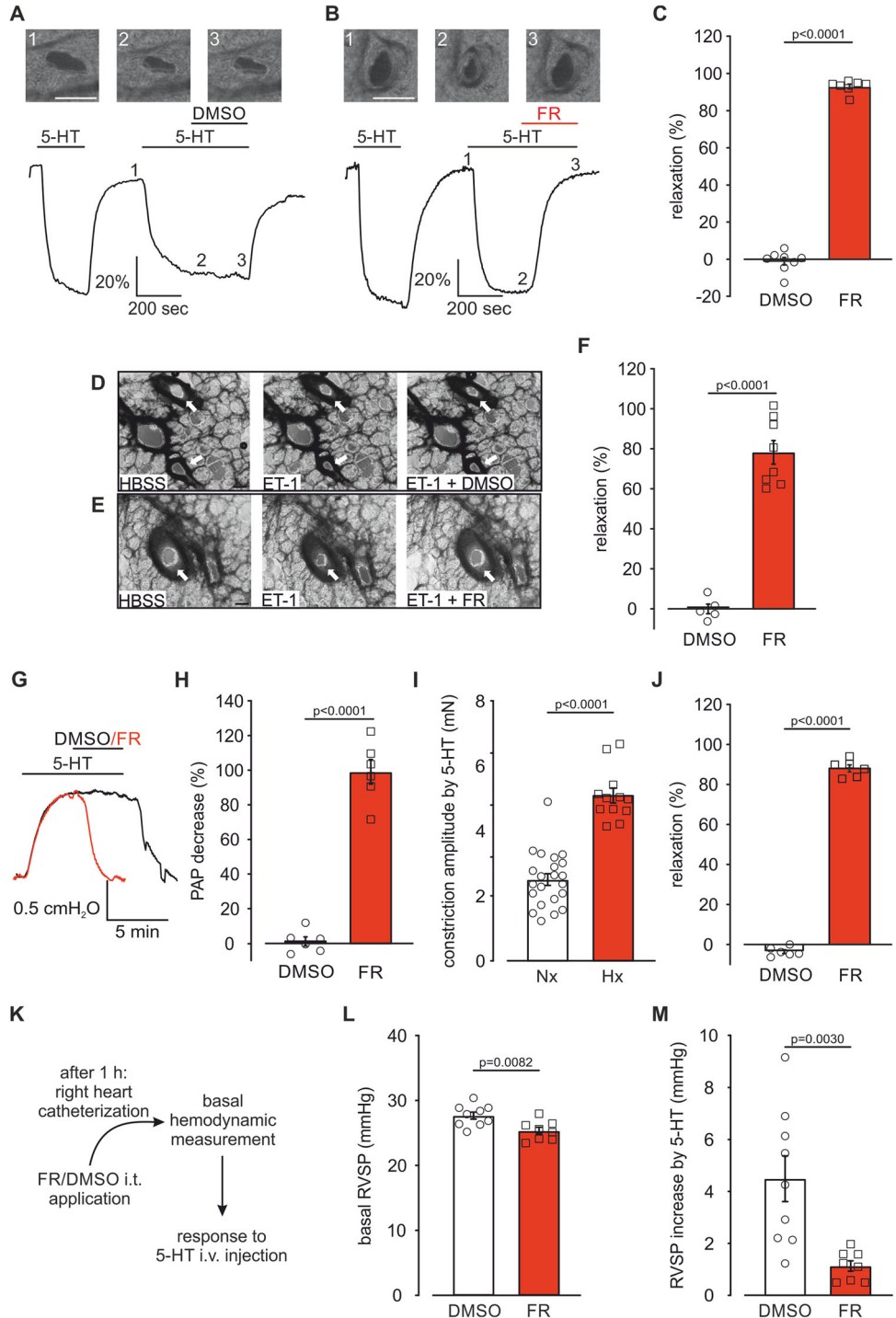

the lung. To examine if Gq proteins contribute to HPV we perfused lungs with the solvent DMSO or FR ($10^{-6}$ M) and then switched ventilation from room air (21% $O_2$) to hypoxic gas (0% $O_2$/100% $N_2$). PAP increased in response to hypoxic gas in lungs perfused either with the solvent DMSO (Fig. EV4A,C) or FR (Fig. EV4B,C), suggesting that HPV is independent of Gq. Taken together, FR acts as a strong vasorelaxant also in small PAs of mice and humans ex vivo, which prompted us to assess its effect on PAs

and pulmonary pressure in mice with Hx-induced PH ex and in vivo.

## FR reduces pulmonary arterial tone and pressure in PH mice ex and in vivo

To test if Gq proteins are also a promising target to relax PAs of mice with PH, we used the Hx-induced PH mouse model (3 weeks

◄

**Figure 5. FR induces pulmonary vasorelaxation in small PAs of mice and humans ex vivo and acutely reduces RVSP in PH mice in vivo.**

(A, B) Phase contrast microscopy pictures of intrapulmonary mouse vessels after solvent DMSO (A) or FR ($10^{-6}$ M) (B) application. Numbers indicate time points shown in the graph (1–3), scale bar: 20 μm. (C) Statistical analysis of small intrapulmonary vessel relaxation evoked by DMSO ($n = 8$ intrapulmonary PAs) or FR ($10^{-6}$ M, $n = 7$ intrapulmonary PAs). (D, E) Phase contrast microscopy pictures of intrapulmonary vessels after solvent DMSO (D) or FR ($10^{-6}$ M, E) application in precision-cut lung slices of human subjects, arrows indicate PAs, scale bar: 200 μm. (F) Statistical analysis of vascular area changes evoked by DMSO ($n = 5$ intrapulmonary PAs) or FR ($10^{-6}$ M, $n = 8$ intrapulmonary PAs). (G) Original traces of PAP in the IPL mouse model during perfusion with 5-HT ($10^{-6}$ M) and DMSO or FR ($10^{-6}$ M). (H) Statistical analysis of DMSO ($n = 6$ mice) or FR ($10^{-6}$ M, $n = 6$ mice)-induced PAP decrease. (I) Statistical analysis of 5-HT ($5 \times 10^{-7}$ M)-induced constriction amplitude in healthy mice housed under Nx (21% $O_2$, $n = 22$) conditions or mice with Hx-induced PH ($n = 12$). (J) Statistical analysis of DMSO ($n = 6$) or FR ($10^{-6}$ M, $n = 6$)-dependent pulmonary vasorelaxation after constriction with 5-HT ($5 \times 10^{-7}$ M) in mice with pre-existing Hx-induced PH. (K) Schematic diagram of the procedure for acute FR application and hemodynamic analysis in mouse in vivo. (L) Statistical analysis of basal RVSP 1 h after DMSO ($n = 9$) or FR (2.5 μg/mouse, $n = 8$) i.t. application in mice with pre-existing Hx-induced PH. (M) Statistical analysis of right ventricular pressure response to acute 5-HT i.v. bolus injection ($5 \times 10^{-3}$ M, 10 μl) in these mice (DMSO: $n = 9$; FR: $n = 8$). Data information: Values are expressed as mean ± SEM. (C, F, H, I, J, L, M) Unpaired student's t-test. Source data are available online for this figure.

of chronic Hx, 10% $O_2$) and performed isometric force measurements with 5-HT ($5 \times 10^{-7}$ M) followed by single-dose FR application ($10^{-6}$ M, see also Fig. 1A). Interestingly, we found that the constriction amplitude in response to 5-HT was strongly increased in mice with pre-existing PH compared to healthy mice (Fig. 5I) reflecting the successful induction of PH in these mice. Importantly, FR relaxed PAs of mice with pre-existing PH by approximately 90% at 8 min (Fig. 5J), this was similar compared to healthy mice (see also Fig. 1F) highlighting the strong vasorelaxing effect of FR even under PH conditions. In addition, after 15 min FR-induced relaxation was nearly complete while DMSO controls were still stable (Hx-DMSO: $-4.7 \pm 1.3$% vs. Hx-FR: $98.0 \pm 0.8$%). To investigate acute Gq inhibition by FR in vivo, we locally applied FR (2.5 μg/mouse) or DMSO to the lung via the intra-tracheal (i.t.) route 1 h before hemodynamic measurements in healthy mice and mice with pre-existing PH (Fig. 5K). Because basal tone is low in the pulmonary vasculature (Wilkins et al, 1996), we did not expect an effect of FR on basal pressure, but rather on the pressure increase induced by intravenous (i.v.) 5-HT bolus injection ($5 \times 10^{-3}$ M, 10 μl) during the catheter measurement.

In healthy normoxic mice (normoxia (Nx): 21% $O_2$) basal RVSP (Fig. EV4D) as well as heart rate (Fig EV4E) were similar 1 h after DMSO or FR treatment, as would be expected. Bolus injection of 5-HT clearly increased RVSP in mice treated with the solvent. This RVSP increase was strongly inhibited in mice that had received FR i.t. 1 h before (Fig. EV4F). Next, we tested the acute hemodynamic effect of FR in mice with chronic Hx-induced PH. In this model basal RVSP was elevated compared to healthy normoxic mice, which reflects the successful induction of PH. In these animals FR i.t. application reduced elevated basal RVSP (Fig. 5L) and slightly attenuated basal heart rate (Fig. EV4G). Similar to the effects in healthy normoxic mice FR efficiently prevented 5-HT-induced RVSP increase also in animals with Hx-induced PH (Fig. 5M). Thus, local FR application to the lung can prevent acute RVSP increase by 5-HT in healthy and PH mice in vivo.

To assess the acute kinetics of the FR effect in PH mice we applied single doses of FR, while recording RVSP and left ventricular systolic pressure (LVSP) simultaneously. We have chosen to apply FR via the i.p. route (10 μg/mouse) in these experiments because i.p. applications of FR have been shown to be effective in a previous study (Annala et al, 2019) and avoid repetitive anesthesia when FR is chronically applied daily. After stable baseline recordings (baseline RVSP: DMSO group: $35.6 \pm 1.3$ mmHg, $n = 4$ vs. FR group: $35.2 \pm 1.6$ mmHg, $n = 5$; Baseline LVSP: DMSO group: $87.1 \pm 3.6$ mmHg, $n = 4$ vs. FR group:

86.8 ± 1.5 mmHg, $n = 5$) a single FR dose was applied and starting from 5 min on a small decrease of pressure could be observed. After 20 min LVSP (Fig. EV4H) as well as RVSP (Fig. EV4I) were reduced by around 13% each, while in controls the blood pressure was stable over time.

The reduction of RVSP by FR in mice with pre-existing Hx-induced PH suggests a protective effect of FR on the development of Hx-induced PH.

## FR prevents PH in mice in vivo, and diminishes mPASMC proliferation and migration in vitro

To test if FR is also effective in the prevention of PH, we applied the SuHx model in mouse that has become a standard model to better mimic PAH than the classic model of Hx alone (Boucherat et al, 2022; Wu et al, 2022; Tudor and Stenmark, 2020). We applied FR via the i.p. route (10 μg/mouse i.p., Monday to Friday) (Annala et al, 2019) and as readout hemodynamic measurements using a Millar catheter were performed. The results demonstrated that in normoxic mice chronic FR application over 3 weeks had no effect on RVSP. In the SuHx model of PH RVSP increased in DMSO-treated mice, reflecting the development of PH. This elevation of RVSP could be almost completely prevented by repetitive FR application (Fig. 6A). Heart rate remained unchanged in response to FR administration in both healthy mice and mice developing PH (Fig. 6B) while LVSP was reduced, as expected (EV5A). Next, we assessed pulmonary vessel wall thickness, another key parameter of PH, in H&E-stained lung sections. We found that FR had no effect on vascular wall thickness under normoxic conditions, but it strongly prevented the increase caused by SuHx in particular in small intrapulmonary arteries, in which media/cross sectional area (CSA) ratios were also reduced (Fig. 6C–F). Finally, we examined cardiomyocyte size in right ventricular heart sections as marker for right heart hypertrophy by using wheat germ agglutinin (WGA) stainings. FR did not alter cardiomyocyte size under normoxic conditions, but strongly diminished cardiomyocyte size in response to SuHx (Fig. 6G–I).

Because FR strongly diminished pulmonary vascular wall thickening in vivo, we then investigated potential effects of FR on mPASMC proliferation and migration in vitro.

Effects of FR on mPASMC cell growth and migration were analyzed in native cells and cells stimulated with a combination of platelet-derived growth factor (PDGF, 40 ng/ml) and 5-HT ($10^{-6}$ M) to mimic PH conditions in cell culture (Hervé et al, 1995; Eddahibi et al, 2006; Schermuly et al, 2005). In native cells FR and the solvent

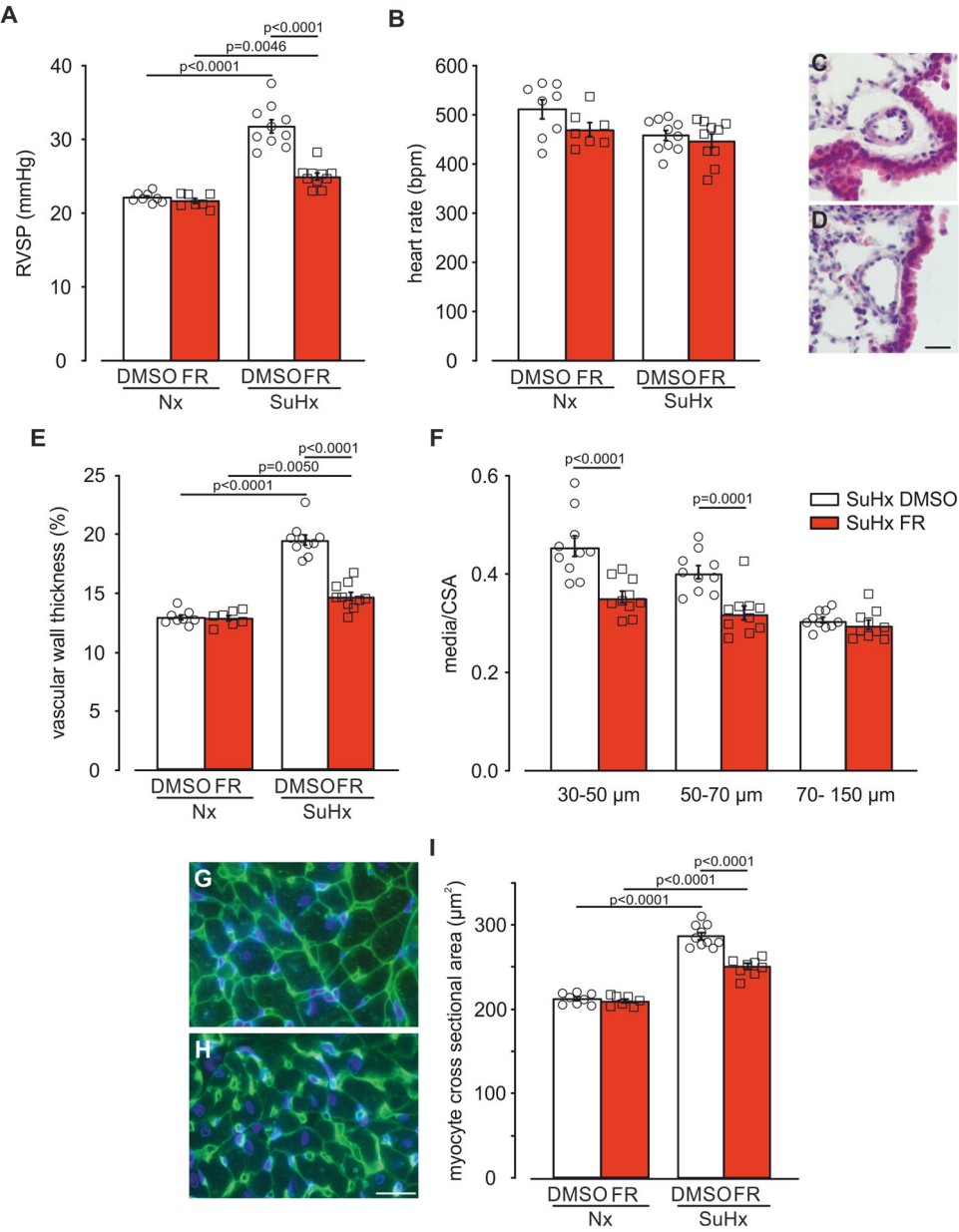

**Figure 6. FR prevents the development of SuHx-induced PH in mouse.**

(A, B) Statistical analysis of RVSP (A) and heart rate (B) in mice treated with the solvent DMSO or FR (10 µg/mouse i.p., Monday to Friday) during exposure to Nx (DMSO: $n = 8$, FR: $n = 7$) or SuHx, (10% $O_2$, DMSO: $n = 10$, FR: $n = 10$) for 3 weeks. (C, D) H&E-staining of lung sections from SuHx-DMSO (C) and SuHx-FR-treated mice (D), scale bar: 20 µm. (E, F) Statistical analysis of vascular wall thickness in mice treated with the solvent DMSO or FR during exposure to Nx (DMSO: $n = 8$, FR: $n = 7$) or SuHx (DMSO: $n = 10$, FR: $n = 10$) (E) and ratio of the media/CSA of small (30–50 µm), medium-sized (50–70 µm) and large (70–150 µm) PAs of mice exposed to SuHx for 3 weeks (DMSO: $n = 10$, FR: $n = 10$) (F). (G, H) WGA staining of transversal heart sections from right ventricles of SuHx-DMSO (G) and SuHx-FR-treated mice (H), scale bar: 20 µm. (I) Statistical analysis of right ventricular cardiomyocyte cross sectional area in mice treated with DMSO or FR during exposure to Nx (DMSO: $n = 8$, FR: $n = 7$) or SuHx (DMSO: $n = 10$, FR: $n = 8$). Data information: Values are expressed as mean ± SEM. (A, B, E, I) One-way ANOVA, Tukey's post hoc test, (F) Two-way ANOVA, Bonferroni post hoc test. Source data are available online for this figure.

DMSO neither affected cell growth (Fig. 7A) nor migration in a wound healing/scratch assay (Fig. 7B). Stimulation of mPASMCs with PDGF and 5-HT strongly increased cell growth and wound healing representing the PH condition. In this setting FR but not the solvent DMSO significantly reduced mPASMC cell growth and wound healing (Fig. 7A,B). To identify the Gq family member that is responsible for this effect we also treated mPASMCs with lentiviral sh-RNAs against

$G\alpha_{11}$, $G\alpha_q$ or both. Interestingly, only knockdown of $G\alpha_q$ reduced cell growth (Fig. 7C) and wound healing (Fig. 7D), while the knockdown of $G\alpha_{11}$ had no effect implicating that $G\alpha_q$ is more relevant than $G\alpha_{11}$ for mPASMC growth and migration under PH conditions. To further analyze if FR-dependent inhibition of PASMC proliferation and migration is mediated via intracellular $Ca^{2+}$ signaling, we first analyzed Orai1, TRPC1, and TRPC3 expression as these $Ca^{2+}$ permeant

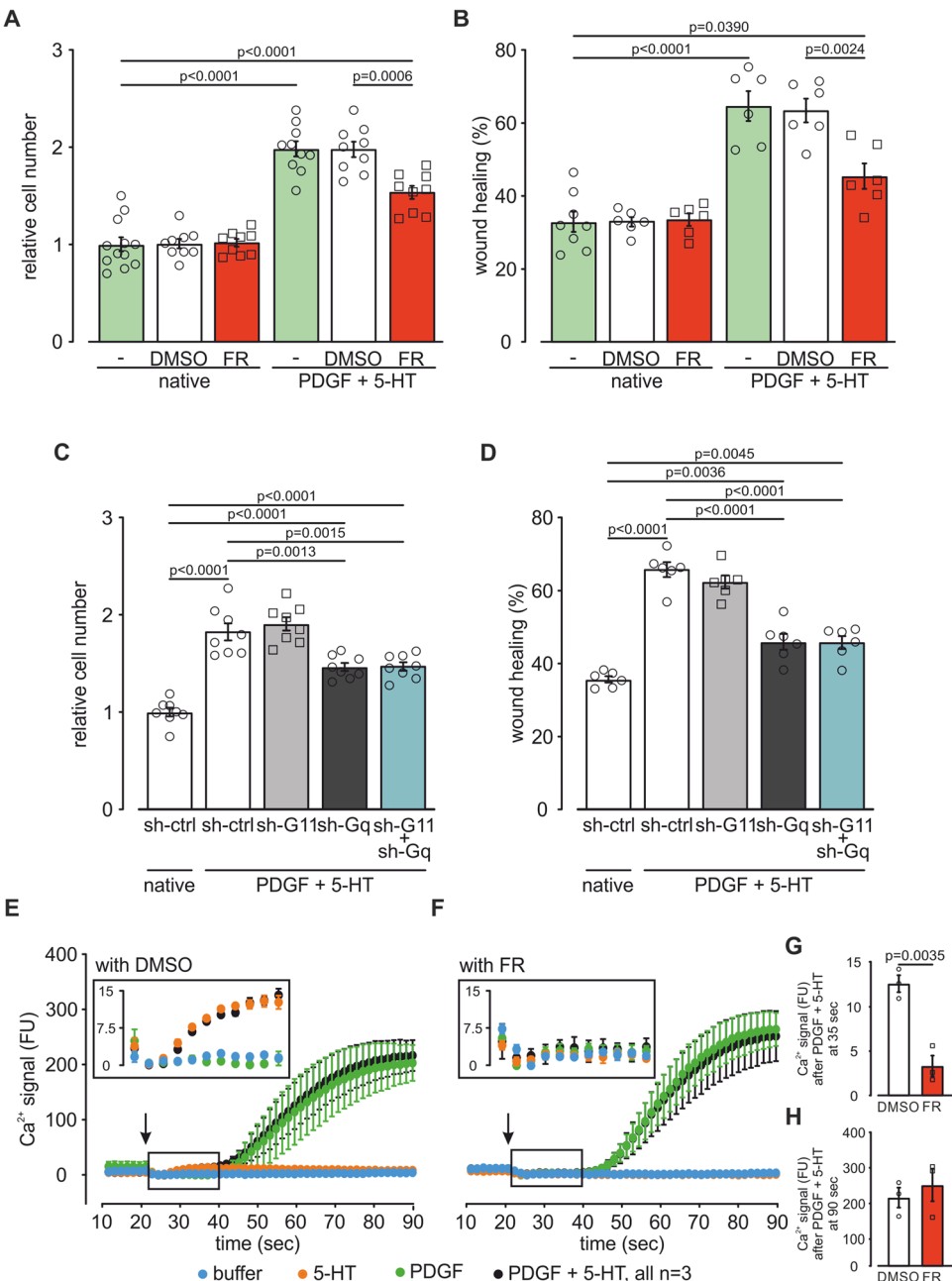

**Figure 7. FR diminishes proliferation and migration in mPASMCs in vitro.**

(A) Statistical analysis of the relative cell number of native mPASMCs ($n = 12$) and native cells treated with DMSO ($n = 9$) or FR ($10^{-6}$ M, $n = 9$) as well as mPASMCs treated with PDGF (40 ng/ml) + 5-HT ($10^{-6}$ M, $n = 10$) and PDGF + 5-HT-treated mPASMCs with DMSO ($n = 9$) or FR ($n = 9$) treatment. (B) Statistical analysis of wound healing of native ($n = 8$) and PDGF (40 ng/ml) + 5-HT ($10^{-6}$ M)-treated mPASMCs with or without DMSO or FR ($10^{-6}$ M) treatment ($n = 6$ each). (C, D) Statistical analysis of the relative cell number ($n = 8$ each) (C) and wound healing ($n = 6$ each) (D) of native and PDGF (40 ng/ml) + 5-HT ($10^{-6}$ M)-treated mPASMCs after lentiviral transduction with sh-ctrl, sh-G11, sh-Gq RNA, or both. (E–H) Acute changes of intracellular $Ca^{2+}$ signal (fluorescence units (FU), corrected for minimum within 5 points after addition) after stimulation with 5-HT ($10^{-6}$ M) or PDGF (40 ng/ml) or both (arrows indicate timepoint of application) without DMSO (E) or with FR ($10^{-6}$ M) (F) pre-treatment and statistical analysis of $Ca^{2+}$ signal of - cells 35 s ($n = 3$) (G) or 90 s ($n = 3$) (H) after PDGF (40 ng/ml) + 5-HT ($10^{-6}$ M) addition, insets show magnified view of the curves within the boxes. Data information: Values are expressed as mean ± SEM. (A–D) One-way ANOVA, Tukey's post hoc test, (G, H) Unpaired student's t-test. Source data are available online for this figure.

channels were shown to be increased in human and/or rat PASMCs in PH (Masson et al, 2022; Masson et al, 2023). Our qPCR analysis revealed that only Orai1 and TRPC1 but not TRPC3 channels were expressed on mRNA level in our mPASMCs. Under PH conditions (PDGF + 5-HT) Orai1 but not TRPC1 expression was increased. Interestingly, FR did neither affect Orai1 nor TRPC1 expression suggesting that FR acts via pathways different from Orai1 and TRPC1/3 expression (Fig. EV5B,C).

Next we also investigated the effect of FR on intracellular $Ca^{2+}$ concentration, because $IP_3$-dependent $Ca^{2+}$ release is known to be involved in the regulation of proliferation and migration of PASMCs (Yan et al, 2019; Landsberg and Yuan, 2004). Therefore, we again stimulated PASMCs with a combination of PDGF (40 ng/ml) and 5-HT ($10^{-6}$ M) or applied them separately and measured the changes of the intracellular $Ca^{2+}$ concentration in the presence of the solvent DMSO (Fig. 7E) or FR ($10^{-6}$ M, Fig. 7F). The $Ca^{2+}$ response to the combined PDGF and 5-HT treatment (PH condition, black line) was found to be biphasic: First an immediate, relatively small $Ca^{2+}$ increase with a maximum at about 35 s (quantified in Fig. 7G), that also occurred after 5-HT treatment alone (orange line, Fig. 7E inset) but not after PDGF treatment alone (green line, Fig. 7E inset), this first $Ca^{2+}$ response could be inhibited by FR (Fig. 7E–G). Then a second, delayed stronger $Ca^{2+}$ increase with a maximum at about 90 s (quantified in Fig. 7H) that also occurred after PDGF but not 5-HT treatment alone, and could not be inhibited by FR (Fig. 7E,F,H). Taken together, FR could only inhibit the small first intracellular Gq-mediated $Ca^{2+}$ increase evoked by 5-HT but not the second larger one induced by PDGF. Nevertheless, FR had a strong inhibitory effect on proliferation and migration of PASMCs (Fig. 7A,C).

Because endothelial cell (EC) injury and apoptosis can induce extensive repair processes that promote PH (Evans et al, 2021) we also analyzed potential detrimental effects of FR on native murine lung ECs (mLECs) in response to FR and performed TUNEL staining. We found that neither DMSO nor FR induced apoptosis in these cells (Fig. EV5D).

### FR reverses SuHx-induced PH in mice in vivo

Efficient therapeutic strategies targeting PH are required to combat established disease, therefore we also tested the potential of FR to reverse pre-existing PH. Mice were exposed to SuHx over 5 weeks, and FR application was started after 3 weeks when PH had already developed. FR was administered for two weeks (10 μg/mouse i.p., Monday to Friday) in parallel to the ongoing PH induction. Catheter analysis revealed that FR could strongly reduce RVSP compared to DMSO controls (Fig. 8A). The heart rate was similar in both groups (Fig. 8B), while LVSP was reduced, as expected (Fig. EV5E). Histological analysis demonstrated that FR strongly diminished vessel wall thickness (Fig. 8C–E), media/CSA ratios of small and intermediate size PAs (Fig. 8F) and the number of fully muscularized PAs (Fig. 8G) when compared to controls; moreover, FR also decreased macrophage infiltration in the lung (Fig. 8H). In addition, FR was able to reduce the Fulton index (Fig. 8I), which indicates the degree of right heart hypertrophy. In accordance with this effect FR also decreased the size of single cardiomyocytes of the right ventricle (Fig. 8J–L). The reduction of right ventricular hypertrophy was confirmed by echocardiography (Fig. 8M–O). These results highlight the critical role of Gq proteins in PH

pathophysiology and further imply the potential of direct Gq inhibition as a therapeutic strategy.

## Discussion

In this study, we show that pharmacological Gq protein inhibition by FR effectively prevents and even reverses Gq-dependent pulmonary vasoconstriction in mice, pigs, and humans. Pulmonary vasorelaxation by FR alone is as efficient as the currently used most aggressive triple therapy. Upon acute local lung application, FR prevented 5-HT-induced RVSP increase in vivo. More importantly, chronic application of FR prevented and even reversed SuHx-induced PH in mice. These findings are important, as in addition to current advances in anti-remodeling approaches, the optimization of vasodilative strategies is thought to improve therapeutic options in PH.

FR is known to be a highly specific and potent Gq inhibitor (Schrage et al, 2015; Pfeil et al, 2020) and our data demonstrate that it inhibits Gq protein signaling completely in PAs of different species. The use of this novel pharmacological agent also enabled us to identify the extent of Gq activation in pulmonary vasoconstriction by different agonists in mice. This could so far be only assessed using genetic mouse models that are known to have potential shortcomings (e.g., compensation, incomplete activation etc.). We demonstrate that 5-HT-induced pulmonary vasoconstriction is mediated by Gq only, while pulmonary vasoconstriction by ET-1 and U-46619, as proposed in earlier studies, are characterized by a Gq-independent component, that can be most likely attributed to $G\alpha_{12/13}$/ROCK (Barman, 2007; McKenzie et al, 2009). In order to discriminate between the contribution of the two Gq family members $G\alpha_q$ and $G\alpha_{11}$ in mPASMCs we used a lentiviral shRNA strategy and revealed that each family member is responsible for about half of the acute 5-HT effect in native mPASMCs similar to previous reports on histamine-stimulated inositol phosphate accumulation in HeLa cells (Krumins and Gilman, 2006) and thrombin-induced $Ca^{2+}$ entry in endothelial cells (Gavard and Gutkind, 2008) while mainly $G\alpha_q$ appears to control cell growth and migration under PH conditions. Because the extent of pulmonary vasorelaxation by currently approved PH drugs varies across species and is also dependent on the type of pre-constrictor used (Benyahia et al, 2015; Li et al, 2012; Jain et al, 2014), we directly compared the effect of the pan-Gq inhibitor FR with Bos, Ilo and Sil and their combination (triple therapy). For pre-constriction ET-1 was used, as this agonist is known to be a central mediator in the pathophysiology of PH (Bressollette et al, 2001; Giaid et al, 1993). Interestingly, the competitive $ET_A$/$ET_B$ antagonist Bos was much less effective in reversing than preventing ET-1 constriction. This may be due to the quasi-irreversible binding of ET-1 at the second of two $ET_A$ receptor binding sites (de Mey et al, 2009; Meens et al, 2010). Similarly, the stronger effect of the pan-Gq inhibitor FR than that of each clinically applied drug as well as the additional vasorelaxing response by FR on top of triple therapy is most likely caused by the pharmacological mode of action of pseudo-irreversible binding due to the particular long residence time and slow dissociation rate of the inhibitor (Schrage et al, 2015; Kuschak et al, 2020; Schlegel et al, 2021; Voss et al, 2021). These findings strongly favor the concept of

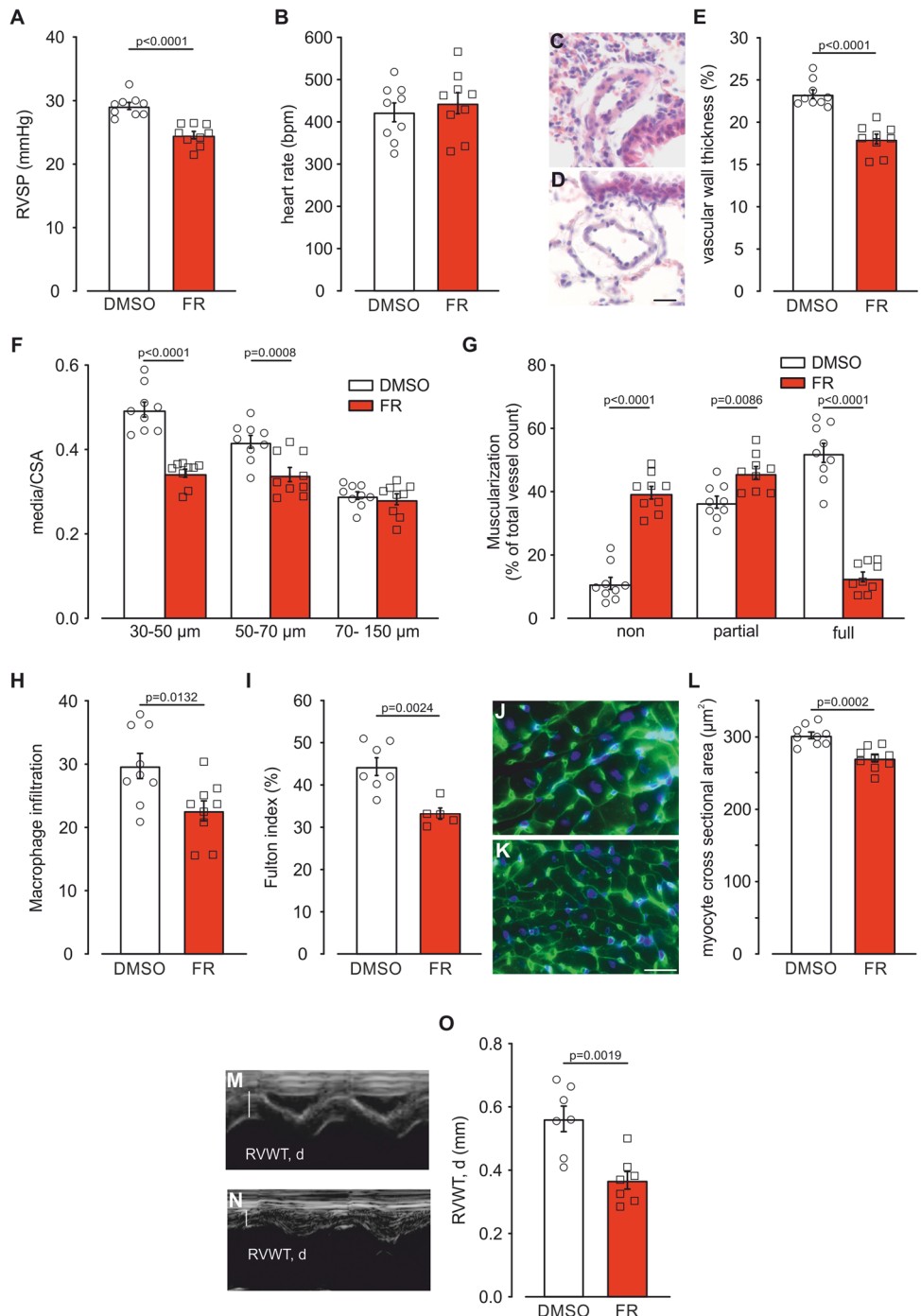

**Figure 8.  FR reverses SuHx-induced PH in mouse.**

(**A, B**) Statistical analysis of RVSP (**A**) and heart rate (**B**) in mice treated with the solvent DMSO ($n = 9$) or FR (10 µg/mouse i.p., Monday to Friday, $n = 9$) in the last 2 weeks of 5 weeks SuHx (10% $O_2$) exposure. (**C, D**) H&E-staining of lung sections from SuHx-DMSO (**C**) and SuHx-FR-treated mice (**D**), scale bar: 20 µm. (**E–G**) Statistical analysis of vascular wall thickness in mice treated with the solvent DMSO or FR during exposure SuHx (DMSO: $n = 9$, FR: $n = 9$) (**E**), ratio of media/CSA of small (30–50 µm), medium-sized (50–70 µm) and large (70–150 µm) PAs of mice exposed to SuHx (DMSO: $n = 9$, FR: $n = 9$) and muscularization of PAs (30–70 µm) of mice exposed to SuHx (DMSO: $n = 9$, FR: $n = 9$) (**G**). (**H**) Statistical analysis of macrophage infiltration in mice treated with the solvent DMSO or FR during SuHx exposure (DMSO: $n = 9$, FR: $n = 9$). (**I**) Statistical analysis of Fulton index in mice treated with the solvent DMSO or FR during SuHx exposure (DMSO: $n = 7$, FR: $n = 5$). (**J, K**) WGA staining of transversal heart sections from right ventricles of SuHx-DMSO (**J**) and SuHx-FR-treated mice (**K**), scale bar: 20 µm. (**L**) Statistical analysis of right ventricular cardiomyocyte cross sectional area in mice treated with DMSO ($n = 9$) or FR ($n = 9$) during exposure to SuHx. (**M, N**) Representative M-mode pictures of SuHx-DMSO (**M**) and SuHx-FR-treated mice (**N**) displaying diastolic right ventricular wall thickness (RVWT, d) during echocardiography. (**O**) Statistical analysis of RVWT, d in mice treated with DMSO ($n = 7$) or FR ($n = 7$) during exposure to SuHx. Data information: Values are expressed as mean ± SEM. (**A, B, E, H, I, L, O**) Unpaired student's t-test, (**F, G**) Two-way ANOVA, Bonferroni post hoc test. Source data are available online for this figure.

inhibiting a central signaling hub downstream of various GPCRs rather than individual Gq-coupled GPCRs.

We also assessed the effect of FR in small intrapulmonary arteries in precision-cut lung slices of mouse and human tissue and the IPL of mouse. These preparations enable to examine vascular tone of pre-capillary arteries that are known to mainly determine pulmonary vascular resistance in health and disease (Abdel Kafi et al, 1998). In these experiments, FR induced nearly complete relaxation of PAs or strong pulmonary arterial pressure decrease underscoring the efficacy of FR. Given the reported involvement of Gq downstream molecules in HPV (Sommer et al, 2008), we tested the potential contribution of Gq itself on HPV by applying FR in the IPL system. Because FR showed no effect on the HPV response we conclude that the so far identified pathways involved in HPV (L-type calcium channels, Kv channels or nonselective cation channels (Sommer et al, 2008)) act in a Gq-independent fashion.

FR application in animals with PH but not in healthy animals resulted in a small transient drop of heart rate that was detected after 1 h but it was absent after 72 h. This could be explained by inhibition of the accelerating function of Gq proteins for G protein-coupled inwardly rectifying potassium channel deactivation kinetics resulting in an increased heart rate (Mark et al, 2000) or the inhibition of Gq-dependent $IP_3$-mediated $Ca^{2+}$ release that has also been shown to increase pacemaker activity in the heart (Ju et al, 2012). In addition, it has been reported that Gq inhibition alters arterial baroreflex control (Meleka et al, 2019). The fact that this short response on heart rate is only found in mice with PH further indicates that Gq signaling is activated in this pathology, again proposing Gq as a promising target in PH.

Even though Gq proteins are expressed in many tissues and organs neither local i.t. application nor systemic i.p. application of FR in this or a previous study (Matthey et al, 2017) resulted in obvious adverse effects but it was effective in the lung. This can be explained by similar pulmonary FR enrichment after both types of application, which could be demonstrated by LC-MS experiments in an earlier study (Schlegel et al, 2021) and which is most likely due to high Gq expression in the lung and pseudo-irreversible binding of FR (Schrage et al, 2015; Kuschak et al, 2020; Schlegel et al, 2021; Voss et al, 2021). As could be expected from literature on the effects of FR on systemic arteries (Meleka et al, 2019; Crüsemann et al, 2018) acute and chronic FR i.p. application in our experiments also resulted in a reduction of LVSP under normoxic conditions and when applied in our disease models. This was expected as a reduction of the systemic blood pressure is known for nearly all drug classes that are currently used in the clinics to treat PAH (Sildenafil (Preston et al, 2005), Iloprost (Kingman et al, 2017), Riociguat (Ghofrani et al, 2013; Dumitrascu et al, 2006), Bosentan (Williamson et al, 2000)). Except effects on blood pressure pan-Gq inhibition by FR could potentially affect platelet aggregation in vivo. Even though we could not detect spontaneous bleeding in response to the FR concentration used in our in vivo experiments, reports on the structural similar Gq inhibitor YM-254890 (Uemura et al, 2006) as well as on $G\alpha_q$ deficient mice (Offermanns et al, 1997) provide evidence that Gq inhibition can result in impaired platelet aggregation. While anticoagulation is no longer a standard treatment in PAH (Roldan et al, 2016; Jose et al, 2019), inhibition of hemostasis may still be beneficial for some patients (Humbert et al, 2022).

Importantly, when we chronically applied FR during development of SuHx-induced PH we found a reduction of RVSP, smooth muscle thickening and right ventricular hypertrophy. This indicates that FR can prevent adverse remodeling processes in PH, which was confirmed by the reduction of proliferation and migration in mPASMCs by FR in vitro. This may be at least in part mediated by reduction of $Ca^{2+}$ release via $IP_3$ receptors (Yan et al, 2019; Landsberg and Yuan, 2004). Since therapy of PH typically starts after the onset of disease (Brown et al, 2011) the additional potential of FR to reverse PH makes it a promising compound for future therapeutic approaches. A limitation of the study is the focus on the SuHx-induced PH mouse model that should be complemented with more severe rat PH models. Furthermore, the concomitant decrease of the systemic blood pressure by FR treatment should be more thoroughly investigated in future studies. Except their important role for blood pressure regulation, Gq proteins are also critical targets for the treatment of obstructive lung diseases (Matthey et al, 2017) which at an advanced stage can cause PH. FR may be well-suited to tackle both pathologies but this will have to be examined in a future study.

In conclusion, our data demonstrate that Gq signaling represents a central therapeutic target for PH. Besides the ongoing search for anti-remodeling agents targeting growth factor signaling in the vascular wall, our data reveals the extraordinary vasorelaxing property combined with growth-inhibitory effects of direct Gq inhibition by FR. This approach could be very helpful to further optimize therapeutic concepts in PH.

## Methods

### Cell isolation, cell growth, and wound healing/scratch assay

MPASMCs were isolated from female CD1 mice. Therefore, isolated PAs with endothelium removed were cut into 2 mm sections and placed inside-down in small droplets of M231 Medium supplemented with smooth muscle growth supplement (SMGS) followed by liquid evaporation. After successful attachment of the tissue on the 6-well plate 2 ml of fresh medium was added and changed every 2–3 days until the mPASMCs reached 80% confluency and could be passaged. For cell growth assays 60,000 mPASMCs/6-well were seeded. On the next day cells were starved for 24 h in DMEM without FCS. Afterwards the solvent or FR ($10^{-6}$ M, 1 h pre-incubation) was applied in DMEM + 0.1% FCS with or without stimulation with a combination of PDGF (40 ng/ml) and 5-HT ($10^{-6}$ M) for 2 days with change of medium/growth factors and solvent/FR after one day. Cell number was determined by counting in a hemocytometer. To analyze migration of mPASMCs a wound healing/scratch assay was performed. Therefore, 250,000 cells/6-well were seeded in the morning. After 6–8 h cells were starved overnight. 1 h before the scratch cells were pre-incubated with the solvent DMSO or FR ($10^{-6}$ M). The scratch was created in the cell monolayer using a pipet tip and medium was changed to DMEM + 0.1% FCS with or without stimulation with PDGF (40 ng/ml) and 5-HT ($10^{-6}$ M) after the scratch. Pictures of wound healing were taken directly after the scratch and 12 h later. To isolate mouse lung endothelial cells (mLEC) magnet-associated cell sorting (MACS) technology was

used. Therefore, the lung dissociation Kit (Miltenyi) in combination with LS Columns (Miltenyi) and CD31 (PECAM) microbeads (Miltenyi) was applied according to the manufacturer's instruction. After magnetic separation 20,000 cells/24-well were seeded on glass cover slips coated with 0.1% gelatin in Endothelial cell proliferation medium with supplement (Provitro). After the cells were attached, they were treated with solvent or FR ($10^{-6}$ M) for 2 days with medium change after 24 h.

## Reverse transcription PCR (RT-PCR)

RT-PCR experiments were performed as described earlier (Wenzel et al, 2012b; Vosen et al, 2016). Briefly, RNA was extracted using TRIzol®. For cDNA generation the SuperScript VILO Kit (Invitrogen, USA) was used. $G\alpha_q$ (forward): 5′-AGATCGAGCGGCAG CTGCGC-3′, $G\alpha_q$ (reverse): 5′-GTTGTGTAGGCAGATAGGAA GG-3′, $G\alpha_{11}$ (forward): 5′-ACGAGGTGAAGGAGTCGAAGC-3′, $G\alpha_{11}$ (reverse): 5′-CCATCCTGAAGATGATGTTCTCC-3′, $G\alpha14$ (forward): 5′-TCACTGCACTCTCTAGAGACC-3′, $G\alpha14$ (reverse): 5′-GACATCTTGCTTTGGTCCTGTG-3′, GAPDH (forward): 5′-GTGTTCCTACCCCCAATGTG-3′, GAPDH (reverse): 5′-CTTGC TCAGTGTCCTTGCTG-3′.

## Lentiviral transduction

shRNA constructs directed against $G\alpha_q$, $G\alpha_{11}$ and the control shRNA were purchased from Sigma Aldrich and expressed under the U6 promoter (pLKO.1-U6-sh-Control-PGK-Puro sequence: 5′-GCATGCAGAAGTGTAAAGCTA-3′, pLKO.1-U6-sh-Gq-287918-PGK-Puro sequence: 5′-GCTTGTGGAATGATCCTGGAA-3′, pLKO.1-U6-sh-G11-98126-PGK-Puro sequence: 5′-GCACTCA-CACTTGGTCGATTA-3′). HEK293T cells were transfected with vector constructs and packaging plasmids to produce lentiviral particles, which were concentrated by ultracentrifugation (Haas et al, 2009; Jennissen et al, 2012). For lentiviral transduction, 250,000 mPASMC were seeded in 6-well plates. 6–8 h later cells were transduced using 200 ng virus in 800 µl serum-free medium. On the next day, 1.2 ml M231 medium containing SMGS (Thermo Fisher) was added. Afterwards medium was changed every 2–3 days. After 6 days cells were passaged and used the next day for DMR measurement, cell growth assay, wound healing/scratch assay, qPCR or Western blotting.

## Quantitative PCR (qPCR)

cDNA was generated as described above. For expression analysis, QuantiTect Primer Assays (Qiagen) were applied using pre-designed QuantiTect Primers (G11: QT00147686, Gq: QT00133826, Gs: QT00134127, Gi: QT00283997, TRPC1: QT00134988, TRPC3: QT00124194, Orai1: QT00285775). As housekeeper 18s rRNA was used (18s rRNA: QT01036875).

## Western blot

mPASMC were scraped in radioimmunoprecipitation assay buffer containing Protease Inhibitor C (Roche). After centrifugation at $13,000 \times g$ for 5 min the protein concentration of the supernatant was determined by a Bradford Assay (Thermo Fisher Scientific). SDS/PAGE was performed using an 8% polyacrylamide gel

electrophoresis and blotting was performed as previously described (Herz et al, 2012; Wenzel et al, 2009). The antibodies used were $G\alpha_{q/11/14}$ (1:200; sc-365906, Santa Cruz) and GAPDH (1:10,000; 2118, Cell Signaling).

## Label-free DMR measurements

DMR technology is an optical-based label-free detection platform that provides phenotypic measures of cellular activity when cells are exposed to pharmacologically active stimuli (Schröder et al, 2010; Schröder et al, 2011). If used to capture GPCR activation, it reports signaling along all four major G protein pathways (Schröder et al, 2010; Schröder et al, 2011; Camp et al, 2016). DMR measurements were performed according to the previously published protocol (Schröder et al, 2011) with the following modifications: 6 days after lentiviral transduction 7000 cells per well were seeded in M231 medium containing SMGS and cultivated on the DMR plate overnight at 37 °C and 5% $CO_2$. The next day, cells were washed and equilibrated in assay buffer (HBSS supplemented with 20 mM HEPES) at 37 °C for 1 h on the DMR reader (EPIC2 by Corning) to stabilize DMR baseline reads. Then, DMR signals in response to the 5-$HT_{2A/C}$ agonist α-methyl-5-HT ($10^{-14}$ M–$10^{-6}$ M) that is known to signal mainly via $G\alpha_{q/11}$ (McCorvy and Roth, 2015; Maroteaux et al, 2017) were recorded. As positive control FR ($10^{-6}$ M) was applied to native cells only before the highest α-methyl-5-HT ($10^{-6}$ M) concentration. For analysis, the maximum DMR responses to agonist stimulation within the first 700 s normalized to epidermal growth factor (EGF) were compared.

## Intracellular $Ca^{2+}$ measurement

For intracellular $Ca^{2+}$ measurements, mPASMCs were seeded at 25,000 cells per well, in black 96-well cell culture plates with a clear flat bottom, and grown overnight. Cells were pre-incubated with FR or the solvent DMSO for 1 h. The media was replaced with 50 µl Calcium 5 Dye (Molecular Devices, Sunnyvale, CA, USA) per well containing $10^{-6}$ M FR or 0.1% DMSO. After 45 min of incubation at 37 °C, the dye was diluted with 150 µl HBSS supplemented with 20 mM HEPES per well and incubated for another 15 min at 37 °C. The change in intracellular $Ca^{2+}$ concentration was measured as fluorescence units (FU) over time using a Flex Station 3 MultiMode Benchtop reader (Molecular Devices, Sunnyvale, CA, USA). After 20 s baseline read, 50 µl of the compounds were added. The $Ca^{2+}$ signal of every replicate was corrected for minimum within 5 points after addition.

## Isometric force measurements of murine and porcine PAs

Analysis of mouse PAs was performed as described previously (Wenzel et al, 2012a; Welschoff et al, 2014). Briefly, mouse PAs were isolated and cut into 2 mm rings in cold low-calcium PSS, containing 118 mM NaCl, 5 mM KCl, 1.2 mM $MgCl_2$, 1.5 mM $NaH_2PO_4$, 0.16 mM $CaCl_2$, 10 mM glucose and 24 mM Hepes (pH 7.4). The endothelium was preserved. Rings were mounted on a wire-myograph (Multi Myograph 610 M, Danish Myo Technology, Denmark) and pre-stretched to 5 mN in PSS, containing 118 mM NaCl, 5 mM KCl, 1.2 mM $MgCl_2$, 1.5 mM $NaH_2PO_4$, 1.6 mM

CaCl$_2$, 10 mM glucose and 24 mM Hepes (pH 7.4). After a resting time of 20 min vascular function was tested by maximal constriction with phenylephrine ($10^{-5}$ M). After washout the vessels were pre-treated (5 to 7 min, PTX (1 µg/ml) overnight) and stimulated with different compounds in single dose or dose–response experiments. In single-dose experiments, the FR effect was analyzed 8 min after FR application. In isometric force experiments on PAs derived from PH mice vasorelaxation by FR was also quantified after 15 min.

Fresh lungs from adult pigs of either sex (~50 kg) were collected from a local abattoir and transported to the laboratory on ice. A section (~10 cm) of distal PA was then dissected out of the base of one lobe of the lung. Tissues were stored in Krebs-Henseleit buffer at 4 °C overnight. The following day, tissues were cleaned of any adherent parenchyma and connective tissue and ring segments (4 mm in length) suspended in a tissue bath, filled with 5 ml Krebs'-Henseleit solution, maintained at 37 °C and gassed with carbogen. The endothelium was preserved. Tissues were left to equilibrate for ~20 min and then 4 grams tension was added to each ring. Changes in tension in the tissues were recorded using isometric force transducers connected to a Powerlab data acquisition unit and measured using LabChart version 8 (ADInstruments). After 20 min of equilibration, responses to 60 mM KCl were determined twice. Tissues were then exposed to $1 \times 10^{-7}$ M FR for 45 min. Control tissues received 0.1% v/v DMSO as a vehicle control. After the incubation period, a concentration–response curve was carried out by cumulative addition of endothelin-1 ($1 \times 10^{-10}$ M to $3 \times 10^{-8}$ M).

In another set of experiments, tissues were pre-contracted with phenylephrine ($1-3 \times 10^{-5}$ M), 30 mM KCl or U-46619 ($1-3 \times 10^{-7}$ M) and concentration–response curves were carried out by cumulative addition of FR ($1 \times 10^{-8}$ M to $1 \times 10^{-6}$ M) or solvent.

## Precision-cut lung slices of mouse and human subjects

Precision-cut lung slices of mouse were generated as described earlier (Wenzel et al, 2009; Neumann et al, 2018). Mice were sacrificed and lungs were filled with 4% low-melting point agarose (Carl Roth) via the trachea by the use of a Saf-T-Intima catheter (Becton Dickinson GmbH) followed by a small volume of air to flush the agarose into the alveoli. Subsequently, a gelatin solution (6%, type A, porcine skin; Sigma) was perfused through the pulmonary vasculature via the right ventricle. After gelling at 4 °C lungs were removed and 200 µm thick lung slices of the different lobes were cut by a vibratome (VT1200S, Leica). After overnight incubation in serum-free medium at 37 °C, the slices were perfused in a custom-made perfusion chamber. After maximal stimulation with 5-HT (1 µM) and washout a submaximal 5-HT contraction was induced (0.1 µM) followed by FR (1 µM) or DMSO (0.1%). The change of vessel lumen was detected with a CCD camera on an inverted microscope. Changes in lumen area of the small pulmonary vessels were determined by a custom-written software (Lumen Calc 2.4, National Instruments, Texas, USA).

The human donor lungs in this study were purchased from the International Institute for the Advancement of Medicine (IIAM) and de-identified. Since no human subject is involved in the study, ethics (or IRB) approval is waived. Likewise, the WMA Declaration of Helsinki and the Department of Health and Human Services Belmont Report are not applicable to the study. The tissue was sectioned into 250 µm thick slices with a vibratome (VF-300; Precisionary Instruments, Greenville, NC). The slices were then cryopreserved for long-term storage using a published protocol (Bai et al, 2016). Before the experiment frozen human slices were thawed and incubated in DMEM/F-12 solution overnight. Then, slices were stimulated with endothelin-1 (10 nM) for 10 min, followed by endothelin-1 + FR (1 µM) or DMSO (0.1%) for 5 min. The images were captured by a Nikon DS-Ri2 camera on an inverted phase-contrast microscope (Nikon Eclipse TS 100; Nikon, Tokyo, Japan). The area of the arterial lumen was measured using NIH Image J (National Institutes of Health, Bethesda, MD). The vascular constriction and relaxation were expressed by normalizing the vessel area to the baseline value.

## Isolated perfused lung

Measurements were performed as described previously (Wenzel et al, 2013) using a setup of Hugo Sachs Elektronik. Briefly, mice were sacrificed, the trachea was cannulated and the lungs were ventilated with positive pressure (80 breaths/min, 200 µl tidal volume). Then, the PA and the left ventricle were cannulated und the pulmonary vascular system was perfused with PSS (340 mOsmol/l) using a roller pump. Experiments with room air were exerted using negative-pressure ventilation (perfusion velocity 1 ml/min). For acute Hx experiments, FR or solvent were added first, then, mice were ventilated with 100% N$_2$ (positive-pressure ventilation, perfusion velocity 0.8 ml/min) to induce hypoxic pulmonary vasoconstriction (HPV). In all experiments, pulmonary arterial pressure (PAP) was continuously monitored by means of a pressure transducer connected to the cannula in the PA.

## Acute single-dose application and chronic application of drugs in vivo

For acute i.t. application of single doses of FR mice were anaesthetized with isoflurane, intubated, and ventilated by a rodent ventilator (1% isoflurane during application, 100% O$_2$, Minivent, Hugo Sachs Elektronik). Then, mice were carefully removed from the ventilator and solvent (50 µl 0.9% NaCl, 0.5% DMSO) or FR (2.5 µg/mouse) was applied directly into the tube. After that, mice were connected to the ventilator for another 30 s, then, ventilation was stopped, and the animals recovered. 1 h later right ventricular catheter measurements were performed. After baseline recordings the pressure increase induced by i.v. 5-HT bolus injection ($5 \times 10^{-3}$ M, 10 µl) during the catheter measurement was tested. In experiments assessing the kinetics of the FR effect acute i.p. applications of solvent or FR (10 µg FR/mouse, 100 µl in 0.9% NaCl, 1% DMSO) during catheter measurements were applied.

For chronic drug application the solvent or FR was applied via the i.p. route (10 µg FR/mouse, 100 µl in 0.9% NaCl, 1% DMSO).

## Model of Hx- and SuHx-induced PH

For some isometric force measurements and acute FR in vivo application the Hx-induced PH model was used. Therefore, mice were housed in special Plexiglas chambers at 10% O$_2$ (or 21% O$_2$ for controls) for 3 weeks. The pre-set O$_2$ concentration was

controlled by the OxyCycler (BioSpherix). For experiments with chronic FR application the SuHx-induced PH model was applied. Hereby, hypoxic treatment for 3 or 5 weeks was accompanied with weekly Sugen5416 (0.6 mg/mouse in carboxymethylcellulose, 30% DMSO) s.c. application. FR (10 μg) was injected i.p. daily from Monday to Friday during 3 weeks of Hx (prevention study) or during the final 2 weeks of the 5-week protocol (reversal study). Catheter measurements were performed 72 h after the last FR application, then, lungs and hearts were harvested for sectioning or Fulton index analysis.

## Right and left ventricular catheter measurements

Catheter analysis was performed as described previously (Wenzel et al, 2009). For analgesia ketamine (50 mg/kg, i.p.) and xylazine (5 mg/kg, i.p.) or carprofen (5 mg/kg) were applied and isoflurane (1–1.5% during experiment) was used for anesthesia. A Millar pressure catheter (1 F) (Millar, USA) was inserted via the right jugular vein into the right ventricle or via the left carotid artery in the left ventricle. Then, pressure was recorded with the Millar Aria 1 system. In in vivo experiments with acute i.t. single dose drug applications a serotonin i.v. bolus injection (10 μl, $5 \times 10^{-3}$ M) was applied after basal RVSP measurement to test the pressure response to a constrictor.

## Right ventricular echocardiography

Cardiac ultrasound was carried out with a Vevo 3100 Imaging System (FUJIFILM Visualsonics, Inc., Toronto, ON, Canada) using a MX550D Transducer (25–55 MHz, FUJIFILM Visualsonics, Inc., Toronto, ON, Canada). Mice were anaesthetized with isoflurane inhalation (1–2%) and placed in supine position on a heating pad. Electrocardiogram was obtained with integrated electrodes. Respiration rate (RR = 80–120/min) was controlled by adjusting the depth of anesthesia. A standardized workflow was followed as previously described (Müller et al, 2022). Diastolic right ventricular wall thickness (RVWT, d) was measured in a modified parasternal short axis view. The mice were tilted laterally to obtain a cross-sectional view from the right hemithorax. The RV was visualized anterior to the left ventricle and parameters were gathered from M-mode.

## Fulton index

To determine the Fulton index the right ventricle (RV) was dissected from the left ventricle plus septum (LV + S) and dried overnight at 37 °C. On the next day, the weight of the ventricles was measured and the Fulton index was calculated and given as RV/(LV + S) × 100.

## Histology/immunohistochemistry

After hemodynamic measurements lungs and hearts were isolated and fixed as described earlier (Wenzel et al, 2013). After paraffin embedding lungs and hearts were sectioned into 5 μm slices. Lung sections were stained with H&E and wall thickness and vessel diameter of PAs (30–70 μm) were determined. Relative vessel wall thickness was determined as 2 × wall thickness/diameter or media/CSA. To investigate muscularization of PAs primary antibodies

against von Willebrand factor (vWF, 1:100) (Sigma, AB7356) and alpha smooth muscle actin (α-SMAC, 1:800) (Sigma, A5228) were used. For analysis only PAs with a diameter of 30–70 μm were utilized. For analysis of macrophage infiltration a primary antibody against MAC-2 was applied (1:500) (Biozol, CED-CL8942AP) followed by the VECTASTAIN® Elite ABC-HRP Kit, (Peroxidase (Rat IgG) used according to the manufacturer's instruction. For the analysis of right ventricular cardiomyocyte size, heart sections were stained with fluorescein-labeled WGA (1:1000) (Vector Laboratories, FL-1021) together with Hoechst 3342 (1:1000 in PBS) (Sigma). To analyze the amount of collagen deposition the heart sections were stained with picrosirius red. After fixation of the MAC-sorted cells, ECs were identified using CD31 primary antibody (PECAM, 1:800 in PSS) (BD Bioscience, 550274). Primary antibodies were visualized with secondary antibodies conjugated with Cy3 (1:400) (Jackson ImmunoResearch, 711-165-152) and Cy5 (1:400) (Jackson ImmunoResearch, 115-605-206 or 712-605-153). To investigate apoptosis the In Situ Cell Death Detection Kit, Fluorescein (Roche) was used according to manufacturer's instruction.

## Animal experiments

Experiments with mice were performed in female 8–12-week-old CD1 mice (Charles River).

Mouse housing and experiments complied to the guidelines of the German law and were approved by the Landesamt für Natur und Verbraucherschutz (LANUV) NRW, Germany. All mice were housed in groups of maximal 5 animals in standard individually ventilated cages with a 12 h light-dark cycle at 22 ± 2 °C and 55 ± 10% humidity and had ad libitum access to food and water. Fresh lungs from adult pigs of either sex (~50 kg) were collected from a local abattoir, therefore use of the tissue does not require ethical approval.

---

### The paper explained

**Problem**

Sustained pulmonary vasoconstriction and pulmonary remodeling are hallmarks of pulmonary arterial hypertension, a severe disease for which there is currently no cure. Therefore, new target molecules and treatment options are necessary to improve therapy outcome.

**Results**

We show that the specific pan-Gq protein inhibitor FR900359 (FR) induces pronounced pulmonary vasorelaxation in mouse, pig and human tissues. The response is at least as strong as that induced by current most aggressive triple therapy. Our results reveal that FR is also effective in mouse in vivo and can prevent Sugen/hypoxia (SuHx)-induced pulmonary hypertension (PH) as well as pulmonary arterial smooth muscle cell (PASMC) proliferation and migration in vitro. Importantly, FR reverses SuHx-PH in mouse.

**Impact**

In this study, we demonstrate that pharmacological Gq protein inhibition strongly reduces pulmonary arterial tone and PASMC proliferation/migration and may provide an alternative strategy for the treatment of pulmonary hypertension.

## Statistical analysis

Data are indicated as mean ± standard error of the mean (SEM). Animals were randomly assigned into groups. Group sizes were calculated for animal protocol approval using power analysis. All datapoints shown in this paper represent biological replicates. Wire-myograph experiments were excluded if pre-constriction was below 0.5 mN, functional lung slices measurements were excluded if contraction was below 40% to avoid unstable constrictions. The investigator was not blinded, because large differences were found. Statistical differences were determined by one-way ANOVA followed by Tukey's post hoc test or two-way ANOVA followed by Bonferroni's post hoc test. A comparison within one group was performed by paired student's t-test. A comparison of two groups was performed by unpaired student's t-test. $P < 0.05$ was considered significant. GraphPad Prism 8.0 (GraphPad Software, San Diego, USA) was used for analysis.

## For more information

For more information about PH, please visit the Pulmonary Hypertension Association (https://phassociation.org/).

# Data availability

This study includes no data deposited in external repositories. Expanded View for this article is available online.

The source data of this paper are collected in the following database record: biostudies:S-SCDT-10_1038-S44321-024-00096-0.

# Peer review information

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

## Acknowledgements

We thank Astrid Markert (University of Bonn, Germany), Claudia Groll and Anja Vöge (Ruhr University of Bochum) for technical assistance and Stefan Kehraus (University of Bonn, Germany) for FR900359. We also thank Stephan Rosenkranz (University of Cologne, Germany) for providing helpful suggestions for the writing of the manuscript. The project was funded by the Deutsche Forschungsgemeinschaft (DFG, German Research Foundation): FOR2372—DW (WE4461/2-1 and 2, project number: 273251628), BF (FL-276/8-1 and 2), GK, EK, and AP and INST 213/973-1. The project was also funded by the InnovationsFoRUM program of the Ruhr University of Bochum (IF-017-22).

## Author contributions

**Alexander Seidinger**: Formal analysis; Investigation; Visualization; Methodology; Writing—original draft; Project administration. **Richard Roberts**: Formal analysis; Investigation. **Yan Bai**: Formal analysis; Investigation. **Marion Müller**: Formal analysis; Investigation. **Eva Pfeil**: Formal analysis; Investigation. **Michaela Matthey**: Formal analysis; Investigation. **Sarah Rieck**: Formal analysis; Investigation. **Judith Alenfelder**: Formal analysis; Investigation. **Gabriele M König**: Resources. **Alexander Pfeifer**: Resources; Writing—original draft. **Evi Kostenis**: Supervision; Writing—review and editing. **Anna Klinke**: Formal analysis; Supervision. **Bernd K Fleischmann**: Writing—review and editing. **Daniela Wenzel**: Conceptualization; Supervision; Funding acquisition; Writing—original draft; Project administration.

Source data underlying figure panels in this paper may have individual authorship assigned. Where available, figure panel/source data authorship is listed in the following database record: biostudies:S-SCDT-10_1038-S44321-024-00096-0.

## Funding

## Disclosure and competing interests statement

The authors declare no competing interests.

# Expanded View Figures

**Figure EV1. FR prevents 5-HT-induced constriction and G$\alpha_q$/G$\alpha_{11}$ can be downregulated by lentiviral transduction.**

(A) Original traces of 5-HT dose–response curves ($10^{-9}$ M–$10^{-5}$ M) after pre-incubation with DMSO, Ket ($10^{-6}$ M) or FR ($10^{-6}$ M) in PAs. (B) PCR analysis of Gq protein subtypes in native mPASMCs and mPASMCs transduced with lentiviral sh-G11, sh-Gq RNA (negative controls). Murine lung tissue was used as positive control. (C–F) Statistical analysis of relative G$\alpha_{11}$ (C), G$\alpha_q$ (D), G$\alpha_i$ (E), and G$\alpha_s$ (F) mRNA expression in native mPASMCs ($n = 3$) and mPASMCs transduced with lentivirus (sh-control (ctrl), sh-G11, sh-Gq RNA or both, $n = 3$ independent experiments normalized to 18S housekeeping gene. (G, H) Original Western Blot (G) and analysis (H) of G$\alpha_{q/11/14}$ protein expression of native mPASMCs and mPASMCs transduced with lentivirus (sh-control (ctrl), sh-G11, sh-Gq RNA or both, $n = 3$ independent experiments). GAPDH was used as housekeeper. Data information: Values are expressed as mean ± SEM. (C–F, H) One-way ANOVA, Tukey's post hoc test. Source data are available online for this figure.

►

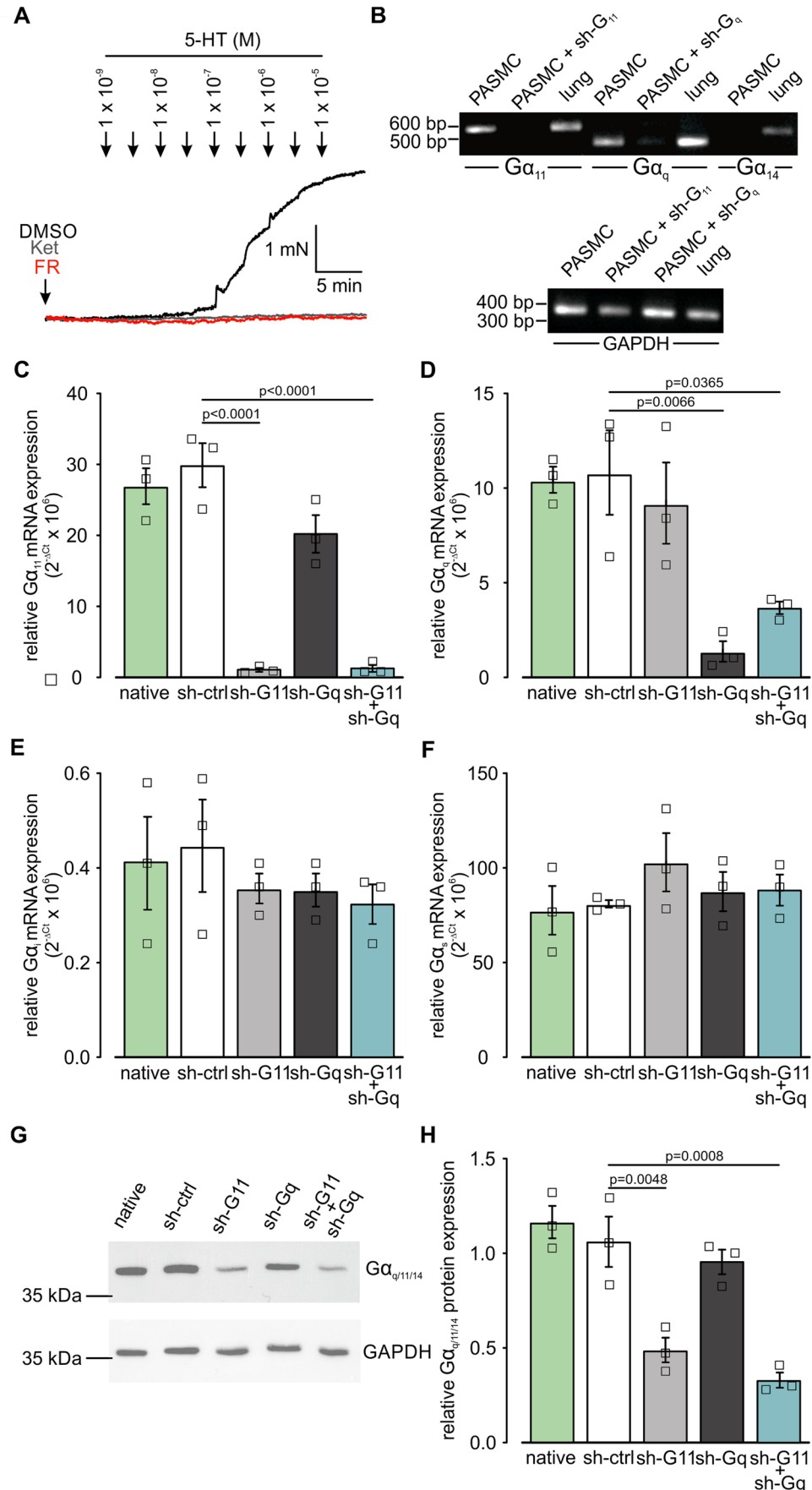

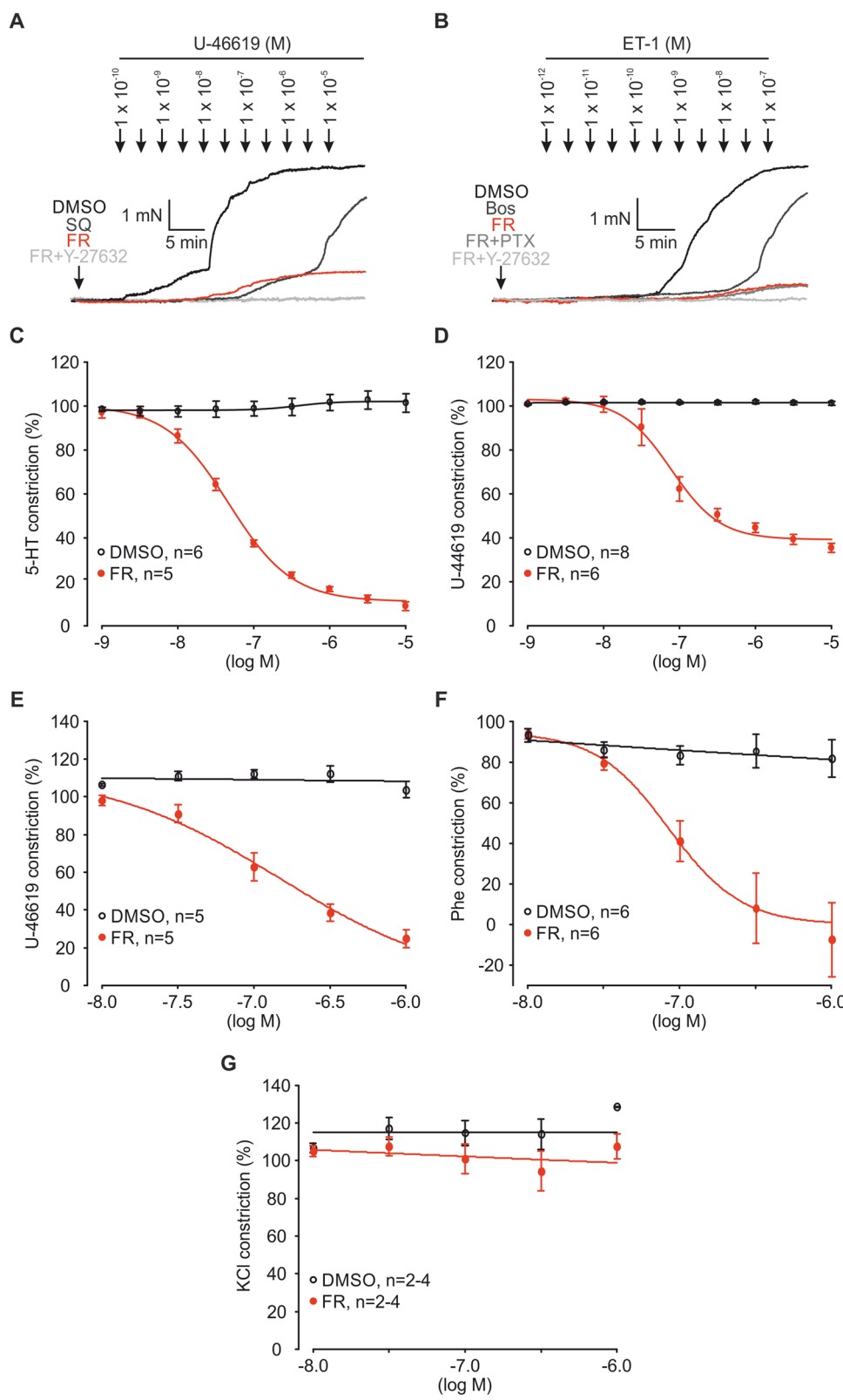

**Figure EV2. FR prevents and reverses Gq-mediated constriction in mouse and pig PAs.**

(A) Original traces of U-46619 dose–response curves ($10^{-10}$ M–$10^{-5}$ M) after pre-incubation with DMSO, SQ ($10^{-6}$ M), FR ($10^{-6}$ M), or FR + Y-27632 ($10^{-5}$ M) in mouse PAs. (B) Original traces of ET-1 dose–response curves ($10^{-12}$ M–$10^{-7}$ M) after pre-incubation with DMSO, Bos ($10^{-6}$ M), FR ($10^{-6}$ M), FR + PTX (1 µg/ml) or FR + Y-27632 ($10^{-5}$ M) in mouse PAs. (C, D) Dose–response curves of DMSO and FR ($10^{-9}$ M–$10^{-5}$ M) after 5-HT ($5 \times 10^{-7}$ M, DMSO: $n = 6$, FR: $n = 5$, C) or U-46619 ($10^{-7}$ M, DMSO: $n = 8$, FR: $n = 6$, D) pre-constriction in murine PAs. (E–G) Dose–response curves of DMSO and FR ($10^{-8}$ M–$10^{-6}$ M) after U-46619 ($3 \times 10^{-7}$ M, DMSO: $n = 5$, FR: $n = 5$, E), Phe ($3 \times 10^{-5}$ M, DMSO: $n = 6$, FR: $n = 6$, F), or KCl ($3 \times 10^{-2}$ M, DMSO: $n = 2$–4, FR: $n = 2$–4, G) pre-constriction in porcine PAs. Data information: Values are expressed as mean ± SEM. Source data are available online for this figure.

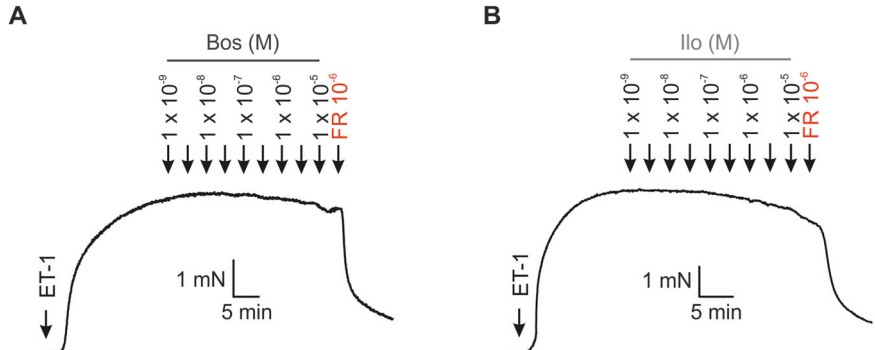

**Figure EV3.   FR strongly relaxes PAs ex vivo.**

(A, B) Original traces of Bos (A) or Ilo (B) dose–response curves ($10^{-9}$ M – $10^{-5}$ M) followed by single dose FR ($10^{-6}$ M) application after pre-constriction with ET-1 ($3 \times 10^{-9}$ M) in mouse PAs.

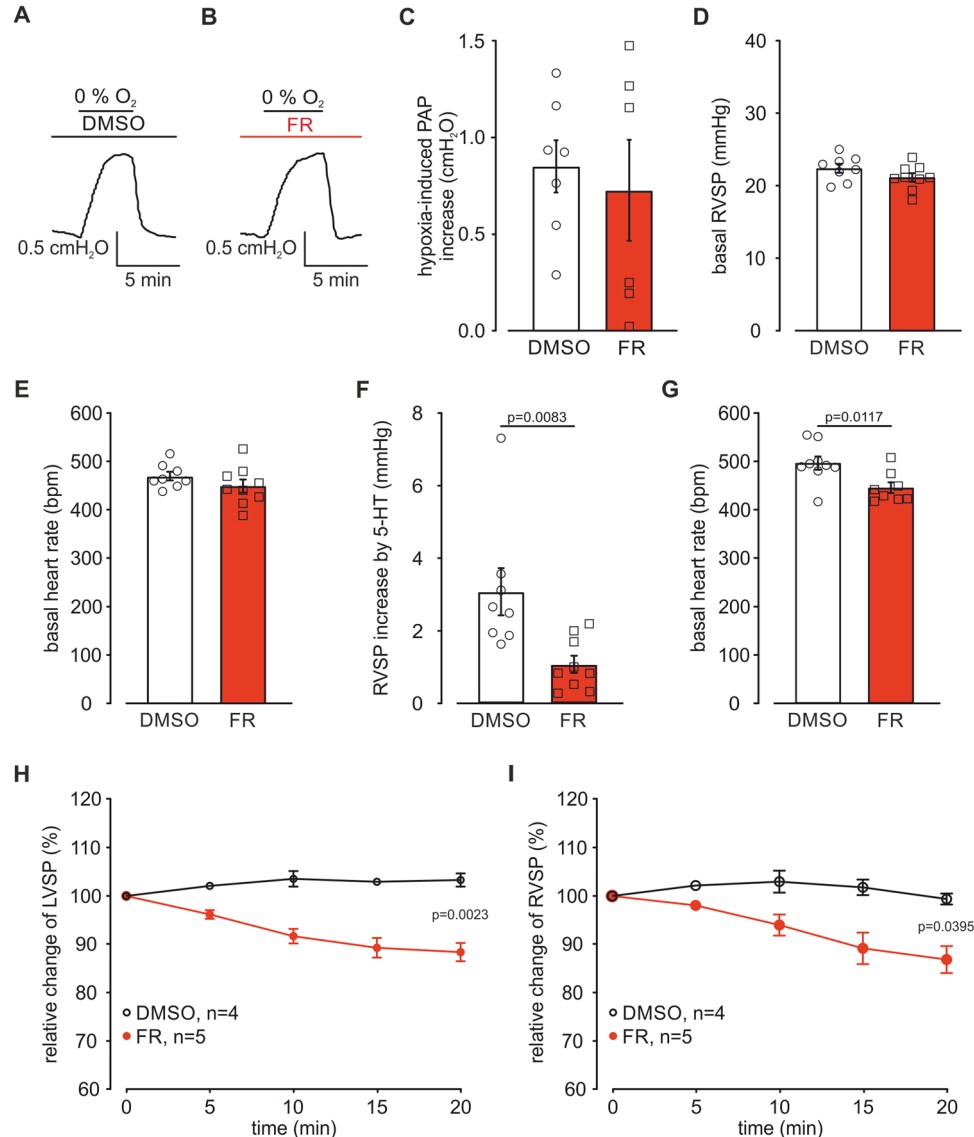

**Figure EV4.  FR does not affect HPV but reduces RVSP in vivo.**

(**A**, **B**) Original traces of PAP in the IPL model during perfusion with DMSO (**A**) or FR ($10^{-6}$ M, **B**) and exposure to hypoxic air (0% $O_2$/100% $N_2$). (**C**) Statistical analysis of PAP increase evoked by hypoxic air during DMSO ($n = 7$ mice) or FR ($n = 6$ mice) perfusion. (**D**) Statistical analysis of basal RVSP 1 h after DMSO ($n = 8$) or FR (2.5 µg/mouse, $n = 8$) i.t. application in healthy mice housed under normoxic (21% $O_2$) conditions. (**E**) Statistical analysis of basal heart rate in these mice (DMSO: $n = 8$; FR: $n = 8$). (**F**) Statistical analysis of RVSP increase in response to 5-HT ($5 \times 10^{-3}$ M, 10 µl) i.v. bolus injection in these mice (DMSO: $n = 8$; FR: $n = 8$). (**G**) Basal heart rate 1 h after DMSO ($n = 9$) or FR (2.5 µg/mouse, 1 h before, $n = 8$) application in mice with pre-existing Hx-induced PH (DMSO: $n = 9$; FR: $n = 8$). (**H**, **I**) Relative change of LVSP (**H**) and RVSP (**I**) after acute DMSO ($n = 4$) or FR (10 µg/mouse i.p., $n = 5$) application in mice with pre-existing Hx-induced PH. Data information: Values are expressed as mean ± SEM. (**C–G**) Unpaired student's t-test. (**H**, **I**) Two-way ANOVA, Bonferroni post hoc test. Source data are available online for this figure.

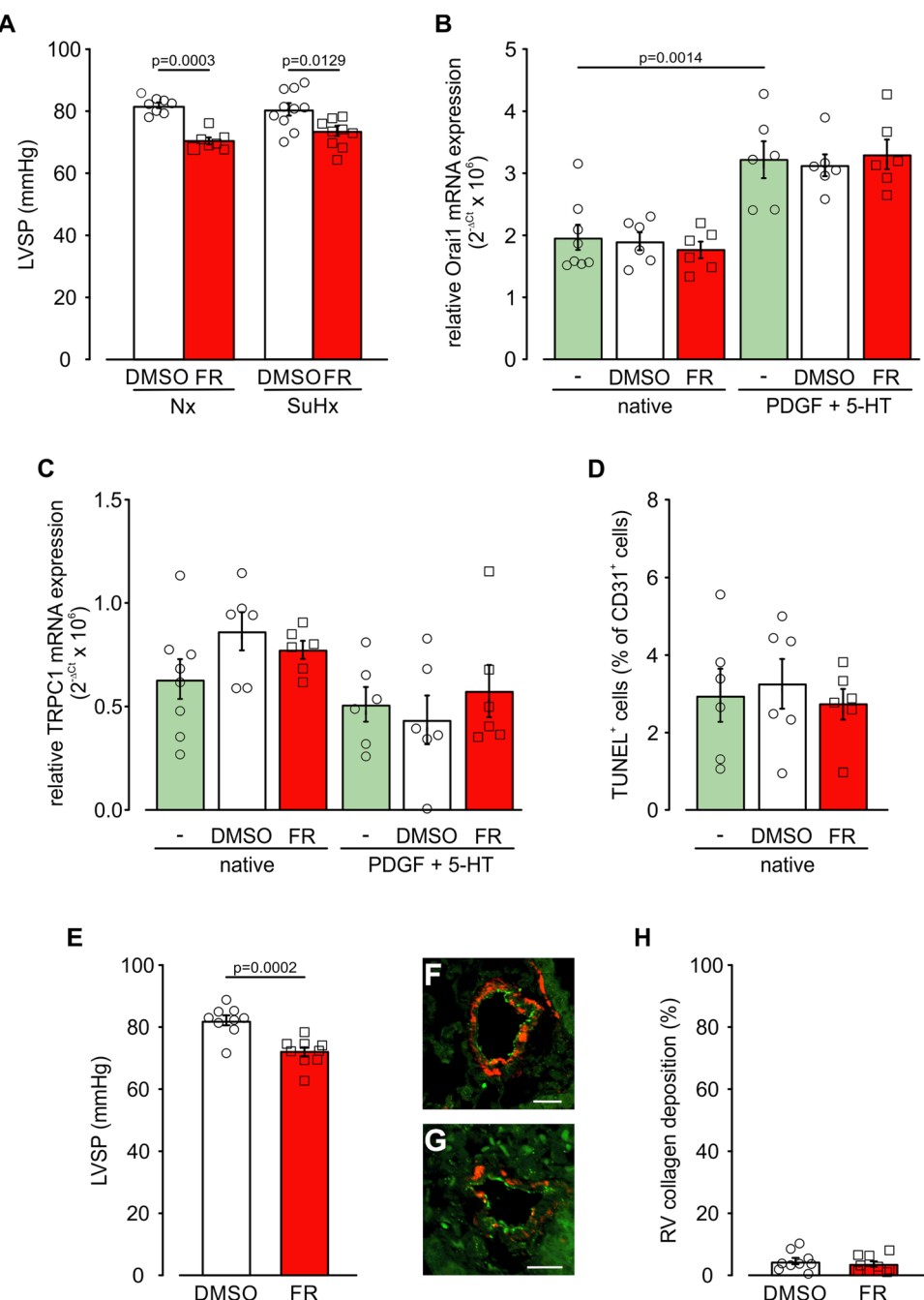

**Figure EV5. FR effects on Hx-induced PH in vivo and mPASMCs as well as mLECs in vitro.**

(A) Statistical analysis of LVSP in mice treated with the solvent DMSO or FR (10 μg/mouse i.p., Monday to Friday) during exposure to Nx (21% $O_2$, DMSO: $n = 8$, FR: $n = 7$) or SuHx (10% $O_2$, DMSO: $n = 10$, FR: $n = 10$) for 3 weeks. (B, C) Statistical analysis of relative Orai1 (B) and TRPC1 (C) mRNA expression in native mPASMCs ($n = 8$) and mPASMCs treated with solvent DMSO or FR ($10^{-6}$ M) with or without additional PDGF (40 ng/ml) + 5-HT ($10^{-6}$ M) stimulation for 12 h, each $n = 6$ normalized to 18 S housekeeping gene, ns indicate different wells derived from at least two different passages. (D) Amount of TUNEL[+] CD31[+] mLECs after 2 days without treatment ($n = 6$) or with DMSO ($n = 6$) or FR ($10^{-6}$ M, $n = 6$) treatment. (E) Statistical analysis of LVSP in mice treated with the solvent DMSO ($n = 9$) or FR (10 μg/mouse i.p., Monday to Friday, $n = 9$) in the last 2 weeks of 5 weeks SuHx exposure. (F, G) vWF/α-SMAC staining of PAs in lung sections from SuHx-DMSO (F) and SuHx-FR-treated mice (G), scale bars: 20 μm. (H) Statistical analysis of collagen deposition in the right ventricle in mice treated with the solvent DMSO ($n = 9$) or FR (10 μg/mouse i.p., Monday to Friday, $n = 9$) in the last 2 weeks of 5 weeks SuHx exposure. Data information: Values are expressed as mean ± SEM. (A–D) One-way ANOVA, Tukey's post hoc test, (E, H) Unpaired student's t-test. Source data are available online for this figure.

