## [Peer Review File · EMBO Molecular Medicine]

Pharmacological Gq inhibition induces strong pulmonary vasorelaxation and reverses pulmonary hypertension

Alexander Seidinger, Richard Roberts, Yan Bai, Marion Müller, Eva Pfeil, Michaela Matthey, Sarah Rieck, Judith Alenfelder, Gabriele König, Alexander Pfeifer, Evi Kostenis, Anna Klinke, Bernd Fleischmann, and Daniela Wenzel

Corresponding author(s): Daniela Wenzel (Daniela.wenzel@ruhr-uni-bochum.de)

Review Timeline:

Submission Date:	23rd Oct 23
Editorial Decision:	21st Nov 23
Revision Received:	26th Apr 24
Editorial Decision:	17th May 24
Revision Received:	28th May 24
Accepted:	5th Jun 24

Editor: Poonam Bheda

Transaction Report:

21st Nov 2023

Dear Prof. Wenzel,

Thank you for the submission of your manuscript to EMBO Molecular Medicine. We have now received feedback from the three reviewers who agreed to evaluate your manuscript. As you will see from the reports below, the referees acknowledge the interest of the study and are overall supporting publication of your work pending appropriate revisions. Please note that editorially and agreed upon in a cross-consultation session with the reviewers, the request from Reviewer 2 to test FR treatment in more severe models of PH, such as in monocrotaline-PH rats or Sugen/Hypoxia-PH rats, will not be considered necessary for a successful revision, though we do agree that this could elevate the translational potential of the manuscript.

Addressing the remaining reviewers' concerns in full will be necessary for further considering the manuscript in our journal, and acceptance of the manuscript will entail a second round of review. EMBO Molecular Medicine encourages a single round of revision only and therefore, acceptance or rejection of the manuscript will depend on the completeness of your responses included in the next, final version of the manuscript. For this reason, and to save you from any frustrations in the end, I would strongly advise against returning an incomplete revision.

We are expecting your revised manuscript within three months, if you anticipate any delay, please contact us.

We require:

4) A .docx formatted letter INCLUDING the reviewers' reports and your detailed point-by-point responses to their comments. As part of the EMBO Press transparent editorial process, the point-by-point response is part of the Review Process File (RPF), which will be published alongside your paper.

5) A complete author checklist, which you can download from our author guidelines (<https://www.embopress.org/page/journal/17574684/authorguide#submissionofrevisions>). Please insert information in the checklist that is also reflected in the manuscript. The completed author checklist will also be part of the RPF.

6) Please note that all corresponding authors are required to supply an ORCID ID for their name upon submission of a revised manuscript.

7) It is mandatory to include a 'Data Availability' section after the Materials and Methods. Before submitting your revision, primary datasets produced in this study need to be deposited in an appropriate public database, and the accession numbers and database listed under 'Data Availability'. Please remember to provide a reviewer password if the datasets are not yet public (see <https://www.embopress.org/page/journal/17574684/authorguide#dataavailability>).

This study includes no data deposited in external repositories.

8) For data quantification: please specify the name of the statistical test used to generate error bars and P values, the number (n) of independent experiments (specify technical or biological replicates) underlying each data point and the test used to calculate p-values in each figure legend. The figure legends should contain a basic description of n, P and the test applied. Graphs must include a description of the bars and the error bars (s.d., s.e.m.). Please provide exact p values.

10) We replaced Supplementary Information with Expanded View (EV) Figures and Tables that are collapsible/expandable online. A maximum of 5 EV Figures can be typeset. EV Figures should be cited as "Figure EV1, Figure EV2" etc... in the text and their respective legends should be included in the main text after the legends of regular figures.

13) Author contributions: CRedit has replaced the traditional author contributions section because it offers a systematic machine readable author contributions format that allows for more effective research assessment. Please remove the Authors Contributions from the manuscript and use the free text boxes beneath each contributing author's name in our system to add specific details on the author's contribution. More information is available in our guide to authors.

Please also suggest a striking image or visual abstract to illustrate your article as a PNG file 550 px wide x 300-600 px high. Share synopsis text and image, as well as eTOC:

Please note that these would be the final versions and changes during proofing are usually not allowed

16) As part of the EMBO Publications transparent editorial process initiative (see our Editorial at <http://embomolmed.embopress.org/content/2/9/329>), EMBO Molecular Medicine will publish online a Review Process File (RPF) to accompany accepted manuscripts.

In the event of acceptance, this file will be published in conjunction with your paper and will include the anonymous referee reports, your point-by-point response and all pertinent correspondence relating to the manuscript. Let us know whether you agree with the publication of the RPF and as here, if you want to remove or not any figures from it prior to publication.

EMBO Molecular Medicine has a "scooping protection" policy, whereby similar findings that are published by others during review or revision are not a criterion for rejection. Should you decide to submit a revised version, I do ask that you get in touch

after three months if you have not completed it, to update us on the status.

I look forward to receiving your revised manuscript.

Yours sincerely,

Poonam Bheda

Poonam Bheda, PhD
Scientific Editor
EMBO Molecular Medicine

**** Reviewer's comments ****

Referee #1 (Comments on Novelty/Model System for Author):

The study entitled "Pharmacological Gq inhibition induces strong pulmonary vasorelaxation and reverses pulmonary hypertension" by Alexander Seidinger et al tested the pulmonary vasodilating effect of the macrocyclic depsipeptide FR900359 (FR), a highly specific inhibitor of Gq/11/14 signalling, and its potential in the treatment of pulmonary hypertension (PH). Pulmonary hypertension (PH), with elevated pulmonary arterial pressure (mPAP {greater than or equal to} 20 mmHg) and pulmonary vascular resistance, affects up to 100 million people worldwide. Patients with PAH (PH Group 1) are presently treated with drugs targeting three dysfunctional endothelial pathways: augmentation of the nitric oxide (NO)/cyclic GMP pathway (soluble guanylyl cyclase activators, phosphodiesterase 5 inhibitors), endothelin-1 (ET-1) receptor antagonists; and prostacyclin analogs. Despite therapeutical progresses, PAH remains a fatal disease with a 3-year survival rate of 58%. Treatment of groups 2 to 5 PH addresses the causal disease and PAH therapeutics are presently not recommended. However, the presence of even moderate PH increases morbidity and mortality. Hence, new approaches to the management of all these patients are urgently needed.

The pathophysiology of PH is multifactorial and includes chronic or recurrent hypoxia and augmented (peri)vascular expression of vasoconstrictors, growth factors and inflammatory mediators, such as endothelin, serotonin, thromboxane A2 and others. Ultimately this leads to endothelial cell dysfunction and increased contraction, proliferation and apoptosis resistance of pulmonary artery smooth muscle cells (PASMCs) with remodeling of small pulmonary arterioles and muscularization of very distal, normally nonmuscularized vessels.

Here the authors combined elegant studies in isolated murine and porcine pulmonary arteries, cultured mouse PASMCs and precision-cut slices from human and murine lungs to study the vasodilatory efficacy and potency of the specific Gq inhibitor FR on serotonin-, thromboxane- and endothelin-1-evoked vasoconstrictions. The results demonstrated that FR markedly prevents and reverses the vasoconstrictor effects of these factors in the pulmonary macro- and microcirculation. These observations were confirmed in isolated perfused murine lungs. The later experiments also demonstrated that FR does not inhibit hypoxia-driven vasoconstrictions, but efficiently counter-regulates vasoconstrictions elicited by agonists acting on Gq-coupled receptors. To elucidate the relevance of this observations for the therapy of PH, the authors firstly tested the effect of local, intratracheal application of FR on acute serotonin-evoked pulmonary vasoconstrictions in anesthetized mice (the subsequent increases in pulmonary arterial pressure, PAP, were indirectly measured through recordings of right ventricular systolic pressure, RVSP). Notably, FR markedly reduced serotonin-evoked increases in RVSP and this effect was preserved in mice with hypoxia-induced PH. Lastly the authors studied the effect of FR in the preclinical model of Sugen/hypoxia (SuHx)-driven chronic PH in mice. Notably FR markedly prevented and even reverted Sugen/HOX-driven PH, pulmonary vascular remodeling as well as right ventricular hypertrophy. Taken together, the results from these complex experiments in vitro/ex vivo and in vivo indicate the potential of pharmacological inhibition of Gq proteins in the treatment of PH. The possible human relevance is emphasized by the results obtained from experiments with "cultured" precision-cut slices from human lungs.

This is a very elegant study with high clinical relevance. The experiments and results are well described and properly discussed. In my view this manuscript will be very interesting for clinicians and scientists working in the fields of lung pathophysiology and vascular physiology and disease.

I have some specific comments and questions:

- In figure 2, the authors showed the relevance of different G proteins in murine pulmonary arteries and a DMR-dose response curve for 5-HT with modified PASMCs treated with sh-RNAs for G11 and Gq. The 5-HT-induced constriction was similarly reduced by downregulation of G11 or Gq, while FR had a higher vasorelaxant effect than the downregulation of these G proteins. Does the downregulation of both G proteins inhibit the vasoconstrictor effect of 5-HT similarly as FR? The expression of these proteins and their downregulation is not shown in the manuscript. Please show the expression of these proteins and their downregulation by qRT-PCR and/or Western Blot.
- The DMR method and its functional meaning should be explained in better way.
- Figure 3 shows that the vasorelaxant effect of FR is superior to other antagonists. Please explain why the Thromboxane antagonist (SQ) or the ETA/B receptor antagonist Bosentan only evoked 50 % vasorelaxation of the vasoconstriction induced by

the Thromboxane analogue or ET-1. Are higher concentrations of SQ or Bosentan more effective?

- In figures 6 and 7 the authors demonstrated that FR prevents and even reverts pulmonary arterial hypertension in the Sugen (Su)Hox mouse model. The long duration of these studies indicates that besides vasorelaxation other mechanisms participate in the protective effects of FR. How does FR impact the proliferation of PSMCs, the survival of endothelial cells and/or pulmonary infiltration by immune cells?

- Lastly: Gq-coupled receptors are also involved in the regulation of peripheral vascular resistance, for instance by Angiotensin II (AT1-R) or adiuretin/vasopressin (V1-R). How does FR affect systemic arterial blood pressure? Which unwarranted effects can be associated with systemic FR therapy? These aspects should be discussed in better way.

Referee #1 (Remarks for Author):

The study entitled "Pharmacological Gq inhibition induces strong pulmonary vasorelaxation and reverses pulmonary hypertension" by Alexander Seidinger et al tested the pulmonary vasodilating effect of the macrocyclic depsipeptide FR900359 (FR), a highly specific inhibitor of Gq/11/14 signalling, and its potential in the treatment of pulmonary hypertension (PH). Pulmonary hypertension (PH), with elevated pulmonary arterial pressure (mPAP {greater than or equal to} 20 mmHg) and pulmonary vascular resistance, affects up to 100 million people worldwide. Patients with PAH (PH Group 1) are presently treated with drugs targeting three dysfunctional endothelial pathways: augmentation of the nitric oxide (NO)/cyclic GMP pathway (soluble guanylyl cyclase activators, phosphodiesterase 5 inhibitors), endothelin-1 (ET-1) receptor antagonists; and prostacyclin analogs. Despite therapeutical progresses, PAH remains a fatal disease with a 3-year survival rate of 58%. Treatment of groups 2 to 5 PH addresses the causal disease and PAH therapeutics are presently not recommended. However, the presence of even moderate PH increases morbidity and mortality. Hence, new approaches to the management of all these patients are urgently needed.

The pathophysiology of PH is multifactorial and includes chronic or recurrent hypoxia and augmented (peri)vascular expression of vasoconstrictors, growth factors and inflammatory mediators, such as endothelin, serotonin, thromboxane A2 and others. Ultimately this leads to endothelial cell dysfunction and increased contraction, proliferation and apoptosis resistance of pulmonary artery smooth muscle cells (PSMCs) with remodeling of small pulmonary arterioles and muscularization of very distal, normally nonmuscularized vessels.

Here the authors combined elegant studies in isolated murine and porcine pulmonary arteries, cultured mouse PSMCs and precision-cut slices from human and murine lungs to study the vasodilatory efficacy and potency of the specific Gq inhibitor FR on serotonin-, thromboxane- and endothelin-1-evoked vasoconstrictions. The results demonstrated that FR markedly prevents and reverses the vasoconstrictor effects of these factors in the pulmonary macro- and microcirculation. These observations were confirmed in isolated perfused murine lungs. The later experiments also demonstrated that FR does not inhibit hypoxia-driven vasoconstrictions, but efficiently counter-regulates vasoconstrictions elicited by agonists acting on Gq-coupled receptors. To elucidate the relevance of this observations for the therapy of PH, the authors firstly tested the effect of local, intratracheal application of FR on acute serotonin-evoked pulmonary vasoconstrictions in anesthetized mice (the subsequent increases in pulmonary arterial pressure, PAP, were indirectly measured through recordings of right ventricular systolic pressure, RVSP). Notably, FR markedly reduced serotonin-evoked increases in RVSP and this effect was preserved in mice with hypoxia-induced PH. Lastly the authors studied the effect of FR in the preclinical model of Sugen/hypoxia (SuHx)-driven chronic PH in mice. Notably FR markedly prevented and even reverted Sugen/HOX-driven PH, pulmonary vascular remodeling as well as right ventricular hypertrophy. Taken together, the results from these complex experiments in vitro/ex vivo and in vivo indicate the potential of pharmacological inhibition of Gq proteins in the treatment of PH. The possible human relevance is emphasized by the results obtained from experiments with "cultured" precision-cut slices from human lungs.

This is a very elegant study with high clinical relevance. The experiments and results are well described and properly discussed. In my view this manuscript will be very interesting for clinicians and scientists working in the fields of lung pathophysiology and vascular physiology and disease.

I have some specific comments and questions:

- In figure 2, the authors showed the relevance of different G proteins in murine pulmonary arteries and a DMR-dose response curve for 5-HT with modified PSMCs treated with sh-RNAs for G11 and Gq. The 5-HT-induced constriction was similarly reduced by downregulation of G11 or Gq, while FR had a higher vasorelaxant effect than the downregulation of these G proteins. Does the downregulation of both G proteins inhibit the vasoconstrictor effect of 5-HT similarly as FR? The expression of these proteins and their downregulation is not shown in the manuscript. Please show the expression of these proteins and their downregulation by qRT-PCR and/or Western Blot.

- The DMR method and its functional meaning should be explained in better way.

- Figure 3 shows that the vasorelaxant effect of FR is superior to other antagonists. Please explain why the Thromboxane antagonist (SQ) or the ETA/B receptor antagonist Bosentan only evoked 50 % vasorelaxation of the vasoconstriction induced by the Thromboxane analogue or ET-1. Are higher concentrations of SQ or Bosentan more effective?

- In figures 6 and 7 the authors demonstrated that FR prevents and even reverts pulmonary arterial hypertension in the Sugen (Su)Hox mouse model. The long duration of these studies indicates that besides vasorelaxation other mechanisms participate in the protective effects of FR. How does FR impact the proliferation of PSMCs, the survival of endothelial cells and/or pulmonary infiltration by immune cells?

- Lastly: Gq-coupled receptors are also involved in the regulation of peripheral vascular resistance, for instance by Angiotensin II (AT1-R) or adiuretin/vasopressin (V1-R). How does FR affect systemic arterial blood pressure? Which unwarranted effects can

be associated with systemic FR therapy? These aspects should be discussed in better way.

Referee #2 (Comments on Novelty/Model System for Author):

For a better translational way, authors should investigate the consequence of curative treatment with FR in the development of PH in more severe experimental models of PH as monocrotaline-PH rats or Sugen/Hypoxia-PH rats.

Referee #2 (Remarks for Author):

In their manuscript "Pharmacological Gq inhibition induces potent pulmonary vasorelaxation and reverses pulmonary hypertension", the authors investigated the potential as a therapeutic target of Gq proteins in PAH. They showed that the inhibition of Gq proteins by FR9000359 induced a potent vasorelaxation ex vivo in PA from healthy mice, pigs, and humans. They also demonstrated that the application of FR in vivo on Sugen-Hypoxia mice reversed PH. They proposed that Gq proteins could be used as a therapeutic target in PH. Their work is interesting. However, this work has methodological issues and requires additional experiments and clarifications.

Major:

1) Authors should use the new hemodynamic definition of PAH from the 2022 guidelines (Humbert et al., ERJ, 2022) and the complete one, including mean pulmonary arterial pressure, pulmonary vascular resistance, and pulmonary arterial wedge pressure. They should also cite more recent papers on the prognosis of PAH.

2) Importantly, for a better translational way, authors should investigate the consequence of curative treatment with FR in the development of PH in more severe experimental models of PH as monocrotaline-PH rats or Sugen/Hypoxia-PH rats.

3) Authors used the shRNA strategy to reduce the expression of Gq and G11 and to measure the consequence of Gq or G11 knockdown on pulmonary arterial contraction.

In addition to pulmonary artery vasoconstriction, PAH is also characterized by strong pulmonary arterial remodeling involving at least exacerbated PAMCS proliferation and migration. In this way, authors should take advantage of their Gq inhibitor and their shRNA to determine the role of Gq and G11 in PASMCM proliferation and migration from control and PH conditions.

4) In line with point #3, as we know, Gq is primordial for intracellular Ca²⁺ signaling. Because Gq activation activates PLC, which leads to diacylglycerol and IP₃ production, and consequently induces an increase of intracellular Ca²⁺ concentration via IP₃R localized at ER membrane as well as store-operated Ca²⁺ channel localized at the plasma membrane including Orai and TRPC channels.

Orai and TRPC channels were recently demonstrated to be upregulated in PAH and to be considered as potential therapeutic targets.

Are there any consequences of Gq inhibition on Orai and TRPC expression and IP₃R Ca²⁺ release? By reducing the function of these Ca²⁺ channels, FR treatments should also reduce PASMCM proliferation, migration since these processes are strongly dependent on intracellular Ca²⁺ concentration.

Authors should investigate and discuss these points in their study.

Moreover, regarding this large spectrum of action of Gq inhibitor, how authors have measured the potential side effects of Gq inhibition in other organs? Authors should investigate the consequences of Gq inhibition in healthy mice or rat.

In addition, authors should not limit their explanations of the increasing heart rate by IP₃R inhibition (page 25).

5) Regarding in vivo experiments in mice (preventive and curative approach), authors should also measure the consequences of FR treatment on right ventricle fibrosis. Authors should add the Fulton index (RV/LV+septum) values for in vivo experiments. Authors should examine the rate of muscularized and non-muscularized pulmonary arteries by using alpha-SMA/VWF staining.

6) Results showing reduction of RVSP in PH mice by FR treatment (1 hour after FR application) are exciting but limited due to the use of the mouse model. Authors should also do these experiments in monocrotaline or Sugen/hypoxia rats to better define the acute consequence of FR infusion in RVSP, cardiac output, and systemic blood pressure. Because of the rapid PA relaxation produced by FR application, we could hypothesize that FR should reduce RVSP in these PH models more rapidly than one hour.

Is there any consequence of Gq inhibition on systemic arterial tone? It should be.

7) In the continuity of point #6, the authors should assess the effect of the Gq-inhibition on the pulmonary vascular tone of the PH condition. Does FR application always induce pulmonary artery relaxation?

Minor:

- 1) Why, in the presence of FR (10⁻⁶ M), there is no constriction of PA-induced by 5-HT dose response (Figure 2A). Because, with FR at 10⁻⁶ M, PA relaxation is around 80% with a residual 20% of contraction.
- 2) The red trace of FR condition is absent from the graph presented in Figure 2C.
- 3) Please add a comma between "mouse" and "pig" in the abstract and between "PH" and "FR" in the discussion (page 25).
- 4) Please homogenize terms as "in vivo" in italics or not.
- 5) Authors should improve the visual quality of Figure 2.

Referee #3 (Comments on Novelty/Model System for Author):

In vitro, ex vivo and in vivo models were all used in this study and the findings seemed all consistent with the overall hypothesis.

Referee #3 (Remarks for Author):

This manuscript reports the pharmacological role of a pan-Gq protein inhibitor FR900359 (FR) in relaxing of pulmonary artery and managing pulmonary hypertension. Specifically, the authors found that the FR compound at low micromolar concentrations induced significant vasodilation in several ex vivo models of mouse, pig, and human pulmonary arteries. In addition, they found that local delivery or i.p. delivery FR decreased pulmonary hypertension in a couple of relevant mouse models. Overall, the study has a translational potential and the experiments appeared to be well-performed. However, the idea of targeting a Gq protein to control vascular tone is not particularly novel, and the quality of this work would be further improved if the following points could be considered or clarified:

1. Page 3, ET-1 and 5-HT should be fully defined when they first appeared.
2. Fig. 1, it would be better if the authors could show sample tracing on the effect of FR on KCL-induced vascular constriction.
3. Fig. EV1, panel B, the RT-PCR study did not have positive and negative controls.
4. Lentiviral transduction decreased Gq and G11 mRNA expression, but it remained unknown how much Gq/11 protein expression was reduced.
5. Why in the pig species, FR showed at least 10-fold more sensitive than in mice?
6. It is not clear for all the vascular contractility study if the endothelium were all removed or not. Please specify this in the method section.

Referee #1 (Comments on Novelty/Model System for Author):

The study entitled "Pharmacological Gq inhibition induces strong pulmonary vasorelaxation and reverses pulmonary hypertension" by Alexander Seidinger et al tested the pulmonary vasodilating effect of the macrocyclic depsipeptide FR900359 (FR), a highly specific inhibitor of Gq/11/14 signalling, and its potential in the treatment of pulmonary hypertension (PH).

Pulmonary hypertension (PH), with elevated pulmonary arterial pressure (mPAP {greater than or equal to} 20 mmHg) and pulmonary vascular resistance, affects up to 100 million people worldwide. Patients with PAH (PH Group 1) are presently treated with drugs targeting three dysfunctional endothelial pathways: augmentation of the nitric oxide (NO)/cyclic GMP pathway (soluble guanylyl cyclase activators, phosphodiesterase 5 inhibitors), endothelin-1 (ET-1) receptor antagonists; and prostacyclin analogs. Despite therapeutical progresses, PAH remains a fatal disease with a 3-year survival rate of 58%. Treatment of groups 2 to 5 PH addresses the causal disease and PAH therapeutics are presently not recommended. However, the presence of even moderate PH increases morbidity and mortality. Hence, new approaches to the management of all these patients are urgently needed. The pathophysiology of PH is multifactorial and includes chronic or recurrent hypoxia and augmented (peri)vascular expression of vasoconstrictors, growth factors and inflammatory mediators, such as endothelin, serotonin, thromboxane A2 and others. Ultimately this leads to endothelial cell dysfunction and increased contraction, proliferation and apoptosis resistance of pulmonary artery smooth muscle cells (PASMCs) with remodeling of small pulmonary arterioles and muscularization of very distal, normally nonmuscularized vessels.

Here the authors combined elegant studies in isolated murine and porcine pulmonary arteries, cultured mouse PASMCs and precision-cut slices from human and murine lungs to study the vasodilatory efficacy and potency of the specific Gq inhibitor FR on serotonin-, thromboxane- and endothelin-1-evoked vasoconstrictions. The results demonstrated that FR markedly prevents and reverses the vasoconstrictor effects of these factors in the pulmonary macro- and microcirculation. These observations were confirmed in isolated perfused murine lungs. The later experiments also demonstrated that FR does not inhibit hypoxia-driven vasoconstrictions, but efficiently counter-regulates vasoconstrictions elicited by agonists acting on Gq-coupled receptors. To elucidate the relevance of this observations for the therapy of PH, the authors firstly tested the effect of local, intratracheal application of FR on acute serotonin-evoked pulmonary vasoconstrictions in anesthetized mice (the subsequent increases in pulmonary arterial pressure, PAP, were indirectly measured through recordings of right ventricular systolic pressure, RVSP). Notably, FR markedly reduced serotonin-evoked increases in RVSP and this effect was preserved in mice with hypoxia-induced PH. Lastly the authors studied the effect of FR in the preclinical model of Sugen/hypoxia (SuHx)-driven chronic PH in mice. Notably FR markedly prevented and even reverted Sugen/HOX-driven PH, pulmonary vascular remodeling as well as right ventricular hypertrophy. Taken together, the results from these complex experiments in vitro/ex vivo and in vivo indicate the potential of pharmacological inhibition of Gq proteins in the treatment of PH. The possible human relevance is emphasized by the results obtained from experiments with "cultured" precision-cut slices from human lungs.

This is a very elegant study with high clinical relevance. The experiments and results are well described and properly discussed. In my view this manuscript will be very interesting for clinicians and scientists working in the fields of lung pathophysiology and vascular physiology and disease.

I have some specific comments and questions:

Response: We thank the reviewer for the very positive comments and important suggestions to further improve our manuscript.

- In figure 2, the authors showed the relevance of different G proteins in murine pulmonary arteries and a DMR-dose response curve for 5-HT with modified PASMCs treated with sh-RNAs for G11 and Gq. The 5-HT-induced constriction was similarly reduced by

downregulation of G11 or Gq, while FR had a higher vasorelaxant effect than the downregulation of these G proteins. Does the downregulation of both G proteins inhibit the vasoconstrictor effect of 5-HT similarly as FR? The expression of these proteins and their downregulation is not shown in the manuscript. Please show the expression of these proteins and their downregulation by qRT-PCR and/or Western Blot.

Response: We thank the reviewer for raising this interesting point. mRNA expression of $G\alpha_{11}$ and $G\alpha_q$ in PSMCs with or without sh-RNA treatment is displayed in the expanded view (Fig. EV1C,D). As suggested by the reviewer we performed new experiments to assess protein expression after sh-RNA treatment using Western Blot analysis (Fig. EV1G,H). This analysis shows the downregulation of $G\alpha_{11}$ but not of $G\alpha_q$ as there are neither $G\alpha_{11}$ nor $G\alpha_q$ -specific antibodies commercially available and an antibody against both $G\alpha_q$ and $G\alpha_{11}$ still detects the more prevalent $G\alpha_{11}$ isoform after $G\alpha_q$ downregulation in PSMCs (Fig. EV1G,H). MRNA and protein analysis reveal that sh-RNA treatment strongly lowers expression of both $G\alpha_q$ and $G\alpha_{11}$ but does neither abrogate $G\alpha_q$ nor $G\alpha_{11}$ completely. Accordingly, the combination of sh-Gq and sh-G11 does not diminish the DMR signal to a similar extent as FR (Fig. 2C). This can be explained by residual $G\alpha_q$ and $G\alpha_{11}$ expression further highlighting the inhibitory power of the pharmacological pan-Gq inhibitor FR.

We included Western Blot analysis of $G\alpha_{q/11}$ expression in new Fig. EV 1G,H. The effect of combined treatment with sh-Gq and sh-G11 is included in Fig. 2C and described in the results section (p. 8, ll. 2-6) of the revised manuscript.

- The DMR method and its functional meaning should be explained in better way.

Response: As requested, we improved the description of DMR in the revised manuscript: DMR technology is an optical-based label-free detection platform that provides phenotypic measures of cellular activity when cells are exposed to pharmacologically active stimuli (Schröder *et al*, 2010; Schröder *et al*, 2011). If used to capture GPCR activation, it reports signaling along all four major G protein pathways (Schröder *et al*, 2010; Schröder *et al*, 2011; Camp *et al*, 2016).

We included this detailed explanation in the methods section of the revised manuscript (p. 22, ll. 17-21).

- Figure 3 shows that the vasorelaxant effect of FR is superior to other antagonists. Please explain why the Thromboxane antagonist (SQ) or the ETA/B receptor antagonist Bosentan only evoked 50 % vasorelaxation of the vasoconstriction induced by the Thromboxane analogue or ET-1. Are higher concentrations of SQ or Bosentan more effective?

Response: Fig. 3 shows dose response curves of pulmonary vasoconstriction induced by the constrictors U-46619 (Fig 3A) and ET-1 (Fig 3C) after pre-treatment with the inhibitor FR or the respective receptor antagonists (i.e. SQ, Bos). Both, SQ and Bos are competitive antagonists and therefore induced a right-shift of the dose-response curve by the agonist based on agonist/antagonist competition at the receptor ligand binding site. Therefore, a higher antagonist concentration results in a more pronounced right shift of the dose-response curve of U-46619 or ET-1. Since FR is known to show pseudo-irreversible binding with very slow dissociation kinetics at the Gq protein level downstream of the receptor (Schrage *et al*, 2015) the mechanisms of action of the receptor blockers and the Gq inhibitor FR as well as their efficacies at the same concentration are different.

We have emphasized the different mechanisms of action of receptor antagonists and the Gq inhibitor FR in the results section of the revised version of the manuscript (p. 8, ll. 26-27 + p. 9, ll. 1-2).

- In figures 6 and 7 the authors demonstrated that FR prevents and even reverts pulmonary

arterial hypertension in the Sugen (Su)Hox mouse model. The long duration of these studies indicates that besides vasorelaxation other mechanisms participate in the protective effects of FR. How does FR impact the proliferation of PASMCs, the survival of endothelial cells and/or pulmonary infiltration by immune cells?

Response: We thank the reviewer for this interesting question. To examine the effect of FR on PASMC proliferation and endothelial cell survival, we isolated murine PASMC (mPASMCs) by an outgrowth method and murine lung endothelial cells (mLECs) by magnet-associated cell sorting (MACS). Effects of FR on mPASMCs were analyzed in native cells and cells stimulated with a combination of PDGF and 5-HT to mimic PH conditions in cell culture (Hervé *et al*, 1995; Eddahibi *et al*, 2006; Schermuly *et al*, 2005). In native cells neither FR nor the solvent DMSO affected cell growth. Stimulation of mPASMC with PDGF and 5-HT strongly increased cell growth as assessed by cell counting after 2 days. While the solvent DMSO had no effect, Gq inhibition by FR significantly reduced mPASMC cell growth under these conditions (PDGF + 5-HT: 1.98 ± 0.07 , n=10, PDGF + 5-HT + DMSO: 1.97 ± 0.07 , n=9, PDGF + 5-HT + FR: 1.53 ± 0.06 , n=9, $p < 0.001$ vs. PDGF + 5-HT + DMSO) (new Fig. 7A). To identify the Gq family member that is responsible for this effect we also treated mPASMCs with lentiviral sh-RNAs against $G\alpha_{11}$, $G\alpha_q$ or both. Interestingly, only knockdown of $G\alpha_q$ reduced cell growth, while the knockdown of $G\alpha_{11}$ had no effect (sh-ctrl: 1.00 ± 0.04 , n=8, sh-ctrl + PDGF + 5-HT: 1.82 ± 0.08 , n=8, sh-G11 + PDGF + 5-HT: 1.90 ± 0.06 , n=8, sh-Gq + PDGF + 5-HT: 1.46 ± 0.03 , n=8, $p < 0.01$ vs. sh-ctrl + PDGF + 5-HT, sh-G11 + sh-Gq + PDGF + 5-HT: 1.46 ± 0.04 , n=8, $p < 0.01$ vs. sh-ctrl + PDGF + 5-HT), implicating that $G\alpha_q$ is more relevant than $G\alpha_{11}$ for mPASMC growth under PH conditions (new Fig. 7C).

To exclude potential detrimental effects of FR on ECs we analyzed survival of mLECs in response to FR and performed TUNEL staining in native mLECs. We found that neither DMSO nor FR induced apoptosis in these cells (new Fig. EV5D).

Pulmonary infiltration by immune cells, including macrophages, monocytes, mast cells, dendritic cells and several T cell subpopulations (Savai *et al*, 2012) is an important hallmark of PAH. Hypoxia exposure in mice leads to increasing amounts of macrophages in the lung, which promotes muscularization at least in part due to macrophage-derived PDGF and this mechanism is also of relevance in human PAH (Liu *et al*, 2023; Ntokou *et al*, 2021). To examine the effect of FR on pulmonary infiltration of macrophages in PH, we generated paraffin sections of lungs from mice that had developed PH during 5 weeks of Hx and were treated with FR or solvent for the last 2 weeks. Then, stainings against macrophages using a MAC-2 antibody were performed. We found that pulmonary infiltration by macrophages was reduced in response to FR application compared to solvent DMSO controls (DMSO: 29.7 ± 1.8 , n=9, FR: 22.6 ± 1.5 , n=9, $p < 0.05$) suggesting that FR can also reduce immune cell infiltration in the lung.

We included the data on mPASMC cell growth and mLEC survival in new Fig. 7A,C/Fig. EV5D and the results section (p. 13, ll. 12-25 + p. 14, ll. 16-19). Data on pulmonary macrophage invasion were included into new Fig. 8H and the results section of the revised manuscript (p. 15, ll. 4-5)

- Lastly: Gq-coupled receptors are also involved in the regulation of peripheral vascular resistance, for instance by Angiotensin II (AT1-R) or adiuretin/vasopressin (V1-R). How does FR affect systemic arterial blood pressure? Which unwarranted effects can be associated with systemic FR therapy? These aspects should be discussed in better way.

Response: We thank the reviewer for raising this important point. We performed additional experiments to determine the effect of chronic FR application on LVSP in healthy mice and our disease models. Similar as reported in the literature regarding the effects of FR on systemic arteries (Meleka *et al*, 2019; Crüsemann *et al*, 2018) chronic FR application in our experiments resulted in a decrease of LVSP under normoxic conditions and also when applied during 3 weeks of SuHx PH development (new Fig. EV5A). A similar effect was also found when FR was used for treatment of pre-existing PH (new Fig. EV5E). This was

expected as a reduction of the systemic blood pressure is known for nearly all drug classes that are currently used in the clinics to treat pulmonary hypertension (Sildenafil (Preston *et al*, 2005), Iloprost (Kingman *et al*, 2017), Riociguat (Ghofrani *et al*, 2013; Dumitrascu *et al*, 2006), Bosentan (Williamson *et al*, 2000)).

Pan-Gq inhibition by FR could also potentially affect platelet aggregation *in vivo*. Even though we could not detect spontaneous bleeding in response to the FR concentration used in our *in vivo* experiments, reports on the structural similar Gq inhibitor YM-254890 (Uemura *et al*, 2006) as well as on G α_q deficient mice (Offermanns *et al*, 1997) provide evidence that Gq inhibition can result in impaired platelet aggregation. While anticoagulation is no longer a standard treatment in PAH (Roldan *et al*, 2016; Jose *et al*, 2019), inhibition of hemostasis may still be beneficial for some patients (Humbert *et al*, 2022).

We included our data on LVSP in response to FR in new Fig. EV5A,E and the results section of our revised manuscript (p. 13, l. 3 + p. 15, l. 1). Moreover, we included effects of FR on systemic blood pressure and platelet aggregation into the discussion section of the revised manuscript (p. 18, ll. 10-23).

Referee #2 (Comments on Novelty/Model System for Author):

For a better translational way, authors should investigate the consequence of curative treatment with FR in the development of PH in more severe experimental models of PH as monocrotaline-PH rats or Sugen/Hypoxia-PH rats.

Referee #2 (Remarks for Author):

In their manuscript "Pharmacological Gq inhibition induces potent pulmonary vasorelaxation and reverses pulmonary hypertension", the authors investigated the potential as a therapeutic target of Gq proteins in PAH. They showed that the inhibition of Gq proteins by FR9000359 induced a potent vasorelaxation *ex vivo* in PA from healthy mice, pigs, and humans. They also demonstrated that the application of FR *in vivo* on Sugen-Hypoxia mice reversed PH. They proposed that Gq proteins could be used as a therapeutic target in PH. Their work is interesting. However, this work has methodological issues and requires additional experiments and clarifications.

Major:

1) Authors should use the new hemodynamic definition of PAH from the 2022 guidelines (Humbert *et al.*, ERJ, 2022) and the complete one, including mean pulmonary arterial pressure, pulmonary vascular resistance, and pulmonary arterial wedge pressure. They should also cite more recent papers on the prognosis of PAH.

Response: We thank the Reviewer for making this helpful suggestion. We updated and completed the hemodynamic definition from the current PAH guideline (p. 4, ll. 6-8) and cited more recent papers on the prognosis of PAH (p. 4, ll. 9-10).

2) Importantly, for a better translational way, authors should investigate the consequence of curative treatment with FR in the development of PH in more severe experimental models of PH as monocrotaline-PH rats or Sugen/Hypoxia-PH rats.

Response: As suggested by the reviewer, we performed more *in vivo* experiments to analyze the effect of FR on the systemic blood pressure in healthy and PH mice. We assessed the Fulton index, potential right ventricular fibrosis and the amount of muscularized PAs in mouse; moreover, we analyzed the kinetics of the acute FR effect on RVSP and LVSP in PH mice. Additionally, we induced PH in mice to perform isometric force measurements in order to test the effect of FR in isolated vessels of PH mice (see all below).

We agree that in the future these experiments should be also performed in another species, but we consider this to go beyond the scope of the current manuscript.

3) Authors used the shRNA strategy to reduce the expression of Gq and G11 and to measure the consequence of Gq or G11 knockdown on pulmonary arterial contraction. In addition to pulmonary artery vasoconstriction, PAH is also characterized by strong pulmonary arterial remodeling involving at least exacerbated PASMCM proliferation and migration. In this way, authors should take advantage of their Gq inhibitor and their shRNA to determine the role of Gq and G11 in PASMCM proliferation and migration from control and PH conditions.

Response: We thank the reviewer for this important question.

To examine the effect of FR, $G\alpha_q$ - and/or $G\alpha_{11}$ sh-RNA on PASMCM proliferation and migration, we isolated murine PASMCM (mPASMCMs) by an outgrowth method and analyzed the effect in native cells and cells stimulated with a combination of PDGF and 5-HT. These compounds were chosen to mimic PH conditions in cell culture as both are known to be drivers of PH pathophysiology (Hervé *et al*, 1995; Eddahibi *et al*, 2006; Schermuly *et al*, 2005). In native cells neither FR nor the solvent DMSO affected cell growth (native: 1.00 ± 0.06 , n=12, DMSO: 1.01 ± 0.03 , n=9, FR: 1.00 ± 0.04 , n=9). Stimulation of mPASMCMs with a combination of PDGF and 5-HT strongly increased cell growth as assessed by cell counting after 2 days. While the solvent DMSO had no effect, Gq inhibition by FR significantly reduced mPASMCM cell growth under these conditions (PDGF + 5-HT: 1.98 ± 0.07 , n=10, PDGF + 5-HT + DMSO: 1.97 ± 0.07 , n=9, PDGF + 5-HT + FR: 1.53 ± 0.06 , n=9, $p < 0.001$ vs. PDGF + 5-HT + DMSO) (new Fig. 7A). To identify the Gq family member that is responsible for this effect we also treated mPASMCMs with lentiviral sh-RNAs against $G\alpha_{11}$, $G\alpha_q$ or both. Interestingly, only knockdown of $G\alpha_q$ reduced cell growth, while the knockdown of $G\alpha_{11}$ had no effect (sh-ctrl + PDGF + 5-HT: 1.82 ± 0.08 , n=8, sh-G11 + PDGF + 5-HT: 1.90 ± 0.06 , n=8, sh-Gq + PDGF + 5-HT: 1.46 ± 0.03 , n=8, $p < 0.01$ vs. sh-ctrl + PDGF + 5-HT, sh-G11 + sh-Gq + PDGF + 5-HT: 1.46 ± 0.04 , n=8, $p < 0.01$ vs. sh-ctrl + PDGF + 5-HT), implicating that $G\alpha_q$ is more relevant than $G\alpha_{11}$ for mPASMCM growth under PH conditions (new Fig. 7C).

Migration was investigated by a wound healing/scratch assay. Under native conditions FR had no effect. Stimulation of the cells with PDGF and 5-HT (PH condition) increased wound healing after 12 hours (native: 32.9 ± 2.5 %, n=8 vs. PDGF + 5-HT: 64.6 ± 3.7 %, n=6, $p < 0.001$). This increase was not affected by the solvent DMSO but significantly reduced by pharmacological Gq inhibition using FR (PDGF + 5-HT + DMSO: 63.4 ± 2.9 %, n=6 vs. PDGF + 5-HT + FR: 45.4 ± 3.1 %, n=6, $p < 0.01$) (new Fig. 7B). Again, this result was confirmed in cells treated with lentivirus-based sh-G11, sh-Gq or sh-G11+sh-Gq transduction. Similar to our results in the cell growth assay the knockdown of $G\alpha_{11}$ alone had no effect, whereas the knockdown of $G\alpha_q$ reduced wound healing significantly (sh-ctrl + PDGF + 5-HT: 65.7 ± 1.8 %, n=6, sh-G11 + PDGF + 5-HT: 62.3 ± 1.6 %, n=6, sh-Gq + PDGF + 5-HT: 45.9 ± 2.0 %, n=6, $p < 0.001$ vs. sh-ctrl + PDGF + 5-HT, sh-G11 + sh-Gq + PDGF + 5-HT: 45.7 ± 1.6 %, n=6, $p < 0.001$ vs. sh-ctrl + PDGF + 5-HT) indicating that the inhibition of $G\alpha_q$ is responsible for the anti-remodeling effect of FR in PASMCMs (new Fig. 7D). These data were included in new Fig. 7 and the results section of the revised manuscript (p. 13, ll. 12-28 + p. 14, ll. 1-15).

4) In line with point #3, as we know, Gq is primordial for intracellular Ca^{2+} signaling. Because Gq activation activates PLC, which leads to diacylglycerol and IP3 production, and consequently induces an increase of intracellular Ca^{2+} concentration via IP3R localized at ER membrane as well as store-operated Ca^{2+} channel localized at the plasma membrane including Orai and TRPC channels.

Orai and TRPC channels were recently demonstrated to be upregulated in PAH and to be considered as potential therapeutic targets.

Are there any consequences of Gq inhibition on Orai and TRPC expression and IP3R Ca^{2+} release? By reducing the function of these Ca^{2+} channels, FR treatments should also reduce PASMCM proliferation, migration since these processes are strongly dependent on intracellular Ca^{2+} concentration.

Authors should investigate and discuss these points in their study.

Response: To analyze the effect of FR on Orai1 and TRPC expression we applied native mPASCs and cells exposed to PDGF+5-HT (PH condition). We analyzed Orai1, TRPC1 and TRPC3 expression as these were shown to be increased in human and/or mouse PASCs in PH (Masson *et al*, 2022; Masson *et al*, 2023). Our qPCR analysis revealed that only Orai1 and TRPC1 but not TRPC3 channels were expressed in our mPASCs. Under PH conditions (PDGF+5-HT) Orai1 (native: $2.0 \pm 0.2 \cdot 2^{-\Delta Ct} \times 10^6$, n= 8 vs. PDGF + 5-HT: $3.2 \pm 0.3 \cdot 2^{-\Delta Ct} \times 10^6$, n=6, p<0.01) but not TRPC1 expression was found to be increased. Interestingly, FR did neither affect Orai1 nor TRPC1 expression under both conditions suggesting that FR does not affect Ca²⁺ signaling via modulation of Orai1 and TRPC1/3 expression.

As suggested by the reviewer we also investigated effects on intracellular Ca²⁺ in mPASCs. Therefore, we applied a combination of PDGF and 5-HT to mimic PH conditions and measured intracellular Ca²⁺ concentration. The Ca²⁺ response to PDGF + 5-HT consisted of two phases: a fast and small Ca²⁺ increase induced by 5-HT, that could be inhibited by FR (new Fig. 7 E-G) and a delayed stronger Ca²⁺ elevation that was induced by PDGF and was FR-insensitive (new Fig. 7E,F,H). Interestingly, the impact of FR on intracellular Ca²⁺ was minor, while its inhibitory effect of proliferation and migration was much more pronounced (see above).

The qPCR data on Orai1 and TRPC expression were included in new Fig. EV5B,C and the Ca²⁺ data in new Fig. 7E-H as well as the discussion section (p. 18, ll. 26-28 + p. 19, ll. 1) of the revised manuscript.

Moreover, regarding this large spectrum of action of Gq inhibitor, how authors have measured the potential side effects of Gq inhibition in other organs? Authors should investigate the consequences of Gq inhibition in healthy mice or rat.

Response: We thank the reviewer for raising this important point. The application of FR did not produce overt alterations in mouse behavior or locomotion. In healthy mice we can show that FR does neither affect basal RVSP nor heart rate after acute (Fig. EV4D,E) or chronic application (Fig. 6A,B). Effects on RVSP are most likely absent because there is little or no resting tone in pulmonary arteries (Wilkins *et al*, 1996).

We also performed new experiments and determined the effect of chronic FR application on LVSP in healthy mice and our disease models. In accordance with earlier results on the effects of FR on systemic arteries (Meleka *et al*, 2019; Crüsemann *et al*, 2018) we found that acute and chronic FR application resulted in a decrease of LVSP under normoxic conditions and when applied in our disease models (new Fig. EV5A,E). This was expected as a reduction of the systemic blood pressure is known for nearly all drug classes that are currently used in the clinics to treat PAH (Sildenafil (Preston *et al*, 2005), Iloprost (Kingman *et al*, 2017), Riociguat (Ghofrani *et al*, 2013; Dumitrascu *et al*, 2006), Bosentan (Williamson *et al*, 2000)).

Pan-Gq inhibition by FR could potentially also affect platelet aggregation *in vivo*. Even though we could not detect spontaneous bleeding in response to the FR concentration used in our *in vivo* experiments, reports on the structural similar Gq inhibitor YM-254890 (Uemura *et al*, 2006) as well as on Gα_q deficient mice (Offermanns *et al*, 1997) provide evidence that Gq inhibition can result in impaired platelet aggregation. While anticoagulation is no longer a standard treatment in PAH (Roldan *et al*, 2016; Jose *et al*, 2019), inhibition of hemostasis may still be beneficial for some patients (Humbert *et al*, 2022).

We included data on LVSP changes in response to FR in new Fig. EV5A,E and the results section (p. 13, l. 3 + p. 15, l. 1). Moreover, we included effects of FR on systemic blood pressure and platelet aggregation into the discussion section of the revised manuscript (p. 18, ll. 10-23).

In addition, authors should not limit their explanations of the increasing heart rate by IP3R inhibition (page 25).

Response: We found that 1h after FR application heart rate was unaffected in healthy but slightly reduced in PH mice. This could be explained by inhibition of the accelerating function of Gq proteins for GIRK channel deactivation kinetics resulting in an increased heart rate (Mark *et al*, 2000) or the inhibition of Gq-dependent IP₃-mediated Ca²⁺ release that has also been shown to increase pacemaker activity in the heart (Ju *et al*, 2012). In addition, it has been reported that Gq inhibition alters arterial baroreflex control (Meleka *et al*, 2019). We included all these explanations in the discussion section of our revised manuscript (p. 17, ll. 24-28 + p.18, l. 1).

5) Regarding *in vivo* experiments in mice (preventive and curative approach), authors should also measure the consequences of FR treatment on right ventricle fibrosis. Authors should add the Fulton index (RV/LV+septum) values for *in vivo* experiments. Authors should examine the rate of muscularized and non-muscularized pulmonary arteries by using alpha-SMA/vWF staining.

Response: We thank the reviewer for this helpful suggestion. We focused our analysis on the curative approach since this is most relevant for translation (Brown *et al*, 2011). We analyzed right ventricular collagen deposition as a sign for potential fibrosis of the heart by morphometry of Sirius red-stained paraffin sections. Quantification of collagen revealed that there is hardly any collagen deposition and no cardiac fibrosis after 5 weeks in the PH mouse model of SuHx (see also Fig. 1 of the rebuttal below). In addition, chronic FR application did not induce cardiac fibrosis as potential adverse effect. (DMSO: 4.5 ± 0.9 %, n=9 vs. FR: 3.6 ± 0.8 %, n=9)

We included the quantitative analysis for right ventricular collagen deposition after FR treatment in new Fig. EV5H.

Rebuttal Fig. 1. 5 weeks of Hx does not induce right ventricular fibrosis in mouse. Sirius red staining of a paraffin section derived from a DMSO- treated mouse with Su/Hx after 5 weeks. Boxes indicate areas magnified in the pictures on the right. Magnification: left: 1x, middle: 10x, right: 40x.

As suggested by the reviewer we also analyzed the Fulton index in our therapeutic approach. Our data demonstrate that FR was able to reverse right heart hypertrophy (DMSO: 44.3 ± 1.9 %, n=7 vs. FR: 33.2 ± 1.1 %, n=5, p<0.01). This result is in line with our analysis of the cross-sectional area of cardiomyocytes (see Fig. 8J-L).

We included these data regarding the Fulton index in new Fig 8I of the revised manuscript.

As suggested by the reviewer, we performed alpha-SMA/vWF stainings to examine the amount of muscularized and non-muscularized PAs in lung sections of PH animals treated with FR. In accordance with our results on PA vascular wall thickness (Fig. 8C-E) and

media/CSA ratio (Fig. 8F) FR reduced the amount of fully muscularized lung vessels and enhanced the amount of non and partially muscularized lung vessels. Also these data were included into new Fig. EVF,G and Fig. 8G.

6) Results showing reduction of RVSP in PH mice by FR treatment (1 hour after FR application) are exciting but limited due to the use of the mouse model. Authors should also do these experiments in monocrotaline or Sugen/hypoxia rats to better define the acute consequence of FR infusion in RVSP, cardiac output, and systemic blood pressure. Because of the rapid PA relaxation produced by FR application, we could hypothesize that FR should reduce RVSP in these PH models more rapidly than one hour.

Is there any consequence of Gq inhibition on systemic arterial tone? It should be.

Response: We thank the reviewer for this suggestion. To characterize the kinetics of FR on blood pressure in the pulmonary and systemic circulation in detail, we performed additional *in vivo* experiments in mice with established SuHx PH. In these animals we recorded the RVSP and LVSP simultaneously using two Millar pressure catheters in the right and left ventricle, respectively. After baseline recordings (RVSP: DMSO group: 35.6 ± 1.3 mmHg, n=4 vs. FR group: 35.2 ± 1.6 mmHg, n=5; LVSP: DMSO group: 87.1 ± 3.6 mmHg, n=4 vs. FR group: 86.8 ± 1.5 mmHg, n=5) we applied FR (10 μ g/mouse) or DMSO as solvent control via the i.p. route as in our chronic FR experiments and monitored changes in pressure over the first 20 minutes (see Fig. EV4H,I). When FR was administered a small decrease of pressure could be observed starting from 5 min on. After 20 minutes RVSP as well as LVSP were reduced by around 13% each, while in controls blood pressure was stable over time (RVSP: 99.3 ± 0.9 %, LVSP: 103.3 ± 1.1 %)

These data suggest that the acute relative effects of FR on the pulmonary and systemic circulation are similar.

We added these data in new Fig. EV4H,I and the results as well as the discussion section (p. 18, ll. 10-16) of the revised manuscript.

Experiments in another species are beyond the scope of the current manuscript and need to be performed in future studies.

7) In the continuity of point #6, the authors should assess the effect of the Gq-inhibition on the pulmonary vascular tone of the PH condition. Does FR application always induce pulmonary artery relaxation?

Response: As suggested by the reviewer we investigated the effect of FR on pulmonary vascular tone of mice with pre-existing PH (3 weeks SuHx) in isometric force measurements. Interestingly, we found that the pre-constriction amplitude by 5-HT was strongly increased in mice with pre-existing PH compared to our experiments in healthy mice (see Fig. 1) (Nx: 2.5 ± 0.1 mN, n=22 vs. Hx: 5.1 ± 0.2 , n=12, $p < 0.001$) reflecting the successful induction of PH in these mice. Importantly, the relaxation induced by FR in PAs of mice with pre-existing PH was similar compared to that of healthy mice (Nx FR: 79.2 ± 2.5 %, n=12 vs Hx FR: 88.0 ± 1.5 %, n=6 $p < 0.05$, at 8 min after FR application), highlighting the strong vasorelaxant effect of FR even under PH conditions.

We included these data into new Fig. 5I,J and the results section of the revised manuscript (p. 11, ll. 7-16).

Minor:

1) Why, in the presence of FR (10^{-6} M), there is no constriction of PA-induced by 5-HT dose response (Figure 2A). Because, with FR at 10^{-6} M, PA relaxation is around 80% with a residual 20% of contraction.

Response: In our single dose experiments of FR after 5-HT constriction, we analyzed vasorelaxation by FR at 8 min after FR application (Fig. 1F). After this time-point isometric force continued to decrease slowly, but it was unclear if this could be attributed to the action

of FR and therefore it was not systematically analyzed. To test if FR can completely abolish constriction by 5-HT in the next series of experiments we pre-incubated PAs with FR and then performed 5-HT dose response curves (Fig. 2A). In these experiments FR prevented 5-HT-induced vasoconstriction completely (Fig. 2B) indicating that FR can completely abrogate 5-HT-dependent constriction.

In addition, in myograph experiments with PAs of mice with pre-existing PH (see point 7 above) after 5-HT pre-constriction and FR application we recorded the vasorelaxation by FR for a longer time-period and found that at 15 min after FR application relaxation reached about 100% demonstrating that FR can also completely diminish constriction when applied after 5-HT.

To make this clearer, we improved the description of the experimental approach related to isometric force measurements in the method section (p. 24, ll. 6-8) and included values for FR-induced relaxation in PH animals at 15 min (p. 11, ll. 15-16).

2) The red trace of FR condition is absent from the graph presented in Figure 2C.

Response: This experiment was designed to characterize the differential contribution of $G\alpha_q$ and $G\alpha_{11}$ to acute 5-HT-dependent effects in PSMCs. Accordingly, we used FR (10^{-6} M) as a positive endpoint control only at the highest α -methyl-5-HT (10^{-6} M) concentration in native cells. Therefore, only one data point has been recorded and can be displayed.

We better explained this procedure in the methods section of the revised manuscript (p. 23, ll. 3-5).

3) Please add a comma between "mouse" and "pig" in the abstract and between "PH" and "FR" in the discussion (page 25).

Response: Done

4) Please homogenize terms as "in vivo" in italics or not.

Response: Done

5) Authors should improve the visual quality of Figure 2.

Response: Done

Referee #3 (Comments on Novelty/Model System for Author):

In vitro, ex vivo and in vivo models were all used in this study and the findings seemed all consistent with the overall hypothesis.

Referee #3 (Remarks for Author):

This manuscript reports the pharmacological role of a pan-Gq protein inhibitor FR900359 (FR) in relaxing of pulmonary artery and managing pulmonary hypertension. Specifically, the authors found that the FR compound at low micromolar concentrations induced significant vasodilation in several ex vivo models of mouse, pig, and human pulmonary arteries. In addition, they found that local delivery or i.p. delivery FR decreased pulmonary hypertension in a couple of relevant mouse models. Overall, the study has a translational potential and the experiments appeared to be well-performed. However, the idea of targeting a Gq protein to control vascular tone is not particularly novel, and the quality of this work would be further improved if the following points could be considered or clarified:

Response: We thank the reviewer for the positive comments and helpful suggestions.

1. Page 3, ET-1 and 5-HT should be fully defined when they first appeared.

Response: We defined ET-1 and 5-HT at first appearance in the manuscript (p. 4, l. 11).

2. Fig. 1, it would be better if the authors could show sample tracing on the effect of FR on KCl-induced vascular constriction.

Response: We thank the reviewer for this helpful suggestion. As suggested by the reviewer, we included an original trace of KCl-induced constriction followed by solvent DMSO or FR (10^{-6} M) application in the plateau phase in new Fig. 1D,E.

3. Fig. EV1, panel B, the RT-PCR study did not have positive and negative controls.

Response: We thank the reviewer for raising this important point. In Fig. EV1B we added mouse lung tissue as positive control and mPASCs after knockdown of $G\alpha_q$ or $G\alpha_{11}$ as negative controls.

4. Lentiviral transduction decreased Gq and $G11$ mRNA expression, but it remained unknown how much $Gq/11$ protein expression was reduced.

Response: As suggested by the reviewer we performed additional experiments to analyze protein expression of $G\alpha_q$ and/or $G\alpha_{11}$ after lentiviral downregulation. This analysis shows the downregulation of $G\alpha_{11}$ but not $G\alpha_q$ as there are neither $G\alpha_{11}$ nor $G\alpha_q$ -specific antibodies commercially available and an antibody against both $G\alpha_q$ and $G\alpha_{11}$ still detects the more prevalent $G\alpha_{11}$ isoform after $G\alpha_q$ downregulation in PASCs (Fig. EV1G,H). We included the data regarding $G\alpha_{q/11}$ protein expression in EV1G,H and in the results section of the revised manuscript (p. 7, ll. 22-25).

5. Why in the pig species, FR showed at least 10-fold more sensitive than in mice?

Response: To compare the effects of FR in pig and mouse we re-analyzed FR dose response curves after U46619-dependent constriction as only this constrictor was applied in both species so that FR effects can be correlated. (Fig. EV2D,E). When we calculated IC_{50} values for FR we found that these are quite similar in both species (see bar diagram below).

Rebuttal Fig. 2. Analysis of IC_{50} values of FR relaxation after U-46619 pre-constriction in mouse and pig. Unpaired student's t-test.

6. It is not clear for all the vascular contractility study if the endothelium were all removed or not. Please specify this in the method section.

Response: We thank the reviewer for this suggestion. The endothelium was preserved in all experiments. This information was included in the methods section of the revised manuscript (p. 23, l. 25 + p. 24, ll. 14-15).

References

- Brown LM, Chen H, Halpern S, Taichman D, McGoon MD, Farber HW, Frost AE, Liou TG, Turner M, Feldkircher K et al (2011) Delay in recognition of pulmonary arterial hypertension: factors identified from the REVEAL Registry. *Chest* 140: 19–26
- Camp ND, Lee K-S, Cherry A, Wacker-Mhyre JL, Kountz TS, Park J-M, Harris D-A, Estrada M, Stewart A, Stella N et al (2016) Dynamic mass redistribution reveals diverging importance of PDZ-ligands for G protein-coupled receptor pharmacodynamics. *Pharmacol Res* 105: 13–21
- Crüseemann M, Reher R, Schamari I, Brachmann AO, Ohbayashi T, Kuschak M, Malfacini D, Seidinger A, Pinto-Carbó M, Richarz R et al (2018) Heterologous Expression, Biosynthetic Studies, and Ecological Function of the Selective Gq-Signaling Inhibitor FR900359. *Angew Chem Int Ed Engl* 57: 836–840
- Dumitrascu R, Weissmann N, Ghofrani HA, Dony E, Beuerlein K, Schmidt H, Stasch J-P, Gnoth MJ, Seeger W, Grimminger F et al (2006) Activation of soluble guanylate cyclase reverses experimental pulmonary hypertension and vascular remodeling. *Circulation* 113: 286–295
- Eddahibi S, Guignabert C, Barlier-Mur A-M, Dewachter L, Fadel E, Dartevielle P, Humbert M, Simonneau G, Hanoun N, Saurini F et al (2006) Cross talk between endothelial and smooth muscle cells in pulmonary hypertension: critical role for serotonin-induced smooth muscle hyperplasia. *Circulation* 113: 1857–1864
- Ghofrani H-A, D'Armini AM, Grimminger F, Hoeper MM, Jansa P, Kim NH, Mayer E, Simonneau G, Wilkins MR, Fritsch A et al (2013) Riociguat for the treatment of chronic thromboembolic pulmonary hypertension. *N Engl J Med* 369: 319–329
- Hervé P, Launay J-M, Scrobohaci M-L, Brenot F, Simonneau G, Petitpretz P, Poubeau P, Cerrina J, Duroux P, Drouet L (1995) Increased plasma serotonin in primary pulmonary hypertension. *The American Journal of Medicine* 99: 249–254
- Humbert M, Kovacs G, Hoeper MM, Badagliacca R, Berger RMF, Brida M, Carlsen J, Coats AJS, Escribano-Subias P, Ferrari P et al (2022) 2022 ESC/ERS Guidelines for the diagnosis and treatment of pulmonary hypertension. *Eur Heart J* 43: 3618–3731
- Jose A, Eckman MH, Elwing JM (2019) Anticoagulation in pulmonary arterial hypertension: a decision analysis. *Pulm Circ* 9: 2045894019895451
- Ju Y-K, Woodcock EA, Allen DG, Cannell MB (2012) Inositol 1,4,5-trisphosphate receptors and pacemaker rhythms. *J Mol Cell Cardiol* 53: 375–381
- Kingman M, Archer-Chicko C, Bartlett M, Beckmann J, Hohsfield R, Lombardi S (2017) Management of prostacyclin side effects in adult patients with pulmonary arterial hypertension. *Pulm Circ* 7: 598–608
- Liu H, Wang Y, Zhang Q, Liu C, Ma Y, Huang P, Ge R, Ma L (2023) Macrophage-derived inflammation promotes pulmonary vascular remodeling in hypoxia-induced pulmonary arterial hypertension mice. *Immunol Lett* 263: 113–122
- Mark MD, Ruppertsberg JP, Herlitze S (2000) Regulation of GIRK channel deactivation by Gαq and Gai/o pathways. *Neuropharmacology* 39: 2360–2373
- Masson B, Le Ribeuz H, Sabourin J, Laubry L, Woodhouse E, Foster R, Ruchon Y, Dutheil M, Boët A, Ghigna M-R et al (2022) Orai1 Inhibitors as Potential Treatments for Pulmonary Arterial Hypertension. *Circ Res* 131: e102-e119
- Masson B, Saint-Martin Willer A, Dutheil M, Penalva L, Le Ribeuz H, El Jekmek K, Ruchon Y, Cohen-Kaminsky S, Sabourin J, Humbert M et al (2023) Contribution of transient receptor potential canonical channels in human and experimental pulmonary arterial hypertension. *Am J Physiol Lung Cell Mol Physiol* 325: L246-L261
- Meleka MM, Edwards AJ, Xia J, Dahlen SA, Mohanty I, Medcalf M, Aggarwal S, Moeller KD, Mortensen OV, Osei-Owusu P (2019) Anti-hypertensive mechanisms of cyclic depsipeptide inhibitor ligands for Gq/11 class G proteins. *Pharmacol Res* 141: 264–275

- Ntokou A, Dave JM, Kauffman AC, Sauler M, Ryu C, Hwa J, Herzog EL, Singh I, Saltzman WM, Greif DM (2021) Macrophage-derived PDGF-B induces muscularization in murine and human pulmonary hypertension. *JCI Insight* 6
- Offermanns S, Toombs CF, Hu YH, Simon MI (1997) Defective platelet activation in G alpha(q)-deficient mice. *Nature* 389: 183–186
- Preston IR, Klinger JR, Houtches J, Nelson D, Farber HW, Hill NS (2005) Acute and chronic effects of sildenafil in patients with pulmonary arterial hypertension. *Respir Med* 99: 1501–1510
- Roldan T, Landzberg MJ, Deicicchi DJ, Atay JK, Waxman AB (2016) Anticoagulation in patients with pulmonary arterial hypertension: An update on current knowledge. *J Heart Lung Transplant* 35: 151–164
- Savai R, Pullamsetti SS, Kolbe J, Bieniek E, Voswinckel R, Fink L, Scheed A, Ritter C, Dahal BK, Vater A et al (2012) Immune and inflammatory cell involvement in the pathology of idiopathic pulmonary arterial hypertension. *Am J Respir Crit Care Med* 186: 897–908
- Schermuly RT, Dony E, Ghofrani HA, Pullamsetti S, Savai R, Roth M, Sydykov A, Lai YJ, Weissmann N, Seeger W et al (2005) Reversal of experimental pulmonary hypertension by PDGF inhibition. *J Clin Invest* 115: 2811–2821
- Schrage R, Schmitz A-L, Gaffal E, Annala S, Kehraus S, Wenzel D, Büllsbach KM, Bald T, Inoue A, Shinjo Y et al (2015) The experimental power of FR900359 to study Gq-regulated biological processes. *Nat Commun* 6: 10156
- Schröder R, Janssen N, Schmidt J, Kebig A, Merten N, Hennen S, Müller A, Blättermann S, Mohr-Andrä M, Zahn S et al (2010) Deconvolution of complex G protein-coupled receptor signaling in live cells using dynamic mass redistribution measurements. *Nat Biotechnol* 28: 943–949
- Schröder R, Schmidt J, Blättermann S, Peters L, Janssen N, Grundmann M, Seemann W, Kaufel D, Merten N, Drewke C et al (2011) Applying label-free dynamic mass redistribution technology to frame signaling of G protein-coupled receptors noninvasively in living cells. *Nat Protoc* 6: 1748–1760
- Uemura T, Kawasaki T, Taniguchi M, Moritani Y, Hayashi K, Saito T, Takasaki J, Uchida W, Miyata K (2006) Biological properties of a specific Galpha q/11 inhibitor, YM-254890, on platelet functions and thrombus formation under high-shear stress. *Br J Pharmacol* 148: 61–69
- Wilkins MR, Zhao L, al-Tubuly R (1996) The regulation of pulmonary vascular tone. *Br J Clin Pharmacol* 42: 127–131
- Williamson DJ, Wallman LL, Jones R, Keogh AM, Scroope F, Penny R, Weber C, Macdonald PS (2000) Hemodynamic effects of Bosentan, an endothelin receptor antagonist, in patients with pulmonary hypertension. *Circulation* 102: 411–418

17th May 2024

Dear Prof. Wenzel,

Thank you for the submission of your revised manuscript to EMBO Molecular Medicine. Your manuscript has now been re-reviewed by the three original reviewers. Based on their advice (included below), I am pleased to inform you that we will be able to accept your manuscript pending the following final amendments and appropriate response to reviewers (particularly Reviewer 2):

1) Author contributions: Please remove from the manuscript and specify author contributions in our submission system. CRediT has replaced the traditional author contributions section because it offers a systematic machine-readable author contributions format that allows for more effective research assessment. You are encouraged to use the free text boxes beneath each contributing author's name to add specific details on the author's contribution. More information is available in our guide to authors:

<https://www.embopress.org/page/journal/17574684/authorguide#authorshipguidelines>

2) In the Materials and Methods, please take care of the following:

- Studies with human research participants: The use of human samples requires information on the authority granting ethics approval (e.g. IRB) and informed consent. If the need for approval is waived, please cite the reason (e.g. non-human subject research because the samples used were de-identified/coded with no identifying information) and legislation in the relevant methods section. Please also state that the experiments conformed to the principles set out in the WMA Declaration of Helsinki and the Department of Health and Human Services Belmont Report. Finally, please update the "Author Checklist" to indicate that this information was included in the manuscript.

- Animals: Please ensure that an ethics statement and the approval committee for research on animals is included in the section where animal experiments are described in the Materials and Methods. Please also ensure that specific housing conditions as well as gender, age, and origin of the animals involved in experiments is reported (beyond a statement that the experiments complied). The specific conditions are currently missing for the mouse experiments, and no ethics or additional information has been given for the pig experiments.

3) Please place individual sections of the manuscript in the following order: Title page - Abstract & Keywords - Introduction - Results - Discussion - Materials & Methods - Data Availability - Acknowledgements - Disclosure and Competing Interests Statement - The Paper Explained - For More Information - References - Figure Legends - Expanded View Figure Legends.

4) For the figures and figure legends, please take care of the following:

- Please make sure to update the callouts of all figures in the main manuscript text (currently figure callouts are missing for Figure 4e)

- Please note that information related to n is missing in the legends of figures 5m; 7a-h; EV 1h; EV 4e-f; EV 5d.

- Although 'n' is provided, please describe the nature of entity for 'n' in the legends of figures 1f; 2c; 4d, f; 5c, f, h; EV 1c-f; EV 4c, EV 5b-c.

- Please note that the error bars are not defined in the legends of EV 2c-g.

5) Synopsis:

- Synopsis text: The standfirst is over the limit of maximum of 300 characters (including spaces), please shorten this sentence. We would suggest simply removing the first sentence.

6) For more information: This space should be used to list relevant web links for further consultation by our readers. Could you identify some relevant ones and provide such information as well? Some examples are patient associations, relevant databases, OMIM/proteins/genes links, author's websites, etc...

7) As part of the EMBO Publications transparent editorial process initiative (see our policy here:

https://www.embopress.org/transparent-process#Review_Process), EMBO Molecular Medicine will publish online a Peer Review File (PRF) to accompany accepted manuscripts. This file will be published in conjunction with your paper and will include the anonymous referee reports, your point-by-point response and all pertinent correspondence relating to the manuscript. Let us know whether you agree with the publication of the PRF and as here, if you want to remove or not any figures from it prior to publication. Please note that the Authors checklist will be published at the end of the PRF.

8) Please provide a point-by-point letter INCLUDING my comments as well as the reviewer's reports and your detailed responses (as Word file).

I look forward to reading a new revised version of your manuscript as soon as possible.

Yours sincerely,

Poonam Bheda

**** Reviewer's comments ****

Referee #1 (Comments on Novelty/Model System for Author):

The authors performed a "tour de force" of new experiments in vitro, ex vivo and in vivo to address and answer all reviewer's questions. The number and quality of the technical approaches used here to answer the question whether Gq inhibition can attenuate PH is impressive and few laboratories are able to combine such different approaches in such careful dedicated way and at such a high level.

Referee #1 (Remarks for Author):

The study entitled "Pharmacological Gq inhibition induces strong pulmonary vasorelaxation and reverses pulmonary hypertension" by Alexander Seidinger et al tested the pulmonary vasodilating effect of the macrocyclic depsipeptide FR900359 (FR), a highly specific inhibitor of Gq/11/14 signalling, and its potential in the treatment of pulmonary hypertension (PH). Pulmonary hypertension (PH), with elevated pulmonary arterial pressure (mPAP {greater than or equal to} 20 mmHg) and pulmonary vascular resistance, affects up to 100 million people worldwide. Patients with PAH (PH Group 1) are presently treated with drugs targeting three dysfunctional endothelial pathways: augmentation of the nitric oxide (NO)/cyclic GMP pathway (soluble guanylyl cyclase activators, phosphodiesterase 5 inhibitors), endothelin-1 (ET-1) receptor antagonists; and prostacyclin analogs. Despite therapeutical progresses, PAH remains a fatal disease with a 3-year survival rate of 58%. Treatment of groups 2 to 5 PH addresses the causal disease and PAH therapeutics are presently not recommended. However, the presence of even moderate PH increases morbidity and mortality. Hence, new approaches to the management of all these patients are urgently needed.

The pathophysiology of PH is multifactorial and includes chronic or recurrent hypoxia and augmented (peri)vascular expression of vasoconstrictors, growth factors and inflammatory mediators, such as endothelin, serotonin, thromboxane A2 and others. Ultimately, this leads to endothelial cell dysfunction and increased contraction, proliferation, and apoptosis resistance of pulmonary artery smooth muscle cells (PASMCs) with remodeling of small pulmonary arterioles and muscularization of very distal, normally non-muscularized vessels.

Here, the authors combined elegant studies in isolated murine and porcine pulmonary arteries, cultured mouse PASMCs and precision-cut slices from human and murine lungs to study the vasodilatory efficacy and potency of the specific Gq inhibitor FR on serotonin-, thromboxane- and endothelin-1-evoked vasoconstrictions. The results demonstrated that FR markedly prevents and reverses the vasoconstrictor effects of these factors in the pulmonary macro- and microcirculation. Moreover, FR also inhibits the growth factor (e.g. PDGF-BB) stimulated proliferation of cultured PASMC. This is important because besides vasoconstriction, the hyperproliferation of vascular mural cells and vascular remodeling/thickening have a major role in the development of PH.

These interesting observations were extended with experiments in isolated perfused murine lungs. The results demonstrated that FR does not inhibit hypoxia-driven vasoconstrictions, but efficiently counter-regulates vasoconstrictions elicited by agonists acting on Gq-coupled receptors.

To elucidate the relevance of this observations for the therapy of PH, the authors firstly tested the effect of local, intratracheal application of FR on acute serotonin-evoked pulmonary vasoconstrictions in anesthetized mice (the subsequent increases in pulmonary arterial pressure, PAP, were indirectly measured through recordings of right ventricular systolic pressure, RVSP). Notably, FR markedly reduced serotonin-evoked increases in RVSP and this effect was preserved in mice with hypoxia-induced PH. Lastly, the authors studied the effect of FR in the preclinical model of Sugen/hypoxia (SuHx)-driven chronic PH in mice. Notably, FR markedly prevented and even reverted Sugen/HOX-driven PH, pulmonary vascular remodeling as well as right ventricular hypertrophy. Taken together, the results from these complex experiments in vitro/ex vivo and in vivo indicate the potential of pharmacological inhibition of Gq proteins in the treatment of PH. The possible human relevance is emphasized by the results obtained from experiments with "cultured" precision-cut slices from human lungs.

This is a very elegant study with high clinical relevance. The experiments and results are well described and properly discussed. The authors performed a "tour de force" of new experiments in vitro and in vivo to address and answer all reviewer's questions. The number and quality of the technical approaches used here to answer the question whether Gq inhibition can attenuate PH is impressive and few laboratories are able to combine such different approaches at such a high level. The introduction and

discussion sections have also been improved to address reviewers' comments. In particular the possible systemic effects of pharmacological Gq inhibition have been carefully discussed. In my view, this manuscript will be very interesting for clinicians and scientists working in the fields of lung pathophysiology and vascular physiology and disease.

Referee #2 (Comments on Novelty/Model System for Author):

Authors strongly improve the quality of their manuscript. We greatly appreciate the authors' efforts to respond to our concerns. However, we still have some concerns regarding the news of Calcium experiments, which are a little bit strange. We also asked the authors to write a limitation section because they did not use a severe model of PAH and because they found that FR treatment reduced LV blood pressure, which is not necessarily beneficial for PAH patients.

Referee #2 (Remarks for Author):

Authors strongly improve the quality of their manuscript. The results presented in the revised manuscript are still exciting, and we greatly appreciate the authors' efforts to respond to our concerns. However, we have still some concerns.

Major:

1-We understand that the authors did not perform in vivo pharmacological experiments in more severe animal models of PAH (Monocrotaline rats or Su/Hx rats). In our opinion, these experiments should be crucial for a better translational perspective. The authors should at least write a limitation section regarding the use of only the mouse model of PAH.

2-Regarding the new Ca²⁺ imaging experiments. We do not understand the presented traces. Maybe authors should better legend these traces. When drug is applied? What corresponds to the second phase of these Ca²⁺ signals? The authors should perform these Ca²⁺ experiments in a Ca²⁺-free medium to determine where this second phase comes from.

In the text, the authors explain that they used A23187. Does it correspond to the second phase of the Ca²⁺ response? If yes, it is not necessary to present the results of A23187.

In addition, the chosen colors to illustrate the different experimental conditions are confusing. Please change.

3-In the limitation part of their manuscript, the authors should discuss the effect of FR treatment on LV pressure, which is not necessarily beneficial for PAH patients.

Minor:

1-In line 28, page 13, it is not a mouse but a rat.

2-Authors should add the numbered of each figure

Referee #3 (Comments on Novelty/Model System for Author):

Well revised and is ready for acceptance.

Referee #3 (Remarks for Author):

The authors have performed additional experiments, and the new data are very supportive of the conclusion. All other critiques seemed to be well-addressed, and this reviewer does not have any concerns about publishing this work.

Scientific Editor comments

1) Author contributions: Please remove from the manuscript and specify author contributions in our submission system. CRediT has replaced the traditional author contributions section because it offers a systematic machine-readable author contributions format that allows for more effective research assessment. You are encouraged to use the free text boxes beneath each contributing author's name to add specific details on the author's contribution. More information is available in our guide to authors:

Response: We removed the author contribution section from our manuscript and specified author contribution in the submission system.

2) In the Materials and Methods, please take care of the following:

- Studies with human research participants: The use of human samples requires information on the authority granting ethics approval (e.g. IRB) and informed consent. If the need for approval is waived, please cite the reason (e.g. non-human subject research because the samples used were de-identified/coded with no identifying information) and legislation in the relevant methods section. Please also state that the experiments conformed to the principles set out in the WMA Declaration of Helsinki and the Department of Health and Human Services Belmont Report. Finally, please update the "Author Checklist" to indicate that this information was included in the manuscript.

Response: We added the following information to the Materials and Methods section:

The human donor lungs in this study were purchased from the International Institute for the Advancement of Medicine (IIAM) and de-identified. Since no human subject is involved in the study, ethics (or IRB) approval is waived. Likewise, the WMA Declaration of Helsinki and the Department of Health and Human Services Belmont Report are not applicable to the study (p.25, ll.12-15).

The Author Checklist was updated as requested.

- Animals: Please ensure that an ethics statement and the approval committee for research on animals is included in the section where animal experiments are described in the Materials and Methods. Please also ensure that specific housing conditions as well as gender, age, and origin of the animals involved in experiments is reported (beyond a statement that the experiments complied). The specific conditions are currently missing for the mouse experiments, and no ethics or additional information has been given for the pig experiments.

Response: We specified the conditions of mouse experiments. The lungs from adult pigs of either sex (~50 kg) were collected from a local abattoir, therefore use of the tissue does not require ethical approval. We added this information in the animal experiment section (p.29, ll. 7-13). The abattoir received pigs from different farms, that is why the housing conditions are unknown.

3) Please place individual sections of the manuscript in the following order: Title page - Abstract & Keywords - Introduction - Results - Discussion - Materials & Methods - Data Availability - Acknowledgements - Disclosure and Competing Interests Statement - The Paper Explained - For More Information - References - Figure Legends - Expanded View Figure Legends.

Response: We changed the order as requested.

4) For the figures and figure legends, please take care of the following:

- Please make sure to update the callouts of all figures in the main manuscript text (currently figure callouts are missing for Figure 4e)

- Please note that information related to n is missing in the legends of figures 5m; 7a-h; EV 1h; EV 4e-f; EV 5d.

- Although 'n' is provided, please describe the nature of entity for 'n' in the legends of figures 1f; 2c; 4d, f; 5c, f, h; EV 1c-f; EV 4c, EV 5b-c.

- Please note that the error bars are not defined in the legends of EV 2c-g.

Response: We added the missing callout and n-numbers as well as the nature of entity for "n" in the indicated figure legends. Furthermore, we added the definition of error bars in Fig. EV2.

5) Synopsis:

- Synopsis text: The standfirst is over the limit of maximum of 300 characters (including spaces), please shorten this sentence. We would suggest simply removing the first sentence.

- Please check your synopsis text and image before submission with your revised manuscript.

Please be aware that in the proof stage minor corrections only are allowed (e.g., typos).

Response: We removed the first sentence and checked the synopsis text and image.

6) For more information: This space should be used to list relevant web links for further consultation by our readers. Could you identify some relevant ones and provide such information as well? Some examples are patient associations, relevant databases, OMIM/proteins/genes links, author's websites, etc...

Response: We provided web links for further information (p. 32, II.2-4).

7) As part of the EMBO Publications transparent editorial process initiative, EMBO Molecular Medicine will publish online a Peer Review File (PRF) to accompany accepted manuscripts. This file will be published in conjunction with your paper and will include the anonymous referee reports, your point-by-point response and all pertinent correspondence relating to the manuscript. Let us know whether you agree with the publication of the PRF and as here, if you want to remove or not any figures from it prior to publication. Please note that the Authors checklist will be published at the end of the PRF.

We agree with the publication of the complete correspondence.

8) Please provide a point-by-point letter INCLUDING my comments as well as the reviewer's reports and your detailed responses (as Word file).

Done.

Referee #1 (Comments on Novelty/Model System for Author):

The authors performed a "tour de force" of new experiments in vitro, ex vivo and in vivo to address and answer all reviewer's questions. The number and quality of the technical approaches used here to answer the question whether Gq inhibition can attenuate PH is impressive and few laboratories are able to combine such different approaches in such careful dedicated way and at such a high level.

Referee #1 (Remarks for Author):

The study entitled "Pharmacological Gq inhibition induces strong pulmonary vasorelaxation and reverses pulmonary hypertension" by Alexander Seidinger et al tested the pulmonary vasodilating effect of the macrocyclic depsipeptide FR900359 (FR), a highly specific inhibitor of Gq/11/14 signalling, and its potential in the treatment of pulmonary hypertension (PH). Pulmonary hypertension (PH), with elevated pulmonary arterial pressure (mPAP {greater than or equal to} 20 mmHg) and pulmonary vascular resistance, affects up to 100 million people worldwide. Patients with PAH (PH Group 1) are presently treated with drugs targeting three dysfunctional endothelial pathways: augmentation of the nitric oxide (NO)/cyclic GMP pathway (soluble guanylyl cyclase activators, phosphodiesterase 5 inhibitors), endothelin-1 (ET-1) receptor antagonists; and prostacyclin analogs. Despite therapeutical progresses, PAH remains a fatal disease with a 3-year survival rate of 58%. Treatment of groups 2 to 5 PH addresses the causal disease and PAH therapeutics are presently not recommended. However, the presence of even moderate PH increases morbidity and mortality. Hence, new approaches to the management of all these patients are urgently needed.

The pathophysiology of PH is multifactorial and includes chronic or recurrent hypoxia and augmented (peri)vascular expression of vasoconstrictors, growth factors and inflammatory mediators, such as endothelin, serotonin, thromboxane A2 and others. Ultimately, this leads to endothelial cell dysfunction and increased contraction, proliferation, and apoptosis resistance of pulmonary artery smooth muscle cells (PASMCs) with remodeling of small pulmonary arterioles and muscularization of very distal, normally non-muscularized vessels. Here, the authors combined elegant studies in isolated murine and porcine pulmonary arteries, cultured mouse PASMCs and precision-cut slices from human and murine lungs to study the vasodilatory efficacy and potency of the specific Gq inhibitor FR on serotonin-, thromboxane- and endothelin-1-evoked vasoconstrictions. The results demonstrated that FR markedly prevents and reverses the vasoconstrictor effects of these factors in the pulmonary macro- and microcirculation. Moreover, FR also inhibits the growth factor (e.g. PDGF-BB) stimulated proliferation of cultured PASMC. This is important because besides vasoconstriction, the hyperproliferation of vascular mural cells and vascular remodeling/thickening have a major role in the development of PH.

These interesting observations were extended with experiments in isolated perfused murine lungs. The results demonstrated that FR does not inhibit hypoxia-driven vasoconstrictions, but efficiently counter-regulates vasoconstrictions elicited by agonists acting on Gq-coupled receptors.

To elucidate the relevance of this observations for the therapy of PH, the authors firstly tested the effect of local, intratracheal application of FR on acute serotonin-evoked pulmonary vasoconstrictions in anesthetized mice (the subsequent increases in pulmonary arterial pressure, PAP, were indirectly measured through recordings of right ventricular systolic pressure, RVSP). Notably, FR markedly reduced serotonin-evoked increases in RVSP and this effect was preserved in mice with hypoxia-induced PH. Lastly, the authors studied the effect of FR in the preclinical model of Sugen/hypoxia (SuHx)-driven chronic PH

in mice. Notably, FR markedly prevented and even reverted Sugen/HOX-driven PH, pulmonary vascular remodeling as well as right ventricular hypertrophy. Taken together, the results from these complex experiments in vitro/ex vivo and in vivo indicate the potential of pharmacological inhibition of Gq proteins in the treatment of PH. The possible human relevance is emphasized by the results obtained from experiments with "cultured" precision-cut slices from human lungs. This is a very elegant study with high clinical relevance. The experiments and results are well described and properly discussed. The authors performed a "tour de force" of new experiments in vitro and in vivo to address and answer all reviewer's questions. The number and quality of the technical approaches used here to answer the question whether Gq inhibition can attenuate PH is impressive and few laboratories are able to combine such different approaches at such a high level. The introduction and discussion sections have also been improved to address reviewers' comments. In particular the possible systemic effects of pharmacological Gq inhibition have been carefully discussed. In my view, this manuscript will be very interesting for clinicians and scientists working in the fields of lung pathophysiology and vascular physiology and disease.

Response: We thank the reviewer for the very positive comments and the appreciation of our work.

Referee #2 (Comments on Novelty/Model System for Author):

Authors strongly improve the quality of their manuscript. We greatly appreciate the authors' efforts to respond to our concerns.

However, we still have some concerns regarding the news of Calcium experiments, which are a little bit strange. We also asked the authors to write a limitation section because they did not use a severe model of PAH and because they found that FR treatment reduced LV blood pressure, which is not necessarily beneficial for PAH patients.

Referee #2 (Remarks for Author):

Authors strongly improve the quality of their manuscript. The results presented in the revised manuscript are still exciting, and we greatly appreciate the authors' efforts to respond to our concerns.

However, we have still some concerns.

Major:

1-We understand that the authors did not perform in vivo pharmacological experiments in more severe animal models of PAH (Monocrotaline rats or Su/Hx rats). In our opinion, these experiments should be crucial for a better translational perspective. The authors should at least write a limitation section regarding the use of only the mouse model of PAH.

Response: We thank the reviewer for this comment. As requested by the reviewer we included a limitations paragraph discussing the added value of testing FR in more severe animal models of PH and potential side-effects of FR on LV/systemic blood pressure. (p. 19, ll. 3-7).

2-Regarding the new Ca²⁺ imaging experiments. We do not understand the presented traces. Maybe authors should better legend these traces. When drug is applied? What

corresponds to the second phase of these Ca²⁺ signals? The authors should perform these Ca²⁺ experiments in a Ca²⁺-free medium to determine where this second phase comes from.

In the text, the authors explain that they used A23187. Does it correspond to the second phase of the Ca²⁺ response? If yes, it is not necessary to present the results of A23187. In addition, the chosen colors to illustrate the different experimental conditions are confusing. Please change.

Response: In the Ca²⁺ imaging experiments we measured Ca²⁺ responses in mPASMCs pre-treated for 1h with the solvent DMSO (Fig. 7E) or FR (Fig. 7F). After 20 sec of baseline read several stimuli were applied (indicated by an arrow) and the acute Ca²⁺ responses recorded. The response to buffer is depicted by the blue dotted line, the response to 5-HT alone by the orange dotted line, the response to PDGF alone is indicated by the green dotted line and the response to PDGF + 5-HT (PH condition) is depicted by the black dotted line. PDGF+5-HT administration (PH condition) resulted in a two-phased Ca²⁺ response, an immediate relatively small Ca²⁺ increase with a maximum at about 35 sec (quantified in Fig. 7G) and a delayed much stronger Ca²⁺ elevation with a maximum at about 90 sec (quantified in Fig. 7H). Control experiments with 5-HT or PDGF alone revealed that the immediate small Ca²⁺ response was evoked by 5-HT (orange line) and the delayed stronger response by PDGF (green line) (Fig. 7E). In accordance with this the pharmacological Gq inhibitor FR inhibited the immediate, relatively small Ca²⁺ increase while it had no impact on the delayed stronger Ca²⁺ increase (Fig. 7F).

These data clearly show that in the PH condition the first immediate small Ca²⁺ response is caused by 5-HT and Gq-dependent while the second delayed stronger Ca²⁺ increase is induced by PDGF and Gq-independent. To make this clearer we extended our description of the experiment in the results section of the revised manuscript (p.14, ll. 8-20).

The Ca²⁺ ionophore A23187 was applied in separate wells parallel to the PDGF/5-HT application after DMSO or FR pre-treatment to check the viability of the cells. These data are not shown in the figure. To prevent confusion, we deleted the sentence on A23187 controls from the materials & methods section. The colors of the traces in the figure were changed. Experiments in Ca²⁺ free media are beyond the scope of the current manuscript.

3-In the limitation part of their manuscript, the authors should discuss the effect of FR treatment on LV pressure, which is not necessarily beneficial for PAH patients.

Response: As requested by the reviewer we included a limitations paragraph discussing the added value of testing FR in more severe animal models of PH and potential side-effects of FR on LV/systemic blood pressure (p.19, ll. 3-7).

Minor:

1-In line 28, page 13, it is not a mouse but a rat.

Response: We thank the reviewer for this comment and changed mouse to rat.

2-Authors should add the numbered of each figure

Response: We now always included n number in all figure legends.

Referee #3 (Comments on Novelty/Model System for Author):

Well revised and is ready for acceptance.

Referee #3 (Remarks for Author):

The authors have performed additional experiments, and the new data are very supportive of the conclusion. All other critiques seemed to be well-addressed, and this reviewer does not have any concerns about publishing this work.

Response: We thank the reviewer for the very positive comments.

5th Jun 2024

Dear Prof. Wenzel,

We are pleased to inform you that your manuscript is accepted for publication and is now being sent to our publisher to be included in the next available issue of EMBO Molecular Medicine.

Yours sincerely,

Poonam Bheda, PhD
Scientific Editor
EMBO Molecular Medicine
